# Transformers as Measure-Theoretic Associative Memory: A Statistical Perspective and Minimax Optimality

**Ryotaro Kawata**[1,2,*], **Taiji Suzuki**[1,2,§]

[1]Department of Mathematical Informatics, University of Tokyo, Japan
[2]Center for Advanced Intelligence Project, RIKEN, Japan
[*]`kawata-ryotaro725@g.ecc.u-tokyo.ac.jp`
[§]`taiji@mist.i.u-tokyo.ac.jp`

## Abstract

Transformers excel through content-addressable retrieval and the ability to exploit contexts of, in principle, unbounded length. We recast associative memory at the level of probability measures, treating a context as a distribution over tokens and viewing attention as an integral operator on measures. Concretely, for mixture contexts $\nu = I^{-1} \sum_{i=1}^{I} \mu^{(i^*)}$ and a query $x_{\mathrm{q}}(i^*)$, the task decomposes into (i) recall of the relevant component $\mu^{(i^*)}$ and (ii) prediction from $(\mu_{i^*}, x_{\mathrm{q}})$. We study learned softmax attention (not a frozen kernel) trained by empirical risk minimization and show that a shallow measure-theoretic Transformer composed with an MLP learns the recall-and-predict map under a spectral assumption on the input densities. We further establish a matching minimax lower bound with the same rate exponent (up to multiplicative constants), proving sharpness of the convergence order. The framework offers a principled recipe for designing and analyzing Transformers that recall from arbitrarily long, distributional contexts with provable generalization guarantees.

## 1 Introduction

Transformers (Vaswani et al., 2017) have achieved strong empirical performance across natural language (Brown et al., 2020), vision (Dosovitskiy et al., 2021), and speech/audio (Dong et al., 2018). Two properties motivate our study: (i) *content-addressable retrieval* of associated information—an associative-memory view of attention—and (ii) the ability to leverage contexts of variable, in principle unbounded, length.

In this work, we cast associative memory at the *level of probability measures*, treating context as a distribution over tokens, and develop a rigorous statistical analysis of learned softmax-attention Transformers in this measure-theoretic setting.

Associative memory provides a unifying lens on how neural systems store and retrieve from partial cues: from early self-organizing and correlation memories to Hopfield attractors (Amari, 1972; Kohonen, 1972; Nakano, 1972; Hopfield, 1982; 1984). Transformers recast associative memory or recall via content-addressable attention, formally equivalent to Hopfield-style associative updates (Vaswani et al., 2017; Ramsauer et al., 2021). Recent studies quantify memory emergence and capacity (Bietti et al., 2023; Cabannes et al., 2024; Mahdavi et al., 2024; Kim et al., 2023; Jiang et al., 2024; Nichani et al., 2025).

As Transformers are engineered to ingest massive text corpora and long contexts, researchers have formalized this "context" as a *probability measure* over tokens, yielding a measure-theoretic handle on variable-size inputs. Summarizing the text data as one measure by the law of large numbers helps them to show results that are independent of the text length. A measure-theoretic view of Transformers formalizes attention as a map on distributions, enabling analysis of stability and emergent structure (Vuckovic et al., 2020; Sander et al., 2022; Geshkovski et al., 2025; Burger et al., 2025). On the expressivity side, Transformers can interpolate between input/output measures (Geshkovski

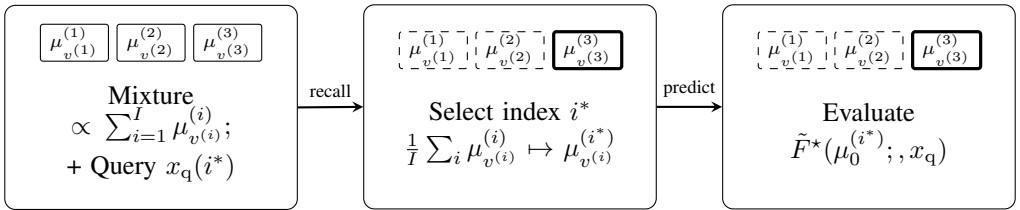

Figure 1: Associative recall at the level of measures (informal): the query $x_{\mathrm{q}}(i^*)$ selects the relevant component measure $\mu_{v^{(i^*)}}^{(i^*)}$ from the mixture $\nu \propto \sum_{i=1}^{I} \mu_{v^{(i)}}^{(i)}$, followed by prediction from $(\mu_0^{(i^*)}, x_{\mathrm{q}})$. Note that each $\mu_{v^{(i)}}^{(i)}$ is constructed by $v^{(i)} \in \mathbb{S}^{d_1}$ and a measure $\mu_0^{(i)}$ on $\mathbb{R}^{d_2}$.

et al., 2024) and even uniformly approximate continuous in-context mappings where the context is itself a probability distribution (Furuya et al., 2025).

Recent work has developed statistical analyses of Transformers with infinite-dimensional inputs. Yet the link to *associative memory*—arguably a defining feature of attention—remains under-specified. Prior generalization results in distribution regression typically assume a *frozen* (non-learnable) attention kernel (Liu & Zhou, 2025), leaving unclear how *learned* attention retrieves the associated measure. Likewise, in a sequence-based in-context setting with infinite-dimensional inputs (Kim et al., 2024), the analysis was carried out under linear attention, whose limited expressiveness makes it difficult to realize the sharp, spiky weight distributions achievable by softmax attention (Han et al., 2024; Fan et al., 2025). These considerations motivate the central question:

**Q.** *Can a learned softmax-attention Transformer recall an infinite-dimensional (measure-valued) context and predict from it with provable generalization guarantees?*

**"Associative memory" at the level of measures (informal).** Consider a text corpus composed of $I$ documents. We model each token as a vector $x = (v, z) \in \mathbb{R}^{d_1} \times \mathbb{R}^{d_2}$, where $v$ encodes a document-level feature (e.g., topic) and $z$ encodes token-level content. For document $i$, the document feature is a fixed vector $v^{(i)} \in \mathbb{S}^{d_1}$, while the content part $z$ is sampled from some distribution $\mu_0^{(i)}$ on $\mathbb{R}^{d_2}$. In the limit of an infinitely long document, the empirical token distribution of document $i$ converges to a probability measure $\mu_{v^{(i)}}$ on $\mathbb{R}^{d_1+d_2}$, namely the law of $x = (v^{(i)}, Z)$ with $Z \sim \mu_0^{(i)}$. The *context* seen by the model is then the mixture

$$\nu = \sum_{i=1}^{I} w_i \, \mu_{v^{(i)}}^{(i)}, \qquad w_i \geq 0, \ \sum_i w_i = 1,$$

representing a whole dataset containing many documents. Given such a mixture $\nu$ and a query $x_{\mathrm{q}} \in \mathbb{R}^{d_1+d_2}$ whose first $d_1$ coordinates align with some document feature $v^{(i^*)}$, the desired "associative memory" behavior is:

- first, *recall* the component indexed by $i^*$ from the mixture $\nu$, and

- then *predict* a scalar quantity that depends only on the associated content distribution $\mu_0^{(i^*)}$ (and possibly on $x_{\mathrm{q}}$).

We denote by $F^\star : (\nu, x_{\mathrm{q}}) \mapsto F^\star(\nu, x_{\mathrm{q}}) \in \mathbb{R}$ this ground-truth recall-and-predict map: by construction, its output depends on $\nu$ only through the single component $\mu_0^{(i^\star)}$ selected by the query (Fig. 1). We study learned softmax-attention Transformers (with an integral/empirical measure view of attention) trained by empirical risk minimization (ERM) to implement this recall-and-predict pipeline. On the statistical side, we work in a very smooth regime: we endow the space of context measures with a reproducing kernel Hilbert space (RKHS) whose Mercer eigenvalues satisfy $\lambda_j \asymp \exp(-c\,j^\alpha)$ for some $\alpha > 0$ (as for Gaussian-type kernels (Schölkopf & Smola, 2002)). This spectral decay encodes strong smoothness of the underlying densities and induces an *effective dimension* that will govern our learning rates.

**Contributions.** We now outline the principal contributions of this work:

1. **Associative memory at the level of measures.** We formalize a general, mathematically rigorous framework for associative recall over measures: given a measure-valued context and a query, a *recall operator* selects the associated measure, and a *predictor* maps the recalled measure together with the query to an output. We formalize query-conditioned selection from arbitrarily long contexts: the model recalls the associated probability measure capturing the relevant content and predicts from its statistics.

2. **Generalization.** We show that a shallow (depth-2) measure-theoretic Transformer composed with an MLP can learn the recall-and-predict mapping at the level of measures (Theorem 1). In contrast to linear attentions (Kim et al., 2024) or frozen kernels (Zhou et al., 2024), softmax attention enables *sparse* and *adaptive* recall of the relevant measure. For empirical risk minimization over a bounded-parameter hypothesis class—provided the number of recall candidates is not excessively large—we establish the *sub-polynomial* population-risk bound $\exp\{-\Theta((\log n)^{\alpha/(\alpha+1)})\}$, showing that the statistical difficulty is governed by the kernel's Mercer eigen-decay $\alpha$.

3. **Minimax Optimality.** We prove a minimax lower bound with the same rate exponent $(\log n)^{\alpha/(\alpha+1)}$, establishing the sharpness of the convergence order (Theorem 2). Thus, under our spectral and mixture-growth assumptions, the proposed measure-theoretic Transformer is minimax-optimal *in the order of the exponent*, though multiplicative constants may differ.

The remainder of this paper is organized as follows. Section 2 reviews related work. Section 3 introduces the problem setting and the student model. Section 4 presents our main theoretical results. Section 5 concludes with discussion. Technical details and proofs are deferred to the appendices.

## 2 RELATED WORK

**Associative Memory and Recall.** Associative memory concepts originated in early neuroscience models (Hopfield, 1982; Amari, 1972; Kohonen, 1972; Nakano, 1972; Hopfield, 1982; 1984), followed by Graves et al. (2014); Weston et al. (2014); Ramsauer et al. (2021); Millidge et al. (2022). The Transformer architecture is closely related to associative memory by employing self-attention as a content-addressable mechanism (Vaswani et al., 2017). Recent work has increasingly focused on how associative memory emerges and scales within Transformer architectures (Bietti et al., 2023; Cabannes et al., 2024; Mahdavi et al., 2024; Kim et al., 2023; Jiang et al., 2024; Nichani et al., 2025).

**Transformers for Infinite-Dimensional Inputs.** A measure-theoretic perspective has enabled insightful analysis of Transformer architectures. Vuckovic et al. (2020); Sander et al. (2022) formalized self-attention as a map on probability measures. Its Lipschitzness is explored in Castin et al. (2024). Building on that framework, Geshkovski et al. (2023; 2025); Burger et al. (2025) modeled self-attention as an interacting particle system. Geshkovski et al. (2024) proved a universality result showing that Transformers can interpolate arbitrary input–output measure pairs, later strengthened by Furuya et al. (2025) to uniform approximation of continuous mappings over distributions and queries. On generalization, Liu & Zhou (2025) studied distribution regression, though restricted to a frozen attention kernel. In the context of sequential, infinite-dimensional inputs, Kim et al. (2024) studied in-context learning with linear attention, which essentially reduces to averaging behaviors; hence their analysis assumed *relaxed sparsity* and *orthonormality* of the recall candidates, reflecting the difficulty of achieving spiky one-hot recall in contrast to softmax attention (Han et al., 2024; Fan et al., 2025).

**MLP Approximations of Functional Mappings.** In statistical learning, Mhaskar & Hahm (1997) laid the groundwork by showing that multi-layer perceptrons (MLPs) can approximate continuous nonlinear functionals over function spaces in a optimal rate that was generalized by Stinchcombe (1999). Rossi et al. (2005) introduced a novel functional MLPs which is applicable to functional data, followed by variants (Yao et al., 2021; Song et al., 2023; Zhou et al., 2024). On the optimization front, Suzuki (2020); Nishikawa et al. (2022) established global optimization assurances for two-layer networks operating in a infinite-dimensional regime.

## 3   PROBLEM SETTING

**Notations.**   For integers $N_1 \leq N_2$ and $\boldsymbol{v} \in \mathbb{R}^N$, we write $\boldsymbol{v}_{N_1:N_2} = (v_{N_1}, \ldots, v_{N_2})^\top$. For a matrix $A$, $\|A\|_0$ denotes the number of nonzero entries and $\|A\|_\infty = \max_{i,j} |A_{i,j}|$. We write $\lambda$ for the Lebesgue measure on $\mathbb{R}^N$, and $f_\sharp \mu$ for the pushforward of $\mu$ by $f$. For a measurable space $\mathcal{X}$, $\mathcal{P}(\mathcal{X})$ denotes the set of probability measures and $\mathcal{M}_+(\mathcal{X})$ the set of nonnegative measures. We use $(\Omega, \mathcal{F}, \mathbb{P})$ for a probability space, $\|f\|_{L^p}$ and $\|f\|_\infty$ for the usual $L^p$ and essential sup norms, and $\mathbb{P}_X$ for the law of a random variable $X$. Expectations are written $\mathbb{E}[\cdot]$ or $\mathbb{E}_X[\cdot]$ with the law of $X$.

**Our Regression Problem.**   We now formalize the informal recall-and-predict scenario from the introduction.

**Definition 1.** Let $\mathcal{X}_0 \subset \mathbb{R}^{d_2}$ be a bounded token-content space and let $\mathcal{X} \subset \mathbb{R}^{d_1+d_2}$ denote the token space with the decomposition $x = (v, z)$, where $v \in \mathbb{R}^{d_1}$ encodes a document-level feature and $z \in \mathcal{X}_0$ encodes token-level content.

*1. Mixture contexts and queries.* Each document $i \in [I]$ is associated with a document feature $v^{(i)} \in \mathbb{S}^{d_1-1}$ and a content distribution $\mu_0^{(i)} \in \mathcal{M}_+(\mathcal{X}_0)$. Informally, $\mu_0^{(i)}$ represents the distribution of token contents (e.g., words or embeddings) appearing in document $i$. The corresponding token distribution on $\mathcal{X}$ is the product measure

$$\mu_{v^{(i)}}^{(i)} := \delta_{v^{(i)}} \otimes \mu_0^{(i)} = (\mathrm{Emb}_{v^{(i)}})_\sharp \mu_0^{(i)},$$

where $\mathrm{Emb}_{v^{(i)}}(z) := (v^{(i)}, z)$ and $f_\sharp \mu$ denotes the pushforward of $\mu$ by $f$, so that $\mu_{v^{(i)}}^{(i)}$ is the joint distribution of the document feature $v^{(i)}$ and the token content in document $i$. A *context* is a mixture of these component measures,

$$\nu := \frac{1}{I} \sum_{i=1}^{I} \mu_{v^{(i)}}^{(i)} \in \mathcal{P}(\mathcal{X}),$$

which represents the token distribution of a whole dataset containing $I$ documents. Concretely, $\nu$ is the law obtained by first sampling a document $i \in [I]$ at random and then a token $x = (v^{(i)}, z)$ from that document. Given such a mixture, a query $x_{\mathrm{q}} \in \mathcal{X}$ is constructed so as to indicate a *distinguished* index $i^\star \in [I]$. For concreteness, we take

$$x_{\mathrm{q}} := \begin{bmatrix} v^{(i^\star)} \\ 0_{d_2} \end{bmatrix} = \mathrm{Emb}_{v^{(i^\star)}}(0_{d_2}) \in \mathbb{R}^{d_1+d_2},$$

that is, the document feature $v^{(i^\star)}$ padded with zeros in the last $d_2$ coordinates.[1]

*2. Ground-truth recall-and-predict map.* The learning task is to predict a real-valued response

$$y = F^\star(\nu, x_{\mathrm{q}}) + \xi, \qquad \xi \sim \mathcal{N}(0, \sigma^2),$$

from the pair $(\nu, x_{\mathrm{q}})$. The key structural assumption is that $F^\star$ depends on the context $\nu$ only through the single component associated with the index $i^\star$ selected by the query. Equivalently, there exists a (hidden) functional $\tilde{F}^\star$ such that

$$F^\star(\nu, x_{\mathrm{q}}) = \tilde{F}^\star\big(\mu_0^{(i^\star)}, x_{\mathrm{q}}\big), \qquad \nu = \frac{1}{I} \sum_{i=1}^{I} \mu_{v^{(i)}}^{(i)},$$

so that the regression map $(\nu, x_{\mathrm{q}}) \mapsto F^\star(\nu, x_{\mathrm{q}})$ decomposes into the two conceptual stages

$$\big(\nu, x_{\mathrm{q}}\big) \xrightarrow{\text{recall } i^\star} \mu_0^{(i^\star)} \xrightarrow{\text{predict}} \tilde{F}^\star\big(\mu_0^{(i^\star)}, x_{\mathrm{q}}\big).$$

For instance, $F^\star(\nu, x_{\mathrm{q}})$ could be a sentiment score of document $i^\star$, or the probability that document $i^\star$ mentions an entity specified by the query.

In particular, $F^\star$ must first *associate* the query with the relevant component of the mixture context and then *predict* a scalar from the recalled component (e.g., in-context learning (Brown et al., 2020)). This formalizes an "associative memory" task at the level of probability measures.

---

[1]More general queries, e.g. with a nonzero content part, can be treated as well; we fix the zero padding here for notational simplicity.

**Statistical Estimation Problem.** Motivated by the recall-and-predict regression task described above, We now formulate the associated statistical estimation problem. We observe $n$ i.i.d. samples

$$\mathcal{S}_n := \big\{(\nu_t, x_{\mathrm{q}_t}, y_t)\big\}_{t=1}^n$$

drawn from the joint distribution of $(\nu, x_{\mathrm{q}}, y)$. Let $\mathcal{F}$ denote a hypothesis class of measurable functions $F : \mathcal{P}(\mathcal{X}_0) \times \mathbb{R}^{d_1 + d_2} \to \mathbb{R}$. Given the training data, we define the empirical risk minimizer

$$\hat{F} := \arg\min_{F \in \mathcal{F}} \ \hat{\mathbb{E}}_n\big[\big(y - F(\nu, x_{\mathrm{q}})\big)^2\big],$$

where $\hat{\mathbb{E}}_n$ denotes the empirical expectation over the $n$ samples. Given a hypothesis (regressor) $\hat{F}$, our goal is to learn $F^\star$ so as to minimize the squared $L^2$ loss

$$R(F^\star, \hat{F}) := \mathbb{E}_{\mathcal{S}_n}\big[\big\|F^\star(\nu, x_{\mathrm{q}}) - \hat{F}(\nu, x_{\mathrm{q}})\big\|_{L^2(\mathbb{P}_{\nu, x_{\mathrm{q}}})}^2\big].$$

**RKHS viewpoint on content measures.** Before stating the assumptions, we briefly recall how the kernel $K$ induces a function space for modeling token distributions. Given a positive definite kernel $K : \mathcal{X}_0 \times \mathcal{X}_0 \to \mathbb{R}$ on the bounded token-content domain $\mathcal{X}_0 \subset \mathbb{R}^{d_2}$, there exists a unique reproducing kernel Hilbert space (RKHS) (e.g., (Schölkopf & Smola, 2002)) $\mathcal{H}_0$ of functions $f : \mathcal{X}_0 \to \mathbb{R}$ such that

$$K(\cdot, x) \in \mathcal{H}_0 \quad \text{and} \quad f(x) = \langle f, K(\cdot, x)\rangle_{\mathcal{H}_0} \quad \text{for all } x \in \mathcal{X}_0.$$

Intuitively, $\mathcal{H}_0$ is the class of "smooth" functions compatible with $K$. In the Mercer basis

$$K(x, x') = \sum_{j \geq 1} \lambda_j e_j(x) e_j(x'),$$

every $f \in \mathcal{H}_0$ can be written as $f(x) = \sum_{j \geq 1} b_j e_j(x)$ with finite RKHS norm $\|f\|_{\mathcal{H}_0}^2 = \sum_{j \geq 1} b_j^2 / \lambda_j$. Large coefficients $b_j$ in high-frequency directions (small $\lambda_j$) are penalized heavily, so $\mathcal{H}_0$ favours functions whose energy is concentrated on low-order eigen-components. In our setting we represent each content measure $\mu_0^{(i)}$ by its density $p_{\mu_0^{(i)}}$ on $\mathcal{X}_0$, and we require these densities to lie in a fixed ball of $\mathcal{H}_0$. The rapid eigenvalue decay $\lambda_j \asymp \exp(-cj^\alpha)$ corresponds to a very smooth (Gaussian Kernel-type (Schölkopf & Smola, 2002)) regime in which the effective dimension of this function class is small; this effective dimension will drive our statistical rates.

### 3.1 ASSUMPTIONS AND TECHNICAL SETTINGS

We now state the high-level assumptions used in our upper- and lower-bound analyses. Full technical versions are deferred to Appendix A.1. Throughout this subsection, contexts and queries are generated as described in the beginning of Section 3.

**Assumption 1** (Smooth kernel and regular content measures). Let $\mathcal{X}_0 \subset \mathbb{R}^{d_2}$ be a bounded token-content domain and let $K : \mathcal{X}_0 \times \mathcal{X}_0 \to \mathbb{R}$ be a positive definite kernel with Mercer expansion (Schölkopf & Smola, 2002)

$$K(x, x') = \sum_{j \geq 1} \lambda_j e_j(x) e_j(x').$$

We assume (i) The eigenvalues decay exponentially, i.e., $\lambda_j \asymp \exp(-cj^\alpha)$ for some $c, \alpha > 0$ (a Gaussian-type smoothness regime), (ii) Each content distribution $\mu_0^{(i)}$ is a finite measure on $\mathcal{X}_0$ whose density lies in a fixed ball of the RKHS $\mathcal{H}_0$ induced by $K$. Informally, the token distributions are very smooth and their effective dimension is small, since most of the mass lies in low-order eigen-components. We focus on this exponentially decaying regime as a first step, to keep the analysis transparent.

**Example 1** (Heat-kernel RKHS (Grigor'yan, 2006)). *Consider $\mathcal{X}_0 = [0, 1]$ and the Laplace operator $\Delta = \frac{\mathrm{d}^2}{\mathrm{d}x^2}$ with Dirichlet boundary conditions. Its eigenfunctions and eigenvalues are $e_k(x) = \sqrt{2}\sin(k\pi x)$ and $\zeta_k = (k\pi)^2$ for $k \geq 1$. The heat kernel, the fundamental solution of the heat equation describing how heat placed at $y$ at time $0$ spreads to $x$ by time $t$, is*

$$K_t(x, y) = \sum_{k=1}^\infty e^{-\zeta_k t} e_k(x) e_k(y) = \sum_{k=1}^\infty e^{-\pi^2 k^2 t} e_k(x) e_k(y),$$

*so the Mercer eigenvalues satisfy $\lambda_k = e^{-\pi^2 k^2 t} \asymp \exp\big(-ck^2\big)$ and Assumption 1 holds with $\alpha = 2$. More general constructions on compact manifolds are recalled in Appendix A.1.*

**Assumption 2** (Separated context vectors). The context vectors $v^{(i)} \in \mathbb{S}^{d_1-1}$ used to construct the mixture (1) are well separated: $\langle v^{(i)}, v^{(i')} \rangle \leq 0$, for all $1 \leq i < i' \leq I$, and we assume $I \leq d_1$. This guarantees that different documents are sufficiently distinguishable for the recall step[2].

**Assumption 3** (Lipschitz ground-truth functional). There exists a metric $d_{\mathrm{prod}}$ on the space of pairs $(\mu_0, x)$, induced by the RKHS structure above, such that the hidden functional $\tilde{F}^\star : \{\mu_0^{(i)}\} \times \mathbb{R}^{d_1+d_2} \to \mathbb{R}$ is $L$-Lipschitz:

$$\left| \tilde{F}^\star(\mu_0, x) - \tilde{F}^\star(\mu_0', x') \right| \leq L \, d_{\mathrm{prod}}\big((\mu_0, x), (\mu_0', x')\big)$$

for all admissible inputs $(\mu_0, x)$ and $(\mu_0', x')$. In the proofs, $d_{\mathrm{prod}}$ will be the sum of an RKHS-induced distance between densities and the Euclidean distance between queries; see Appendix A.1 for its precise form.

**Assumption 4.** Each content distribution $\mu_0^{(i)}$ is a probability measure on $\mathcal{X}_0$.

**Setting 1** (Probability setting for upper bound (Setting 1)). Contexts and queries are generated as in Section 3. Assumptions 1, 2, 3 and 4 hold. In other words, we consider smooth (Gaussian-type) content probability distributions in an RKHS ball, well-separated context vectors, and an $L$-Lipschitz ground-truth functional with respect to an RKHS-induced metric.

**Structured model for lower bound.** For the minimax lower bound, we work with a simplified random model for the content densities, following Lanthaler (2024): the density of $\mu_0$ is generated by random coefficients in the Mercer expansion of $K$.

**Assumption 5** (Informal structural model for densities). Let $(\lambda_j, e_j)$ be the spectrum of $K$ as in Assumption 1. We assume that $\frac{\mathrm{d}\mu_0}{\mathrm{d}\lambda}(x) = \sum_{j \geq 1} \lambda_j^{\Theta(1)} Z_j e_j(x)$, where the coefficients $Z_j$ are independent, bounded random variables with unit variance. The full set of structural conditions is given in Assumption 8 in Appendix A.1.

**Setting 2** (Structured setting for lower bound (Setting 2)). Contexts and queries are generated as in Section 3. Assumption 1, 2 and 3 hold, and the densities of the content measures follow the structural model in Assumption 5. In this setting we derive minimax lower bounds under random Mercer coefficients.

## 3.2 STUDENT MODEL: MEASURE-THEORETIC TRANSFORMERS

We define our student model as a class of measure-theoretic Transformer architectures following Furuya et al. (2025).

**Measure-Theoretic Attention.** Given a set of tokens $X = (x_\ell)_{\ell=1}^w \in \mathbb{R}^{d_{\mathrm{attn}} \times w}$ and a query $x \in \mathbb{R}^{d_{\mathrm{attn}}}$ that encodes information about some of the tokens, a single unmasked attention head with parameters $\theta^{(h)} = (K^h, Q^h, V^h)$ in an "in-context" form (Furuya et al., 2025) computes

$$\mathrm{SAttn}_{\theta^{(h)}}(X, x) = \sum_j \mathrm{Softmax}\big(\langle Q^h x, K^h X \rangle\big) V^h x_j,$$

where $\mathrm{Softmax}((z_1, \ldots, z_N)) := (\exp(z_i)/\sum_j \exp(z_j))_{i=1}^N$. A standard multi-head attention with $H$ heads is then $\mathrm{MSAttn}_\theta(X) = \sum_{h=1}^H W^h \mathrm{SAttn}_{\theta^{(h)}}(X, x)$ with $W^h \in \mathbb{R}^{d_{\mathrm{attn}} \times d_{\mathrm{attn}}}$. In the unmasked case, attention is permutation-equivariant in the token indices. This allows us to represent the input set by its empirical measure, in particular, in the form of a mixture measure

$$\nu_X = \frac{1}{I} \sum_{i=1}^I \left( \frac{1}{w_i} \sum_{\ell_i=1}^{w_i} \delta_{v^{(i)}} \otimes \delta_{u_\ell^{(i)}} \right) \in \mathcal{P}(\mathbb{R}^{d_{\mathrm{attn}}}) \quad \text{in the limit as} \quad w \to \infty,$$

where $v^{(i)} \in \mathbb{R}^{d_1}$ (group-shared, possibly indicated by the query), $u_\ell^{(i)} \in \mathbb{R}^{d_2}$ (token-specific) for $i \in [1:I], \ell \in [1:w_i], \sum_i w_i = w$ and rewrite attention in a measure-theoretic form, where the integral replaces the discrete sum in the limit.

---

[2]A similar separation/orthogonality structure is used in theoretical analyses of factual extraction in transformers, e.g., Ghosal et al. (2024)

**Definition 2** (Measure-theoretic attention layer (Furuya et al., 2025))**.** Let $d_{\text{attn}}$ be the embedding dimension for the attention. The *measure-theoretic attention layer* $\text{Attn}_\theta : \mathcal{P}(\mathbb{R}^{d_{\text{attn}}}) \times \mathbb{R}^{d_{\text{attn}}} \to \mathbb{R}^{d_{\text{attn}}}$ is defined by

$$\text{Attn}_\theta(\nu, x) = Ax + \sum_{h=1}^{H} W^h \int \text{Softmax}(\langle Q^h x, K^h y \rangle) \, V^h y \, \mathrm{d}\nu(y),$$

where $\text{Softmax}(\langle Qx, Ky \rangle) \coloneqq \exp(\langle Qx, Ky \rangle) / \int \exp(\langle Qx, Kz \rangle) \, \mathrm{d}\nu(z)$. Here $A \in \mathbb{R}^{d_{\text{attn}} \times d_{\text{attn}}}$ applies a learned linear transformation to the skip connection.

When $\nu$ is an empirical mixture measure $\nu_X$, this reduces to the standard discrete attention layer (we do not pursue the discrete case in this work). We expect that the query $x$ indicates tokens in the $i^*$-th component in $\nu_X$ based on the group-shared vector $v^{(i^*)}$ and the model can recall them. We constrain these layers via the following bounded-parameter hypothesis class.

**Definition 3** (Attention hypothesis class)**.** For constants $B_a, B_a', S_a, S_a', F > 0$, define the class of $H$-head measure-theoretic attention layers

$$\mathcal{A}(d_{\text{attn}}, H, B_a, B_a', S_a, S_a') \coloneqq \big\{ \text{Attn}_\theta \mid \max_h \big\{ \|W^h\|_\infty, \|Q^h\|_\infty, \|K^h\|_\infty, \|V^h\|_\infty \big\} \leq B_a,$$

$$\max_h \big\{ \|W^h\|_0, \|Q^h\|_0, \|K^h\|_0, \|V^h\|_0 \big\} \leq S_a, \ \|A\|_\infty \leq B_a', \ \|A\|_0 \leq S_a', \ \|\text{Attn}_\theta\|_\infty \leq F \big\},$$

where $\|M\|_\infty \coloneqq \max_{i,j} |M_{ij}|$, $\|M\|_0$ is the number of non-zero entries, for a matrix $M$. We assume $W^h, Q^h, K^h, V^h$ are square matrices for simplicity. We write $\mathcal{A}(d_{\text{attn}}, H, B_a, S_a)$ when $B_a' = B_a$ and $S_a' = S_a$.

**MLP Layer.** In addition to attention, a Transformer block also includes a feedforward component. Since this part does not depend on the underlying measure $\mu$, we model it simply as a standard multilayer perceptron (MLP), defined below.

**Definition 4** (MLP hypothesis class)**.** Let $\ell \in \mathbb{N}$ be the depth and $\boldsymbol{p} = (p_0, p_1, \ldots, p_{L+1}) \in \mathbb{N}^{L+2}$ be the layer widths. A neural network with architecture $(\ell, \boldsymbol{p})$ is any function of the form

$$f : \mathbb{R}^{p_0} \to \mathbb{R}^{p_{\ell+1}}, \quad \mathbf{x} \mapsto f(\mathbf{x}) = W_\ell \sigma_{\mathbf{v}_\ell} W_{\ell-1} \sigma_{\mathbf{v}_{\ell-1}} \cdots W_1 \sigma_{\mathbf{v}_1} W_0 \mathbf{x},$$

where $W_i \in \mathbb{R}^{p_{i+1} \times p_i}$ is the weight matrix of layer $i$ and $\mathbf{v}_i \in \mathbb{R}^{p_i}$ is a shift (bias) vector applied through the activation $\sigma_{\mathbf{v}_i}(\mathbf{z}) \coloneqq \sigma(\mathbf{z} - \mathbf{v}_i)$ with ReLU activation function $\sigma : \mathbb{R} \to \mathbb{R}$. The hypothesis class of MLPs with architecture $(\ell, \boldsymbol{p})$ is denoted

$$\mathcal{F}(\ell, \boldsymbol{p}, s, F) \coloneqq \big\{ f \text{ of the form above} \mid$$

$$\max_j \{ \|W_j\|_\infty, |v_j|_\infty \} \leq 1, \textstyle\sum_j \|W_j\|_0 + |v_j|_0 \leq s, \ \|f\|_\infty \leq F \big\}.$$

**Transformer Hypothesis Class via Composition.** To formally describe a Transformer within our measure-theoretic framework, we introduce the notion of *composition* for measure-theoretic mappings, following Furuya et al. (2025). This will allow us to view a multi-layer Transformer as a successive composition of attention and feedforward layers, in exact analogy with the standard architecture.

The key idea is that such a mapping $\Gamma$ acts simultaneously on a token $x$ and on its generating distribution $\mu$. Given $z \sim \mu$, applying $\Gamma$ produces a transformed token $\Gamma(\mu, z)$ and induces a new distribution on transformed tokens $\Gamma(\mu, z)$, namely $(\Gamma(\mu, \cdot))_\sharp \mu$. Thus, composing two mappings means successively applying these joint transformations at both the sample and distribution levels.

**Definition 5** (Composition of measure-theoretic mappings)**.** Let $\Gamma_1 : \mathcal{P}(\mathbb{R}^{d_1^{(1)}}) \times \mathbb{R}^{d_1^{(1)}} \to \mathbb{R}^{d_1^{(2)}}$ and $\Gamma_2 : \mathcal{P}(\mathbb{R}^{d_2^{(1)}}) \times \mathbb{R}^{d_2^{(1)}} \to \mathbb{R}^{d_2^{(2)}}$ with $d_1^{(2)} = d_2^{(1)}$. Their composition is defined as

$$(\Gamma_2 \diamond \Gamma_1)(\nu, x) \coloneqq \Gamma_2(\mu_1, \Gamma_1(\nu, x)), \quad \text{where} \quad \mu_1 \coloneqq (\Gamma_1(\nu, \cdot))_\sharp \nu.$$

**Remark 1.** If we interpret $\nu$ as a limit of empirical measure $\nu_X = \lim_{w \to \infty} \frac{1}{w} \sum_{\ell=1}^{w} \delta_{x_\ell}$, then the composition $\Gamma_2 \diamond \Gamma_1$ acts by updating both the individual tokens and their empirical distribution. In this case, the construction is consistent with the standard layerwise composition in a Transformer: each layer maps the sequence of tokens $(x_1, \ldots, x_w)$ to a new sequence, while the corresponding empirical distribution is updated accordingly.

Building on the notion of measure-theoretic composition introduced above, we can now formally describe a Transformer as the successive composition of attention and feedforward layers. The following definition specifies the corresponding hypothesis class.

**Definition 6** (Transformer hypothesis class). For parameters $(d_j, H_j, B_{a,j}, B'_{a,j}, S_{a,j}, S'_{a,j}, \ell_j, \boldsymbol{p}_j, s_j)_{j=1}^L$, define TF as the set of mappings of the form

$$\mathrm{Attn}_{\theta_L} \diamond \mathrm{MLP}_{\xi_L} \diamond \cdots \diamond \mathrm{Attn}_{\theta_1} \diamond \mathrm{MLP}_{\xi_1},$$

where $\mathrm{Attn}_{\theta_j} \in \mathcal{A}(d_j, H_j, B_{a,j}, B'_{a,j}, S_{a,j}, S'_{a,j})$ and $\mathrm{MLP}_{\xi_j} \in \mathcal{F}(\ell_j, p_j, s_j)$. MLP layers are independent of $\mu$ (i.e. $\mathrm{MLP}_{\xi_1}(\mu, x) := \mathrm{MLP}_{\xi_1}(x)$), and all intermediate outputs are assumed uniformly bounded.

## 4 MAIN RESULTS

### 4.1 ESTIMATION ERROR OF MEASURE-THEORETIC TRANSFORMERS

We begin with the generalization performance of measure-theoretic Transformers in the recall-and-predict task. Throughout this subsection we work under the Probability Setting (Setting 1), where each content measure $\mu_0^{(i)}$ has a smooth RKHS density on $\mathcal{X}_0$ and the number of mixture components is not too large.

**Theorem 1** (Sub-polynomial convergence; informal version of Theorem 3). *Let $F^\star(\nu, x_{\mathrm{q}}) = \tilde{F}^\star(\mu_0^{(i^\star)}, x_{\mathrm{q}})$ be a Lipschitz recall-and-predict map as in Section 3, and assume that the eigenvalues of the underlying kernel satisfy $\lambda_j \asymp \exp(-cj^\alpha)$ for some $\alpha > 0$. Suppose that either (i) the number of mixture components satisfies $I \leq d_1 \lesssim (\log n)^{1/(\alpha+1)}$, or (ii) $\tilde{F}^\star$ does not depend on $x_{\mathrm{q}}$ and $I \leq d_1 = n^{o(1)}$. Then, for a suitable choice of architecture parameters (defining a depth-$2$ measure-theoretic Transformer class $\mathrm{TF}_n$), any empirical risk minimizer $\hat{F}_n$ over $\mathrm{TF}_n$ satisfies*

$$R(F^\star, \hat{F}_n) \lesssim \exp\big\{ -\Omega\big((\log n)^{\alpha/(\alpha+1)}\big)\big\}$$

*under Setting 1.*

**Statistically unifying associative recall and infinite-token regimes.** Prior work studied (i) *associative recall in Transformers* (Ramsauer et al., 2021) and (ii) *infinite-token / infinite-dimensional* inputs modeled as measures (Vuckovic et al., 2020). Theorem 1 integrate these threads by giving a *statistical* theory of measure-level associative recall: a measure-theoretic Transformer with learned softmax attention can *recall-and-predict at the level of measures*—sparsely isolating the query-relevant component of $\nu$ and basing the prediction on the recalled measure, in contrast to the universality or approximation results (Geshkovski et al., 2024; Furuya et al., 2025).

**Informal interpretation: effective dimension.** The spectral decay $\lambda_j \asymp \exp(-cj^\alpha)$ means that only the first few Mercer modes carry substantial signal. After the recall step, our Transformer effectively aggregates the first $D$ Mercer coefficients

$$b_j \approx \int e_j \, d\mu_0^{(i^\star)}, \quad j = 1, \ldots, D,$$

so an infinite-dimensional measure is compressed into the $D$-dimensional vector $b = (b_1, \ldots, b_D)$. Learning a Lipschitz function of $b$ from $n$ samples behaves like a $D$-dimensional problem (c.f., Schmidt-Hieber (2020)), with estimation error roughly

$$\text{Error of } D\text{-dim. problem} \approx n^{-\Theta(1/D)} \simeq \exp\big(-\Theta((\log n)/D)\big),$$

while truncating the Mercer expansion after $D$ modes incurs a bias of order $\exp(-cD^\alpha)$. Balancing these terms yields an effective dimension $D_{\mathrm{eff}}(n) \asymp (\log n)^{1/(\alpha+1)}$ and the sub-polynomial rate

$$R(F^\star, \hat{F}_n) \approx \exp\big(-\Theta((\log n)^{\alpha/(\alpha+1)})\big)$$

stated in Theorem 1. In this sense, the estimator behaves as if it were fitting only $D_{\mathrm{eff}}(n)$ degrees of freedom, despite each component being an infinite-dimensional measure. As a minimal sanity check, Appendix D presents a synthetic experiment whose convergence rate is consistent.

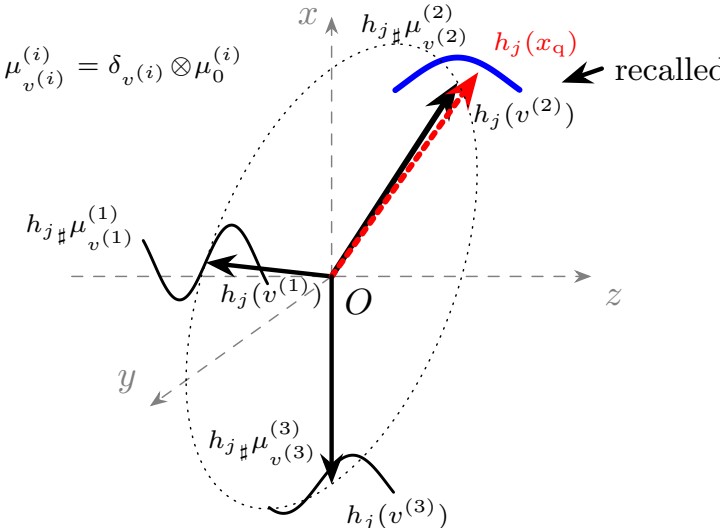

Figure 2: A geometric illustration of how query $x_{\mathrm{q}}$ and components $\mu_{v^{(i)}}^{(i)}$ are mapped by the (simplified) first layer $h_j((v,z)) = (v, e_j(z)) \in \mathbb{R}^{d_1+d_2}$, where $e_j$ is the $j$th Mercer eigenfunction. The product of the first $d_1$ indices of $h_j(x_{\mathrm{q}})$ and $h_j(y) \sim h_{j\sharp}\mu_{v^{(i)}}^{(i)}$ tells whether $y = (v, z)$ is sampled from $\mu_{v^{(i^*)}}^{(i^*)}$ or not.

**Mechanism: softmax attention as measure-valued associative memory.** Given a mixture context $\nu = I^{-1} \sum_{i=1}^{I} \mu_{v^{(i)}}^{(i)}$ and a query $x_{\mathrm{q}} = (v^{(i^*)}, 0)$, our depth-2 Transformer works as follows. An initial MLP embeds each token $y = (v^{(i)}, z)$ into a feature vector $h(y)$ that contains the first $D$ Mercer features $(e_j(z))_{j=1}^{D}$. Softmax attention then computes scores $\langle Qh(x_{\mathrm{q}}), Kh(y)\rangle$ and is parameterized so that these scores are large mainly when the document tag $v^{(i)}$ matches $v^{(i^*)}$ and small otherwise (Fig. 2).

Softmax attention then computes scores $\langle Qh(x_{\mathrm{q}}), Kh(y)\rangle$ so that, after normalization, the weights concentrate on samples from $\mu_{v^{(i^*)}}^{(i^*)}$. Writing $w_{x_{\mathrm{q}}}(y)$ for the resulting attention weight on token $y$, the value path computes, for each $j$,

$$\hat{b}_j \approx \int e_j(z)\, w_{x_{\mathrm{q}}}(y)\, \mathrm{d}\nu(y) \approx \int e_j(z)\, \mathrm{d}\mu_0^{(i^*)}(z), \qquad j = 1, \ldots, D,$$

yielding a $D$-dimensional descriptor $b = (\hat{b}_1, \ldots, \hat{b}_D)$ of the recalled component measure. A final MLP maps $(b, x_{\mathrm{q}})$ to the scalar prediction $\tilde{F}^\star(\mu_0^{(i^*)}, x_{\mathrm{q}})$. In this way, softmax attention *filters* the mixture down to the relevant component and *integrates* its Mercer features, so that predicting from an infinite-dimensional measure reduces to learning a function of $D$ summary statistics. This near one-hot, query-dependent filtering is beyond the limitations of frozen kernels (Zhou et al., 2024) and linear attentions (Kim et al., 2024).

### 4.2 MINIMAX OPTIMALITY (THE LOWER BOUND).

Next, we demonstrate the lower bound in Structured Setting (Setting 2). We show that the exponent of the obtained upper bound is essentially tight, albeit in Setting 2 whose technical assumptions are different from Setting 1 .

**Theorem 2** (Minimax Lower Bound). *In Setting 2, any $L^2(\mathbb{P}_{\nu, x_{\mathrm{q}}}(\mathcal{H}_0))$-estimator $\hat{F}$ satisfies*

$$\sup_{\tilde{F}^\star \in \mathcal{F}^\star} R(\hat{F}, F^\star) \gtrsim \exp\left(-O((\ln n)^{\frac{\alpha}{\alpha+1}})\right)$$

*where $F^\star(\nu, x_{\mathrm{q}}) := \tilde{F}^\star(\mu_0^{(i^*)}, x_{\mathrm{q}})$, $\alpha$ is the decay rate of the eigenvalues of the kernel of $\mathcal{H}_0$.*

**Optimality of Transformers.** Taken together with Theorem 1, our information-theoretic lower bound in Theorem 2 shows that the statistical rate of empirical risk minimization over transformers achieves the minimax rate $(\ln n)^{\frac{\alpha}{\alpha+1}}$ up to multiplicative constants in the exponent. Equivalently, no method can improve the $n$–dependence beyond the stated exponent (up to universal constants), so the learned-softmax Transformer attains the best-possible sample complexity for this problem class. The optimality continues to hold for a fixed or slowly growing number of mixture components $I$, confirming that the learned softmax attention provides the right inductive bias for measure-level recall. This minimax optimality of softmax Transformers is consistent with statistical results for simple infinite-dimensional regression (Takakura & Suzuki, 2023) and with finite-dimensional in-context learning scenarios that require retrieval (Nishikawa et al., 2025).

**Technical Contribution—Minimax Lower Bound.** We first reduce associative recall to infinite-dimensional Lipschitz regression by observing that estimating from the mixed input $\nu$ is no easier than from the pure measure $\mu_0^{(i^*)}$. A truncation of Mercer coefficients plus anisotropic rescaling—modifying prior rescaling arguments (Lanthaler, 2024) to more general geometry with exponential decay—makes the induced geometry essentially isotropic, letting us embed the classical $d$-dimensional Lipschitz class and import standard packing bounds. Combining these bounds with the classical result (Yang & Barron, 1999) yields a rate matching our upper bound.

## 5  CONCLUSION AND DISCUSSION

We introduced the concept of measure-theoretic associative memory (recall) and established that learned softmax attention can realize sharp recall even in infinite-dimensional, measure-valued settings—something beyond the reach of frozen kernels and difficult for linear attention. Our analysis further shows that the statistical efficiency of Transformers extends beyond finite-dimensional contexts, offering a principled explanation of their recall ability. While the present results focus on exponentially decaying spectra under smooth eigenfunctions, they open the door to broader regimes: extending the rates to polynomial decay and incorporating eigenfunction smoothness into the analysis represent natural next steps toward a more complete theory.

## LLM USAGE STATEMENT

Large language models are used for two purposes: to proofread and polish English writing, to help us find related works. We did not use any LLM assistant for designing the problem settings and constructing the proofs.

### 5.0.1  ACKNOWLEDGMENTS

RK and TS were partially supported by JSPS KAKENHI (24K02905) and JST CREST (JP-MJCR2115). This research is supported by the National Research Foundation, Singapore and the Ministry of Digital Development and Information under the AI Visiting Professorship Programme (award number AIVP-2024-004). Any opinions, findings and conclusions or recommendations expressed in this material are those of the author(s) and do not reflect the views of National Research Foundation, Singapore and the Ministry of Digital Development and Information. RK was supported by the FY 2024 Self-directed Research Activity Grant of the University of Tokyo's International Graduate Program "Innovation for Intelligent World" (IIW).

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

## A  PRELIMINARIES AND NOTATIONAL REMARKS

We begin by fixing notation and conventions used throughout the paper. This includes basic vector and matrix operations, measure-theoretic notation, and standard identifications between measures and their densities.

For integers $N_1 \leq N_2$ and a vector $\boldsymbol{v} \in \mathbb{R}^N$, we define $\boldsymbol{v}_{N_1:N_2} := (v_{N_1}, \ldots, v_{N_2})^\top$. For a matrix $A$, we define $\|A\|_0$ as the number of nonzero entries and $\|A\|_\infty := \max_{i,j} |A_{i,j}|$. We denote $\lambda$ as the Lebesgue measure on $\mathbb{R}^N$, for a integer $N$. $f_\sharp \mu(\cdot) := \mu(f^{-1}(\cdot))$ denotes a pushforward of a measure $\mu$ by a mapping $f$. For a measurable space $\mathcal{X}$, we write $\mathcal{P}(\mathcal{X}) := \{ \mu \mid \mu \text{ is a probability measure on } \mathcal{X} \}$, $\mathcal{M}_+(\mathcal{X}) := \{ \mu \mid \mu \text{ is a nonnegative measure on } \mathcal{X} \}$.

Unless otherwise specified, we write $(\Omega, \mathcal{F}, \mathbb{P})$ for an underlying probability space. For a measurable function $f : \Omega \to \mathbb{R}$ and $1 \leq p < \infty$, the $L^p(\mathbb{P})$ norm is defined as $\|f\|_{L^p(\mathbb{P})} := \left( \int_\Omega |f|^p \, d\mathbb{P} \right)^{1/p}$, while the $L^\infty(\mathbb{P})$ norm is given by $\|f\|_{L^\infty(\mathbb{P})} := \operatorname{ess\,sup}_{\omega \in \Omega} |f(\omega)|$. When the underlying measure is clear from context, we simply write $\|f\|_{L^p}$ and $\|f\|_\infty$.

For a random variable $X : \Omega \to \mathcal{X}$, we denote its distribution (the pushforward of $\mathbb{P}$ under $X$) by $\mathbb{P}_X(\cdot) := \mathbb{P}(X \in \cdot)$. Expectations with respect to $\mathbb{P}$ are denoted by $\mathbb{E}[\,\cdot\,]$, and if $X$ is a random variable with law $\mathbb{P}_X$, we also write $\mathbb{E}_X[\,f(X)\,]$ for $\int f(x)\, d\mathbb{P}_X(x)$.

**Remark 2** (Identification of measures and densities). Let $\lambda$ be a reference measure on $X$ (e.g., the Lebesgue measure). Given $f \in \mathcal{H}$ and a constant $c \in \mathbb{R}$, we define a probability measure $\mu$ by

$$\frac{d\mu}{d\lambda}(x) := f(x) + c,$$

where $c$ is chosen so that $f + c \geq 0$ $\lambda$-a.e. and $\mu(X) = \int_X \big(f(x) + c\big)\, dx = 1$. In this case, we write $\mu \in \mathcal{H}$ by identifying $\mu$ with its density $f + c$. When sampling $\mu_0 \in B(\mathcal{H}_0, \|\cdot\|_{\mathcal{H}_0^{\gamma_b}})$, please note that we choose $c$ with no randomness and let $c = 0$ for simplicity, throughout this paper.

**Definition 7** (Mercer expansion). Let $\mathcal{H}_0$ be a reproducing kernel Hilbert space (RKHS) on a domain $\mathcal{X}$ with reproducing kernel $K : \mathcal{X} \times \mathcal{X} \to \mathbb{R}$. By Mercer's theorem, $K$ admits the decomposition

$$K(x, x') = \sum_{j=1}^{\infty} \lambda_j e_j(x) e_j(x'),$$

where $(\lambda_j)_{j \geq 1}$ are the (non-negative, non-increasing) Mercer eigenvalues and $(e_j)_{j \geq 1}$ are the corresponding $L^2$-orthonormal eigenfunctions. For any $\mu \in \mathcal{H}_0$, its representation in the eigenbasis is

$$p_\mu = \sum_{j=1}^{\infty} b_j e_j, \quad \text{with coefficients } b_j = \langle p_\mu, e_j \rangle_{L^2}.$$

**Definition 8** (Generalized RKHS norm). Let $\mathcal{H}_0$ be an RKHS with orthonormal basis $\{e_j\}_{j \geq 1}$ in $L^2$ and associated eigenvalues $\{\lambda_j\}_{j \geq 1}$. For a measure $\mu$ whose associated function in $\mathcal{H}_0$ admits the expansion

$$p_\mu = \sum_{j \geq 1} b_j e_j, \quad \text{where} \quad \frac{d\mu}{d\lambda}(x) = p_\mu(x) + \text{constant}.$$

we define, for $a \in \mathbb{R}$, the *generalized norm*

$$\|\mu\|_{\mathcal{H}_0^a}^2 := \|p_\mu\|_{\mathcal{H}_0^a}^2 := \sum_{j \geq 1} \lambda_j^{-a} b_j^2.$$

Special cases include the case $a = 0$: $\|\cdot\|_{\mathcal{H}_0^0}$ coincides with the $L^2$ norm; $a = 1$: $\|\cdot\|_{\mathcal{H}_0^1}$ is the standard RKHS norm; $a = -1$: $\|\cdot\|_{\mathcal{H}_0^{-1}}$ is the MMD norm.

**Definition 9** (Metric balls). Let $(X, d)$ be a metric space. For $x \in X$ and $\epsilon > 0$, the (closed) metric ball of radius $\epsilon$ centered at $x$ is

$$B(x, d, \epsilon) := \{\, y \in X \mid d(x, y) \leq \epsilon \,\}.$$

When $\epsilon = 1$, we simply write $B(x, d)$ and refer to it as the *unit ball*.

**Definition 10** (Lipschitz functions). Let $(X, d)$ be a metric space and $A \subset X$ be a fixed domain. A function $f : A \to \mathbb{R}$ is said to be *L-Lipschitz* on $A$ with respect to $d$ if

$$|f(x) - f(x')| \leq L\, d(x, x') \quad \forall x, x' \in A.$$

The set of all such functions is denoted by $\text{Lip}_L(A, d)$.

PRELIMINARIES ON METRIC ENTROPY

For the subsequent proofs, we will make repeated use of standard notions from metric entropy. In particular, coverings, packings, and their associated numbers provide a convenient way to quantify the complexity of hypothesis classes. We therefore collect the relevant definitions and basic lemmas here.

**Definition 11** ($\epsilon$-covering). Let $(X, d)$ be a metric space, $A \subset X$, and $\epsilon > 0$. A finite set $\{x_1, \ldots, x_N\} \subset X$ is called an $\epsilon$-*covering* of $A$ (with respect to $d$) if

$$A \subset \bigcup_{i=1}^{N} B(x_i, d, \epsilon),$$

where $B(x_i, d, \epsilon)$ denotes the closed metric ball of radius $\epsilon$ centered at $x_i$.

**Definition 12** ($\epsilon$-packing)**.** Let $(X, \mathrm{d})$ be a metric space, $A \subset X$, and $\epsilon > 0$. A finite set $\{x_1, \ldots, x_N\} \subset A$ is called an $\epsilon$-*packing* of $A$ (with respect to d) if

$$\mathrm{d}(x_i, x_j) > \epsilon \quad \text{for all } i \neq j.$$

Equivalently, the metric balls $B(x_i, \mathrm{d}, \epsilon/2)$ are pairwise disjoint.

**Definition 13** (Covering number)**.** The *covering number* of $A$ at scale $\epsilon$ with respect to d is

$$\mathcal{N}(A, \epsilon)_{\mathrm{d}} := \min \{ N \mid \exists \epsilon\text{-covering of } A \text{ of size } N \}.$$

**Definition 14** (Packing number)**.** The *packing number* of $A$ at scale $\epsilon$ with respect to d is

$$\mathcal{M}(A, \epsilon)_{\mathrm{d}} := \max \{ M \mid \exists \epsilon\text{-packing of } A \text{ of size } M \}.$$

**Lemma 1** (Covering–packing equivalence)**.** *For any metric space $(X, \mathrm{d})$, any $A \subset X$, and $\epsilon > 0$, one has*

$$\mathcal{M}(A, 2\epsilon)_{\mathrm{d}} \leq \mathcal{N}(A, \epsilon)_{\mathrm{d}} \leq \mathcal{M}(A, \epsilon)_{\mathrm{d}}.$$

*In particular, the covering number and the packing number are equivalent up to constant factors in the scale parameter.*

**Lemma 2** (Monotonicity under metric domination)**.** *Let $\mathrm{d}$ and $\mathrm{d}'$ be two metrics on $X$, and let $c > 0$ such that $\mathrm{d}(x, y) \leq c\,\mathrm{d}'(x, y)$ for all $x, y \in X$. Then, for any $A \subset X$ and $\epsilon > 0$,*

$$\mathcal{N}(A, \epsilon)_{\mathrm{d}} \leq \mathcal{N}(A, \epsilon)_{c\,\mathrm{d}'} = \mathcal{N}(A, c^{-1}\epsilon)_{\mathrm{d}'}.$$

*In words: if $\mathrm{d}$ is dominated by $c\mathrm{d}'$, then $\epsilon$-coverings with respect to $c\mathrm{d}'$ are also $\epsilon$-coverings with respect to $\mathrm{d}$, hence covering under $\mathrm{d}$ is no harder.*

## A.1 TECHNICAL VERSION OF SECTION 3.1

Now we will introduce two technical settings for the data generation. We begin with the assumptions required for establishing the estimation upper bound, and then turn to alternative structural assumptions that are used for deriving the lower bound.

### A.1.1 PROBABILITY SETTING FOR UPPER BOUND.

**Assumption 6** (Common Assumptions: The RKHS Structure, the Regularity, and the Lipshictzness: Technical Version of Assumption 1,2,3)**.** Fix an integer $i^* \in \{1, \ldots, I\}$. A query vector $x_{\mathrm{q}}$, an input measure $\nu = \sum_i \delta_{v^{(i)}} \otimes \mu_0^{(i)}$, and an output $y = F^\star(\nu, x_{\mathrm{q}}) + \xi$ are generated as follows:

- *RKHS setting:* We assume the density of $\mu_0 \sim \mathbb{P}_{\mu_0}$ is in a space $\mathcal{H}_0$ ignoring a constant. $\mathcal{H}_0$ is an RKHS on a bounded domain $\mathcal{X}_0 \subset [-M, M]^{d_2}$ with Mercer decomposition (e.g., (Schölkopf & Smola, 2002))

$$K(x, x') = \sum_{j=1}^{\infty} \lambda_j e_j(x) e_j(x'), \quad \lambda_j \simeq \exp(-cj^\alpha),$$

  where $c, \alpha > 0$ and eigenfunctions $(e_j)_{j \geq 1}$ are $L^2$-orthonormal. $\mu_0$ is nonnegative and $\frac{\mathrm{d}\mu_0}{\mathrm{d}\lambda} - c'$ lies in the metric ball $B(\mathcal{H}_0, \|\cdot\|_{\mathcal{H}_0^{\gamma_{\mathrm{b}}}})$ with $\gamma_{\mathrm{b}} > 0$, where $\lambda$ is the Lebesgue measure and $c'$ is a global constant without any randomness. Let $c' = 0$ for simplicity, throughout this paper. The density is infinite-dimensional, but the informative content is sharply concentrated in low-order components, with high-frequency contributions decaying exponentially. For $a \in \mathbb{R}$, the $\mathcal{H}_0^a$-norm on $\mathcal{H}_0$ is defined as $\|\mu_0\|_{\mathcal{H}_0^a}^2 := \|\frac{\mathrm{d}\mu_0}{\mathrm{d}\lambda}\|_{\mathcal{H}_0^a}^2 := \sum_{j \geq 1} \lambda_j^{-a} b_j^2$ where $\frac{\mathrm{d}\mu_0}{\mathrm{d}\lambda} = \sum_j b_j e_j$. Similar RKHS structures can be found in Suzuki (2020); Nishikawa et al. (2022); Zhou et al. (2024); Liu & Zhou (2025).

- *Smoothness of eigenfunctions in $\mathcal{H}_0$:* (a) $(e_j)_{j \geq 1}$ are uniformly bounded, and analytic on $[-M, M]^{d_2}$; (b) $\mathcal{X}_0 \subset [-M + \delta, M - \delta]^{d_2}$ for some $\delta > 0$; (c) Each $e_j$ admits an absolutely convergent power series $e_j(x) = \sum_{k \in \mathbb{N}^{d_2}} a_k x^k$ on $[-M, M]^{d_2}$. The three conditions are required to focus on the sample complexity regarding the decay rate $\alpha$.

- *Distinguishability of sampled context vectors:* The vectors $(\mathbb{S}^{d_1-1})^{\otimes I} \ni (v^{(i)})_{i=1}^I \sim \mathbb{P}_v$ satisfy $\langle v^{(i)}, v^{(i')} \rangle \leq 0$, $\quad 1 \leq i < i' \leq I$. To satisfy this, we also require $I \leq d_1$. This ensures that contexts are sufficiently distinguishable for recall.

- *Lipschitzness of the target functional:* Remember that the output is generated as $y := \tilde{F}^\star(\mu_0^{(i^*)}, x_q) + \xi$, $\quad \xi \sim \mathcal{N}(0, \sigma^2)$. The hidden functional $\tilde{F}^\star : B(\mathcal{H}_0, \|\cdot\|_{\mathcal{H}_0^{\gamma_b}}) \times \mathcal{X}_q \to \mathbb{R}$ is assumed to be Lipschitz: Let the metric on the product set $B(\mathcal{H}_0, \|\cdot\|_{\mathcal{H}_0^{\gamma_b}}) \times \mathcal{X}_q$ be

$$\mathrm{d}_{\mathrm{prod}}\big((\mu_0, x), (\mu_0', x')\big) := \|\mu_0 - \mu_0'\|_{\mathcal{H}_0^{\gamma_f}} + \|x - x'\|_2, \quad \gamma_f < 0,$$

  where $\mu_0, \mu_0' \in B(\mathcal{H}_0, \|\cdot\|_{\mathcal{H}_0^{\gamma_b}})$ and $x, x' \in \mathcal{X}_q$. We write $\mu_0 \in \mathcal{H}_0$ by identifying $\mu_0$ with its density in $\mathcal{H}_0$ up to an additive constant (see Remark 2). We assume $\tilde{F}^\star$ is in $\mathrm{Lip}_L(B(\mathcal{H}_0, \|\cdot\|_{\mathcal{H}_0^{\gamma_b}}) \times \mathcal{X}_q, \mathrm{d}_{\mathrm{prod}})$, a set of $L$-Lipschitz functionals with respect to $\mathrm{d}_{\mathrm{prod}}$.

These assumptions specify the analytic and structural conditions of the RKHS and the distinguishability of contexts, which will be imposed throughout the analysis. We provide an example for our assumptions:

**Example 2** (Rapid eigenvalue decay in Assumption 6). *It is known (Grigor'yan, 2006) that on a compact Riemannian manifold $\mathcal{M}$ the Laplace-Beltrami operator has a discrete spectrum satisfying Weyl's law: the heat kernel expansion is $p_t(x, y) = \sum_{k=0}^\infty e^{-\Theta(k^{2/n})t} \varphi_k(x) \varphi_k(y)$, where $\{\varphi_k\}_k$ are eigenfunctions, $n = \dim(\mathcal{M})$, and $t > 0$. Settings with rapid eigenvalue decay have been investigated as a data structure (Nadler et al., 2005; Coifman & Lafon, 2006; Xia & Shi, 2024). Gaussian kernels, despite being non-compactly supported, are also widely used in ML tasks (e.g. Schölkopf & Smola (2002)).*

Next, we restrict the input measures to be probability measures:

**Assumption 7** (Probability assumption for Setting 3: Technical Version of Assumption 4). Let $\mathcal{P}(X)$ denote the set of Borel probability measures on a measurable set $X$. For $\mu_0 \sim \mathbb{P}_{\mu_0}$, $\mu_0 \in \mathcal{P}(\mathcal{X}_0)$ almost surely (with probability 1).

Based on these assumptions, we can summarize the probabilistic setting for our upper bound analysis:

**Setting 3** (Probability Setting for Upper Bound: Technical Version of Setting 1). Assumption 6 and Assumption 7 are satisfied. In short: $\mu_0 \sim \mathbb{P}_{\mu_0}$ is constrained as a probability measure whose density is in RKHS ball.

### A.1.2  STRUCTURED SETTING FOR LOWER BOUND.

For the minimax lower bound, we relax the probability constraint in Assumption 7 and instead impose structural conditions (following Lanthaler (2024)) on the coefficients of the Mercer expansion:

**Assumption 8** (Structural assumptions for Setting 4: Technical Version of Assumption 5). Let $(\Omega, \mathbb{P})$ be a probability space. Instead of constraining $\mu_0 \sim \mathbb{P}_{\mu_0}$ to be a probability measure, $\mu_0$ is generated in Definition 1 as follows:

- *Random RKHS element:* The "density" $p_{\mu_0} : \Omega \to \mathcal{H}_0$ associated with $\mu_0$ has the expansion

$$\frac{\mathrm{d}\mu_0}{\mathrm{d}\lambda}(\omega)(\cdot) := p_{\mu_0}(\omega)(\cdot) = \sum_{j=1}^\infty \lambda_j^{\gamma_d} Z_j(\omega) e_j(\cdot), \quad \omega \in \Omega,$$

  where $(e_j)_{j \geq 1}$ is an $L^2$-orthonormal basis on $\mathcal{X}_0$, $\gamma_d > 0$, and $\lambda_1^{\gamma_d} \geq \lambda_2^{\gamma_d} \geq \cdots \geq 0$ are summable.

- *Random coefficients:* The variables $Z_j : \Omega \to \mathbb{R}$ are jointly independent, satisfy $\mathbb{E}[|Z_j|^2] = 1$, $Z_j \sim \rho_j(z)\,dz$, and obey the uniform bounds $\sup_j \|\rho_j\|_\infty \leq R$, $\quad \lambda_1^{\gamma_d/2} \leq R$ for some $R > 0$.

**Example 3.** *Assumption 6 and Assumption 8 are compatible: Take $\rho_j = \frac{1}{2}\mathbb{1}_{\{-1,1\}}$ and $\lambda_j \leq A\exp(-cj^\alpha)$. Then, $\|\mu_0\|_{\mathcal{H}_0^{\gamma_b}} \leq \sum_j \lambda_j^{\gamma_d - \gamma_b}$ always converges if $\gamma_d > \gamma_b$ and $\|\mu_0\|_{\mathcal{H}_0^{\gamma_b}} = O(A^{\gamma_d - \gamma_b})$. Therefore, there exists some $A > 0$ such that $\|\mu_0\|_{\mathcal{H}_0^{\gamma_b}} \leq 1$ a.s.*

Finally, we summarize the corresponding setting:

**Setting 4** (Structured Setting for Lower Bound: Technical Version of Setting 2). Assumption 6 and Assumption 8 are satisfied. In short: the density is sampled in the form of Mercer expansion $\mathrm{d}\mu_0/\mathrm{d}\lambda \coloneqq \sum_j \lambda_j^{\gamma_d} Z_j e_j$ where $Z_j$ are independent r.v.s. and $\mu_0$ may not be the probability measure.

# B    ESTIMATION ERROR ANALYSIS (UPPER BOUND)

The overarching goal of this section is to derive a statistical upper bound for transformer-based estimators in Setting 3. Specifically, we establish that the empirical risk minimizer achieves a convergence rate of the form

$$R(\hat{F}, F^\star) \lesssim \exp\big(-\Omega((\log n)^{\alpha/(\alpha+1)})\big).$$

This rate can be regarded as the infinite-dimensional analogue of the classical $n^{-\Theta(1/d)}$ risk bound for $d$-dimensional regression, where the effective dimension scales as $d \sim (\log n)^{1/(\alpha+1)}$. In particular, although the associative recall task requires handling measure-valued components, our analysis demonstrates that its statistical complexity coincides with that of a pure infinite-dimensional regression problem, whose minimax lower bound will be shown in Appendix C. The subsequent subsections establish this result step by step, through successive approximation bounds for the individual network layers.

## B.1    PROOF SKETCH FOR THE ESTIMATION UPPER BOUND

In this part, we will explain how to prove the following theorem:

**Theorem 3** (Sub-Polynomial Convergence Corresponding to Theorem 1, A Simplified Version of Theorems 5 and 6). *Let* $\tilde{F}^\star \in \mathrm{Lip}_1(B(\mathcal{H}_0, \|\cdot\|_{\mathcal{H}_0^{\gamma_b}}) \times \mathcal{X}_q, \mathrm{d}_{\mathrm{prod}})$ *and assume one of the following cases:*

*(i) the number of mixture components is bounded as* $I \le d_1 \lesssim (\ln n)^{\frac{1}{\alpha+1}}$;

*(ii) the hidden target function* $\tilde{F}^\star$ *is independent of* $x_q$ *and* $I \le d_1 \simeq n^{o(1)}$.

*Let* $\hat{F}$ *be the empirical risk minimizer with the transformer class* $\mathrm{TF}(\epsilon)$ *as the set of mappings* $\mathrm{Attn}_{\theta_2} \diamond \mathrm{MLP}_{\xi_2} \diamond \mathrm{Attn}_{\theta_1} \diamond \mathrm{MLP}_{\xi_1}$ *such that,* $H_1, B_{a,1}, \ell_2 = (\log \epsilon^{-1})^{O(1)}$, $\|\boldsymbol{p}_2\|_\infty, s_2 \lesssim \exp\big(O((\log \epsilon^{-1})^{1+\alpha^{-1}})\big)$, $d_{\mathrm{attn2}}, S'_{a,2}, H_2, B'_{a,2} = 1$, $S_{a,2}, B_{a,2} = 0$ *in both cases, and*

*in (i):* $\ell_1, \|\boldsymbol{p}_1\|_\infty, s_1, d_{\mathrm{attn1}}, S_{a,1} = (\log \epsilon^{-1})^{O(1)}$, *; in (ii):* $\ell_1, \|\boldsymbol{p}_1\|_\infty, s_1, d_{\mathrm{attn1}}, S_{a,1} \lesssim \exp\big(O((\log \epsilon^{-1})^{1+\alpha^{-1}})\big)$. *Then in Setting 3,*

$$R(F^\star, \hat{F}) \lesssim \exp\big(-\Omega((\ln n)^{\frac{\alpha}{\alpha+1}})\big),$$

*where* $\alpha$ *is the decay rate of the eigenvalues of the underlying kernel of* $\mathcal{H}_0$.

We will provide a proof sketch for the first case: (i) the number of mixture components is bounded as $I \le d_1 \lesssim (\ln n)^{\frac{1}{\alpha+1}}$.

To derive a statistical rate we must calibrate the size of the measure-theoretic transformer hypothesis class. Concretely, we choose an architecture that grants an $\epsilon$-approximation of $F^\star$ while keeping the covering entropy $V(\delta) \coloneqq \log \mathcal{N}(\mathrm{TF}(\epsilon); \delta)_\infty$ minimal; the risk bound then follows by balancing the approximation error $\epsilon$ with the estimation term governed by $V(\cdot)$. In short, the general theory in Schmidt-Hieber (2020) asserts that the excess risk can be bounded by the sum of approximation terms and a complexity term: for any $\delta > 0$, the $L^2$-risk $R$ is bounded by

$$R(F^\star, \hat{F}) \lesssim \inf_{\hat{F}} \|\hat{F} - F^\star\|_{L^2(\mathbb{P}_{\nu, x_q})}^2 + \delta + \frac{V(\delta)}{n}$$

Here the first term quantifies how well the architecture approximates $F^\star$, while the second reflects the statistical price of searching over a class of size $V(\delta)$. Our proof thus first controls $V(\cdot)$ layerwise and then selects $\epsilon$ to realize the optimal trade-off.

Our estimation bound relies on two main ingredients: (i) an approximation strategy for representing the target functional $F^\star(\nu, x_q) = \tilde{F}^\star(\mu_0^{(i^\star)}, x_q)$ via a depth-$L = 2$ transformer, and (ii) covering entropy bounds for each component of the architecture, combined through a composition lemma for measure-theoretic mappings.

**Step 1: Composition lemma (Appendix B.2).** We first state a generic result for the covering number of compositions of measure-theoretic maps.

**Lemma 3** (Composition lemma). *Let $\mathcal{G}_i$, $i = 1, 2$ be sets of maps $\Gamma_i : \mathcal{P}(\mathcal{X}_i^{(1)}) \times \mathcal{X}_i^{(1)} \to \mathcal{X}_i^{(2)}$ such that $\mathcal{X}_1^{(2)} \subset \mathcal{X}_2^{(1)}$, $\mathcal{N}(\mathcal{G}_i; \epsilon)_\infty \lesssim N_i$, and any $\Gamma_2 \in \mathcal{G}_2$ is $(L_{2,1}, L_{2,2})$-Lipschitz with respect to the 1-Wasserstein and Euclidean metrics. Then,*
$$\mathcal{N}(\{\Gamma_2 \diamond \Gamma_1; \Gamma_i \in \mathcal{G}_i, i = 1, 2\}; \epsilon)_\infty \lesssim \mathcal{N}(\mathcal{G}_2; \tfrac{\epsilon}{2})_\infty \cdot \mathcal{N}(\mathcal{G}_1; \tfrac{\epsilon}{2(L_{2,1}+L_{2,2})})_\infty.$$

The proof follows the standard finite-dimensional composition argument: approximate each map $\Gamma_i$ by the nearest covering element, and bound the difference of the composed maps using the Lipschitz constants.

**Step 2: Approximation by a depth-$2$ transformer.** We approximate $F^\star(\nu, x) = \tilde{F}^\star(\mu_0^{(i^*)}, x_q)$ using the following architecture, focusing on the first $D \gtrsim (\log \epsilon^{-1})^{\alpha^{-1}}$ Mercer coefficients:

1. **First MLP layer (Appendix B.3).** Construct $\mathrm{MLP}_{\xi_1}$ to augment the input $(x_1, x_2)$ with evaluations $(e_i(x_2))_{i=1}^D$ of an analytic basis $\{e_i\}$ up to $O(\epsilon_1)$ error. Analyticity implies $\log \mathcal{N}(\mathcal{F}_1; \epsilon_1)_\infty \lesssim \mathrm{poly} \log \epsilon_1^{-1} \cdot \mathrm{poly} D$.

2. **First attention layer (Appendix B.4** Apply $\mathrm{Attn}_{\theta_1}$ to $\mathrm{MLP}_{\xi_1}(\mu)$ to compute empirical means $\int e_j(y_2) \, d\mu_0(y_2)$ up to $O(\epsilon_1 + \epsilon_2)$ error. We approximate a one-hot selection of measures and compute as, informally,
$$\int \underbrace{\mathrm{Softmax}(\mathrm{MLP}_1(x_q)^\top Q^\top K \, \mathrm{MLP}_1(y))}_{\mathbb{1}[y \sim \mu_{v(i^*)}^{(i^*)}]} \mathrm{MLP}_1(y) \underbrace{d\nu(y)}_{\propto d \sum_i \mu_{v(i)}^{(i)}} \simeq \int \mathrm{MLP}_1(y) d\mu_{v(i^*)}^{(i^*)}(y).$$
By the construction of the first layer $\mathrm{MLP}_1$ and the product decomposition of $\mu_{v(i^*)}^{(i^*)}$, the RHS approximates $\int e_j(y_2) \, d\mu_0(y_2)$. Under sparsity constraints on the attention matrices,
$$\log \mathcal{N}(\mathcal{A}_1(d_{\mathrm{attn}}, H, B_a, S_a); \epsilon_2)_\infty \lesssim \mathrm{poly} D \cdot \mathrm{poly} \log(I \epsilon_2^{-1}).$$

3. **Second MLP layer (Appendix B.5).** Approximate a Lipschitz map on $\mathbb{R}^{D+d_1+d_2}$ whose inputs are retained Mercer coefficients $(\int e_j(y_2) \, d\mu_0(y_2))_{j=1}^D$ and query $x_q$. We also show $D \gtrsim (\log \epsilon^{-1})^{\alpha^{-1}}$ (this is $\epsilon$, not $\epsilon_1$) is sufficient to extract the features. This layer may have a large Lipschitz constant $L_{\mathrm{MLP},2} \lesssim e^{-O(c_\epsilon (\log \epsilon^{-1})^{\frac{2+2\alpha}{\alpha}})}$ for $c_\epsilon \lesssim \mathrm{poly} \log \log \epsilon^{-1}$, which constrains $\epsilon_1 \vee \epsilon_2$ to be super-polynomially small via the measure-theoretic composition discussed in Lemma 5.

4. **Second attention layer (Appendix B.6).** Implemented as a (one-dimensional) skip connection, with $O(1)$ Lipschitz constant, added to fit the formal hypothesis set definition.

**Step 3: Bounding the covering entropy (Appendix B.7).** Applying the composition lemma recursively over the four layers yields
$$\log \mathcal{N}(\mathrm{TF}; \epsilon)_\infty \lesssim \log \mathcal{N}(\mathcal{A}(d + D, 1, 0, 1, 0, d + D); \epsilon)_\infty + \log \mathcal{N}(\mathcal{F}(\ell_2, p_2, s_2); \tilde{\Omega}(\epsilon))_\infty$$
$$+ \log \mathcal{N}(\mathcal{A}(d_{\mathrm{attn}}, H, B_a, S_a); \tilde{\Omega}(L_{\mathrm{MLP},2}^{-1} \epsilon))_\infty$$
$$+ \log \mathcal{N}(\mathcal{F}(\ell_1, p_1, s_1); \tilde{\Omega}(L_{\mathrm{MLP},2}^{-1}(L_{\mathrm{Attn},1,W^1} + L_{\mathrm{Attn},1,\|\|_2})^{-1} \epsilon))_\infty.$$
This yields (also carefully bounding the term with respect to $d_1$ omitted above):

**Lemma 4.** *The covering entropy of the transformer class satisfies*
$$\log \mathcal{N}(\mathrm{TF}; \epsilon)_\infty \lesssim \exp\left(O((\log \epsilon^{-1})^{(1+\min(\alpha,\beta))/\min(\alpha,\beta)})\right)$$

where $\alpha$ is the decay rate of RKHS $\mathcal{H}_0$ and $d_1 \simeq (\ln \epsilon^{-1})^{\beta^{-1}}$.

**Step 4: From covering entropy to risk bound (Appendix B.7).** Applying the regression bound shown in Schmidt-Hieber (2020), with $V(\delta) = \log \mathcal{N}(\text{TF}; \delta)_\infty$, and choosing

$$\epsilon \simeq \exp\left(-\Theta(\log n)^{\frac{\min(\alpha,\beta)}{\min(\alpha,\beta)+1}}\right),$$

where $d_1 \simeq (\ln \epsilon^{-1})^{\beta^{-1}}$, we obtain the sub-polynomial convergence rate:

**Theorem 4** (Sub-polynomial convergence, a generalized version of Theorem 1). *In Probability Setting (Setting 3),*

$$\sup_{F^\star \in \mathcal{F}^\star} R(\hat{F}, F^\star) \lesssim \exp\left(-\Omega\big((\log n)^{\frac{\min(\alpha,\beta)}{\min(\alpha,\beta)+1}}\big)\right),$$

*where the eigenvalues are $\lambda_j \simeq \exp(-cj^\alpha)$ and the number of mixture components is $I \leq d_1 \lesssim (\ln n)^{\beta^{-1} \cdot \min(\alpha,\beta)/(\min(\alpha,\beta)+1)}$.*

## B.2   STEP 1: COMPOSITION LEMMA.

**Lemma 5** (Composition Lemma. Restated). *Let $\mathcal{G}_i$, $i = 1, 2$ be sets of $\Gamma_i$, which are maps from $\mathcal{P}(\mathcal{X}_i^{(1)}) \times \mathcal{X}_i^{(1)}$ to $\mathcal{X}_i^{(2)}$ such that $\mathcal{X}_1^{(2)} \subset \mathcal{X}_2^{(1)}$, $\mathcal{N}(\mathcal{G}_i; \epsilon)_\infty \lesssim N_i$, and any $\Gamma_2 \in \mathcal{G}_2$ is $(L_{2,1}, L_{2,2})$-Lipschitz with respect to 1-Wasserstein distance and Euclidean distance. Then, we have*

$$\mathcal{N}(\{\Gamma_2 \diamond \Gamma_1 \mid \Gamma_i \in \mathcal{G}_i\}; \epsilon)_\infty \lesssim \mathcal{N}(\mathcal{G}_2; \frac{\epsilon}{2})_\infty \cdot \mathcal{N}(\mathcal{G}_1; \frac{\epsilon}{2(L_{2,1}+L_{2,2})})_\infty.$$

*Proof.* First, remember that, for standard one-dimensional function classes $\mathcal{F}, \mathcal{G}$ with covering numbers $N_f, N_g$,

$$\mathcal{N}(\{f \circ g \mid f \in \mathcal{F}, g \in \mathcal{G}\}; \epsilon) \lesssim \mathcal{N}(\mathcal{F}; \epsilon) \cdot \mathcal{N}(\mathcal{G}; \epsilon/L_f)$$

where $L_f$ is the upperbound of Lipschitz constants of $\forall f \in \mathcal{F}$. For measure-theoretic mappings $\Gamma_1 \in \mathcal{G}_1$ and $\Gamma_2 \in \mathcal{G}_2$, take $\Gamma_1^i$ and $\Gamma_2^j$ be the ($\epsilon_1$ and $\epsilon_2$ nearest covering elements (i.e. $\sup_{\mu,x \in \mathcal{P}(\mathcal{X}_i^{(1)}) \times \mathcal{X}_i^{(1)}} |\Gamma_1(\mu, x) - \Gamma_1^i(\mu, x)| \leq \epsilon$). Then we bound the difference of compositions as

$$\begin{aligned}
&|(\Gamma_2 \diamond \Gamma_1)(\mu, x) - (\Gamma_2^j \diamond \Gamma_1^i)(\mu, x)| \\
&= |(\Gamma_2 \diamond \Gamma_1)(\mu, x) - (\Gamma_2 \diamond \Gamma_1^i)(\mu, x)| + |(\Gamma_2 \diamond \Gamma_1^i)(\mu, x) - (\Gamma_2^j \diamond \Gamma_1^i)(\mu, x)| \\
&\leq L_{2,1} W_1(\Gamma_1(\mu)_\sharp \mu, \Gamma_1^i(\mu)_\sharp \mu) + L_{2,2} \|\Gamma_1^i(\mu, x) - \Gamma_1(\mu, x)\|_2 + \epsilon_2 \\
&\leq (L_{2,1} + L_{2,2}) \epsilon_1 + \epsilon_2.
\end{aligned}$$

$\square$

## B.3   STEP 2-1: FIRST MLP LAYER

We begin by formalizing the approximation properties of the first MLP layer, which is responsible for embedding both the input tokens and auxiliary analytic features into a higher-dimensional representation. This layer plays a crucial role in ensuring that subsequent attention and MLP layers can operate on a sufficiently expressive feature space.

**Lemma 6** (E & Wang (2018)). *Let $f$ be an analytic function over $[-M, M]^{d_2}$ such that $\mathcal{X}_0 \subset [-M + \delta, M - \delta]^{d_2}$ for some $\delta > 0$ and the power series $f(x) = \sum_{k \in \mathbb{N}^{d_2}} a_k x^k$ is absolutely convergent over $[-M, M]^{d_2}$. Then, a deep ReLU network $\hat{f}$ with depth $O((\log \epsilon^{-1})^{2d_2})$ and width $d_2 + 4$ (independent of $\epsilon$) satisfies*

$$\sup_{x \in \mathcal{X}_0} |f(x) - \hat{f}(x)| \lesssim \epsilon.$$

Here, the notation $x^k$ denotes the multivariate monomial $\prod_{i=1}^{d_2} x_i^{k_i}$, and the absolute convergence condition ensures that the power series uniformly converges on $[-M, M]^{d_2}$, enabling uniform approximation on the interior domain $\mathcal{X}_0$.

As a corollary of Lemma 6, we obtain the following result:

**Corollary 1.** *Let $d = d_1 + d_2$. For a function $f$ such that*

$$f : [-M, M]^{d_1+d_2} \to \mathbb{R}^{d+D}, \quad f(x_1, x_2) = \begin{bmatrix} x_1 \\ x_2 \\ e_1(x_2) \\ \vdots \\ e_D(x_2) \end{bmatrix},$$

*where $e_j$ are defined in Assumption 6. That is, $f$ preserves the first $d_1 + d_2$ coordinates $(x_1, x_2)$ and augments them with $D$ analytic feature functions $e_j$ that depend only on $x_2$. There exists a network $\hat{f} \in \mathcal{F}(\ell_1, \boldsymbol{p}_1, s_1, \infty)$, where $\ell_1 \lesssim (\log \epsilon_1^{-1})^{2d_2}$, $p_{1,j} \lesssim D + d$, $s_1 \lesssim d + (\log \epsilon_1^{-1})^{2d_2} \cdot d_2^2 \cdot D$, such that*

$$\|f - \hat{f}\|_\infty \lesssim \epsilon_1.$$

*Proof.* For the first $d_1$ indices, we simulate $x_{1,i} = -\mathrm{ReLU}(-\boldsymbol{e}_i \boldsymbol{e}_i^\top \cdot x_1) + \mathrm{ReLU}(\boldsymbol{e}_i \boldsymbol{e}_i^\top \cdot x_1)$. This requires $O(1)$ depth, $O(d_1)$ width, and $O(d_1)$ parameters. The $i + d_1$-th $(i = 1, \ldots, d_2)$ indices require $O(1)$ depth, $O(d_2)$ width, and $O(d_2)$ parameters. We refer to Lemma 6 for the rest of indices. $\square$

In other words, each analytic component $e_j$ can be uniformly approximated by a ReLU network of logarithmic number of parameters, and the concatenated mapping $f$ can be represented by a block-structured network with parameter bounds as stated.

To evaluate the covering number, we use the following lemma:

**Lemma 7** (Schmidt-Hieber (2020)). *Let $V = \prod_{i=0}^{\ell}(p_i + 1)$. Then,*

$$\log \mathcal{N}(\mathcal{F}(\ell, p, s, F); \delta)_\infty \lesssim (s + 1) \log(\delta^{-1} \ell V).$$

The covering number is

$$\log \mathcal{N}(\mathcal{F}(\ell_1, p_1, s_1); \epsilon_1)_\infty \lesssim (d + (\log \epsilon_1^{-1})^{2d_2} \cdot D) \cdot (\log \epsilon_1^{-1} + \log \log \epsilon_1^{-1} + (\log \epsilon^{-1})^{2d_2} \log(d + D)))$$

$$\lesssim C_{D,\epsilon_1}(d + D(\log \epsilon_1^{-1})^{4d_2}),$$

where $C_{D,\epsilon_1} \lesssim \mathrm{poly} \log D + \mathrm{poly} \log d + \mathrm{poly} \log \log \epsilon_1^{-1}$.

### B.4 STEP 2-2: FIRST ATTENTION LAYER

In the preceding section, we have constructed an MLP layer capable of approximating the basis functions $e_j$ (Assumption 6) with high accuracy. The role of the first attention layer is now to process an input mixture measure $\nu_f$, identify the *associated measure* corresponding to a given query component $y_{f,i}$, and then output the concatenation of the raw query coordinates with the integrals of $e_j$ against that associated measure. Formally, this operation is realized by the mapping $\phi_2$ defined below.

We now analyze the approximation properties and complexity of the first attention layer in our architecture. Recall that the attention operator $\mathrm{Attn}_\theta$ has already been defined in the measure-theoretic form

$$\mathrm{Attn}_\theta : \mathcal{P}(\mathbb{R}^{d_{\mathrm{attn}}}) \times \mathbb{R}^{d_{\mathrm{attn}}} \to \mathbb{R}^{d_{\mathrm{attn}}},$$

where the first argument is a probability measure over token representations and the second argument is the query vector. The following lemmas show that, under appropriate structural assumptions on the input measures and functions:

1. the target mapping $\phi_2$ can be realized to accuracy $\epsilon_2$ by a member of the attention class $\mathcal{A}(d_{\mathrm{attn}}, H, B_a, S_a)$ (Lemma 8);

2. such attention mappings are Lipschitz continuous with an explicit bound in terms of the model parameters (Lemma 9);

3. the $\epsilon_2$-covering number of $\mathcal{A}$ admits an upper bound in the parameter regime above (Lemma 10).

We present these results in turn.

Before presenting Lemma 8, we clarify the role of the mapping $\phi_2$ in the composition-of-maps view (cf. Definition 5). In our construction, the first MLP layer $\phi_1$ approximates the Mercer features:

$$\phi_1(\mu, x) \simeq \begin{bmatrix} x_1 \\ \vdots \\ x_d \\ e_1(x_{d_1+1:d}) \\ \vdots \\ e_D(x_{d_1+1:d}) \end{bmatrix},$$

where $\{e_j\}_{j \geq 1}$ is the Mercer (RKHS) eigenbasis on $\mathcal{X}_0 \subset \mathbb{R}^{d_2}$. Accordingly, the push-forward measure after $\phi_1$ is

$$\mu_1 := \big(\phi_1(\mu, \cdot)\big)_\sharp \mu \in \mathcal{P}(\mathbb{R}^{d+D}),$$

and the composition rule yields

$$(\phi_2 \diamond \phi_1)(\mu, x) = \phi_2\big(\mu_1, \phi_1(\mu, x)\big).$$

In the present setting, $\phi_2$ preserves the first $d$ coordinates and replaces the last $D$ coordinates by the (component-wise) integrals of the associated measure against the Mercer basis:

$$\phi_2\big(\mu_1, \phi_1(\mu, x)\big) = \begin{bmatrix} x_1 \\ \vdots \\ x_d \\ \int e_1 \, \mathrm{d}\mu_0^{(i^*)} \\ \vdots \\ \int e_D \, \mathrm{d}\mu_0^{(i^*)} \end{bmatrix},$$

where $i^*$ denotes the index of the associated component selected by the query. Moreover, if $\mu_0^{(i^*)}$ admits a (Borel) density w.r.t. a reference measure $\lambda$, say $\frac{\mathrm{d}\mu_0^{(i^*)}}{\mathrm{d}\lambda} = f_{\mu_0^{(i^*)}}$ with $f_{\mu_0^{(i^*)}} \in L^2(\lambda)$ (cf. Remark 2), then, writing the Mercer expansion $f_{\mu_0^{(i^*)}} = \sum_{j \geq 1} b_j e_j$, we have, up to the immaterial constant term,

$$\int e_j \, \mathrm{d}\mu_0^{(i^*)} = \int e_j \, f_{\mu_0^{(i^*)}} \, \mathrm{d}\lambda = \langle e_j, f_{\mu_0^{(i^*)}} \rangle_{L^2(\lambda)} = b_j.$$

Hence $\phi_2$ produces, in its last $D$ coordinates, the (truncated) Mercer coefficients of the associated density.

Lemma 8 shows that, under our structural assumptions on the input measures and the transformation $f$, the target mapping $\phi_2$ can be uniformly approximated to accuracy $\epsilon_2$ by an attention mechanism with bounded parameters.

**Intuitions of Lemma 8.** The key point is in the structure of the first attention layer: For a fixed $j$, to extract the *associated* Mercer coefficient $b_j = \int e_j \, \mathrm{d}\mu_0^{(i^*)}$, we construct QK-matrix $W_{QK}^j$ such that, with tokens mapped by the (simplified) first MLP layer $(x_1, x_2) \mapsto \psi_j(x_1, x_2) = (x_1, e_j(x_2))$,

$$\psi_j(x_\mathrm{q})^\top W_{QK}^j \psi_j(y) = \begin{cases} \gg 1 & \text{if} \quad i = i^*; \\ \leq 0 & \text{if} \quad i \neq i^*, \end{cases} \quad \text{for} \quad y \sim \mu_{v^{(i)}}^{(i)} = \delta_{v^{(i)}} \otimes \mu_0^{(i)}.$$

Then, the softmax value will be

$$\mathrm{Softmax}(\psi_j(x_\mathrm{q})^\top W_{QK}^j \psi_j(y)) \simeq \mathbb{1}[i = i^*].$$

The construction is simple: take $W_{QK}^j \propto \sum_{k=1}^{d_1} e_k e_k^\top$ such that $\psi_j(x_\mathrm{q})^\top W_{QK}^j \psi_j(y) \propto \langle v^{(i^*)}, v^{(i)} \rangle$, and multiply a large scalar, where $e_k$ is a $k$-th one-hot vector, since $\psi$ preserves the

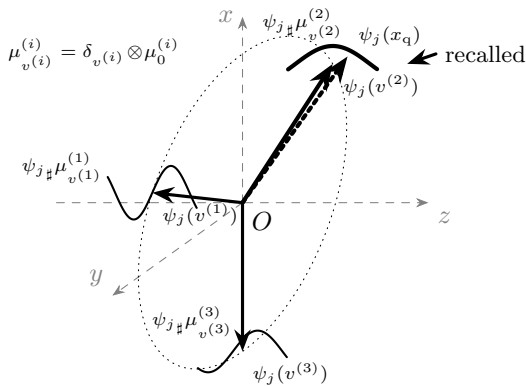

Figure 3: Geometric sketch of associative recall. Components $\mu_{v^{(i)}}^{(i)}$ are separated along a feature axis via the first MLP layer $f_1$; the query maps to $\psi_j(x_q)$ and aligns with anchor $\psi_j(v^{(i^*)}) := \psi_j((v^{(i^*)\top}, \mathbf{0}^\top)^\top)$, thereby recalling $\mu_0^{(i^*)}$. The pushforward $\psi_{j\sharp}\mu_{v^{(i^*)}}^{(i^*)}$ provides features used by $F^\star(\mu^{(i^*)}, x_q)$.

first $d$ coordinates: $\psi(x_q) = [v^{(i^*)\top}; *]^\top$, $\psi(y) = [v^{(i)\top}; *]^\top$, and $(v^{(i)})_i$ are distinguishable (i.e. $\langle v^{(i)}, v^{(j)}\rangle \leq 0$ for $i \neq j$), as described in Fig. 3. Then, with tokens mapped by the first MLP layer $f_1$, we have the $(d+j)$-th output of the $j$th head is given by

$$\mathbf{e}_{d+1}^\top \underbrace{W}_{\mathbf{e}_{d+1}\mathbf{e}_{d+1}^\top} \int \underbrace{\mathrm{Softmax}(\psi_j(x_q^\top)W_{QK}^j\psi_j(y))}_{\simeq \mathbb{1}[i=i^*]} \underbrace{V}_{\mathbf{e}_{d+1}\mathbf{e}_{d+1}^\top} \psi_j(y)\mathrm{d}\nu(y)$$

$$\simeq \int \mathbb{1}[i=i^*]e_j(y_{d_1+1:d})\mathrm{d}\left(\sum_i \delta_{v^{(i)}} \otimes \mu_0^{(i)}\right)(y)$$

$$\simeq \int e_j(y)\mathrm{d}\mu_0^{(i^*)}(y),$$

where we take the parameters as $W = V = \mathbf{e}_{d+1}\mathbf{e}_{d+1}^\top$. Finally, the second MLP layer maps $((\int e_j \, d\mu_0^{(i^*)})_{j=1}^D, v^{(i^*)})$ to $y$, and the second attention acts as a skip connection.

**Lemma 8.** *Let $\mathcal{P}_f$ denote the set of probability measures of the form*

$$\nu_f = \frac{1}{I}\sum_{i=1}^I f_\sharp\mu_i = f_\sharp\left(\frac{1}{I}\sum_{i=1}^I \mu_i\right),$$

*where the measures $\mu_i$ and the mapping $f$ satisfy the following conditions:*

- *$\mu_i$ is a probability measure supported on a bounded subset of $\mathbb{R}^d$, where $d = d_1 + d_2$, and admits the product form*
$$\mu_i = \delta_{v_i} \otimes \tilde{\mu}_i,$$
*with $v_i \in \mathbb{S}^{d_1} \subset [-B_x, B_x]^{d_1}$ satisfying $\langle v_i, v_j\rangle \leq 0$ for all $i \neq j$, and where $\tilde{\mu}_i$ is a probability measure supported on a bounded subset of $\mathbb{R}^{d_2}$.*

- *$f : \mathbb{R}^d \to \mathbb{R}^{d+D}$ is given by*
$$f(x) = \begin{bmatrix} x_1 \\ \vdots \\ x_d \\ \tilde{f}(x_{d_1+1}, \ldots, x_d) \end{bmatrix},$$
*where $\tilde{f} : \mathbb{R}^{d_2} \to \mathbb{R}^D$ is a bounded function.*

- $I \leq d_1$ and $\mathrm{supp}(f_\sharp \mu_i) \subset [-B_y, B_y]^{d+D}$ for all $i$.

*Moreover, for each $i$, define $x_i := [v_i^\top, 0^\top]^\top \in \mathbb{S}^{d_1} \times \{0_{d_2}\}$ and set $y_{f,i} := f(x_i)$.*

*Define the mapping $\phi_2 : \mathcal{P}_f \times \mathbb{R}^{d+D} \to \mathbb{R}^{d+D}$ by*

$$\phi_2(\nu_f, y_{f,i}) := \begin{bmatrix} y_{f,i,1} \\ \vdots \\ y_{f,i,d} \\ \int \mathrm{d}\left(\tilde{f}_1\right)_\sharp \tilde{\mu}_i \\ \vdots \\ \int \mathrm{d}\left(\tilde{f}_D\right)_\sharp \tilde{\mu}_i \end{bmatrix},$$

*where $\tilde{f}_j$ denotes the $j$-th coordinate function of $\tilde{f}$.*

*Then, there exists an attention operator $\hat{\mathrm{Attn}} \in \mathcal{A}(d_{\mathrm{attn}}, H, B_a, S_a)$ such that*

$$\sup_{\nu_f \in \mathcal{P}_f, \, y_{f,i} : v_i \in \mathbb{R}^d} \left\| \phi_2(\nu_f, y_{f,i}) - \hat{\mathrm{Attn}}(\nu_f, y_{f,i}) \right\|_\infty \leq \epsilon_2,$$

*where $d_{\mathrm{attn}} = d + D$, $H = D$, $B_a \lesssim \sqrt{\log\left(I \cdot \epsilon_2^{-1}\right)}$, and $S_a = d$.*

*Proof.* Fix an arbitrary $h \in \{1, \ldots, H = D\}$. We first specify the attention weight matrices as follows:

$$W^h = V^h = \boldsymbol{e}_{d+h} \boldsymbol{e}_{d+h}^\top,$$

where $d = d_1 + d_2$ and $\boldsymbol{e}_{d+h}$ denotes the $(d+h)$-th standard basis vector in $\mathbb{R}^{d+D}$. Similarly, define

$$Q^h = K^h = c \begin{bmatrix} I_{d_1} & O \\ O & O_{d_2+D} \end{bmatrix},$$

for a sufficiently large constant $c \gtrsim \sqrt{\log\left(I^3 \cdot \epsilon_2^{-1}\right)} \gtrsim \sqrt{\log\left(I \cdot \epsilon_2^{-1}\right)}$.

The corresponding attention weight for a query $x_i$ and a key $f(y)$ is given by

$$\mathrm{Softmax}\left(\langle Q^h x_i, \, K^h f(y)\rangle\right)$$
$$= \frac{\exp\left(c^2 \langle x_{i,1:d_1}, \, y_{1:d_1}\rangle\right)}{\int \exp\left(c^2 \langle x_{i,1:d_1}, \, y_{1:d_1}\rangle\right) \mathrm{d}\nu_f(y)}$$
$$= \frac{\exp\left(c^2 \langle x_{i,1:d_1}, \, y_{1:d_1}\rangle\right)}{I^{-1} \sum_{i'=1}^I \int \exp\left(c^2 \langle x_{i,1:d_1}, \, y_{1:d_1}\rangle\right) \mathrm{d}(\delta_{v_{i'}} \otimes \tilde{\mu}_{i'})(y)}$$
$$= \frac{\exp\left(c^2 \langle x_{i,1:d_1}, \, y_{1:d_1}\rangle\right)}{I^{-1} \sum_{i'=1}^I \exp\left(c^2 \langle v_{i'}, \, y_{1:d_1}\rangle\right)}$$
$$= \frac{\exp\left(c^2\right)}{\frac{1}{I}\left(\exp(c^2) + (I-1)\right)} \mathbb{1}_{\{y_{1:d_1} = v_i\}} \quad (v_i \perp v_j, \, i \neq j)$$
$$= I \cdot \mathbb{1}_{\{y_{1:d_1} = v_i\}} + O(I^{-1}\epsilon_2),$$

where the indicator function $\mathbb{1}_{\{y_{1:d_1} = v_i\}}$ arises because the keys $y_{1:d_1}$ take values in the finite set $\{v_1, \ldots, v_I\}$ with mutually nonpositive inner products. The last equality follows from the choice $c \gtrsim \sqrt{\log\left(I \cdot \epsilon_2^{-1}\right)}$, which ensures exponential separation of the correct key from the others.

Next, applying the value and output projection matrices, we obtain

$$W^h \int \mathrm{Softmax}\left(\langle Q^h x_i, \, K^h y\rangle\right) V^h y \, \mathrm{d}\nu_f(y)$$

$$= \frac{1}{I} \sum_{i'=1}^{I} W^h \int \text{Softmax}\big(\langle Q^h x_i,\, K^h f(y)\rangle\big) V^h f(y)\, \mathrm{d}\mu_{i'}(y)$$

$$= \frac{1}{I} \sum_{i'=1}^{I} \int \left(I \cdot \mathbb{1}_{\{y_{1:d_1}=v_i\}} + O(I^{-1}\epsilon_2)\right) \boldsymbol{e}_{d+h} \left(\tilde{f}_h(y_{d_1+1:d}) + O(\epsilon_1)\right) \mathrm{d}(\delta_{v_{i'}} \otimes \tilde{\mu}_{i'})(y)$$

$$\left(\text{where } \boldsymbol{e}_{d+h}^{\top}\tilde{f} = \tilde{f}_h\right)$$

$$= \frac{1}{I} \sum_{i'=1}^{I} \left(\int \left(I \cdot \mathbb{1}_{\{y_{1:d_1}=v_i\}} + O(I^{-1}\epsilon_2)\right)\mathrm{d}\delta_{v_{i'}}(y_{1:d_1})\right)\left(\int \tilde{f}_h(y_{d_1+1:d})\,\mathrm{d}\tilde{\mu}_{i'}(y_{d_1+1:d})\right)\boldsymbol{e}_{d+h}$$

$$= \left(\int \tilde{f}_h(y_{d_1+1:d})\,\mathrm{d}\tilde{\mu}_i(y_{d_1+1:d})\right)\boldsymbol{e}_{d+h} + O(\epsilon_2).$$

Finally, to incorporate the skip connection over the first $d$ coordinates, let

$$A = \begin{bmatrix} I_d & O \\ O & O \end{bmatrix}.$$

Applying $A$ to the input vector $y_{f,i}$ yields

$$A y_{f,i} = \begin{bmatrix} y_{f,i,1} \\ \vdots \\ y_{f,i,d} \\ 0_D \end{bmatrix}.$$

Combining the attention output for each head $h$ with this skip connection reproduces the target mapping $\phi_2$ up to an error of order $O(\epsilon_2)$ in the $\ell_\infty$ norm. This establishes the desired approximation property. $\qquad\square$

**Remark 3** (Why do we need a softmax attention?)**.** We informally demonstrate how linear attentions struggle with one-hot selection of densities without orthogonality. The main problem is that the context vectors $(v^{(i)})_i$ may have a negative correlation. For example, we consider $I = 2$ and

$$v^{(1)} = -v^{(2)}.$$

If we only have access to a linear attention, with the same QK matrices in the lemma,

$$\text{LinAttn}\big(\langle Q^h f(x_i),\, K^h f(y)\rangle\big) \simeq \langle v^{(i)}, v^{(i^*)}\rangle = \begin{cases} 1 & \text{if } i = i^*; \\ -1 & \text{if } i \neq i^*. \end{cases}$$

This implies that it is hard for linear attentions to extract *only* the $i^*$-th measure through integration $\int \text{LinAttn}(\langle Q^h x_i,\, K^h y\rangle) V^h y\mathrm{d}\nu_f(y)$. See Han et al. (2024); Fan et al. (2025) for empirical discussions; see also Kim et al. (2024), where strong assumptions such as relaxed sparsity and orthogonality of recall candidates were required to bypass this difficulty.

Thus, the first attention layer is expressive enough to implement the "association and extraction" operation: given a mixture, it can select the relevant component and compute the $e_j$-integrals needed for downstream processing. We next turn to the *stability* of such an operator with respect to perturbations in both the measure and the query vector.

The following lemma establishes a Lipschitz property of $\text{Attn}_\theta$ in both arguments. This lemma is inspired by Vuckovic et al. (2020). This quantitative stability will be essential for the subsequent covering number analysis.

The attention operator $\text{Attn}_\theta$ computes a weighted average of values using a softmax over inner products $\langle Q^h x,\, K^h y\rangle$. To bound its change when $(\mu, x)$ varies, we split the effect of $\mu$ and $x$.

For the measure part, Kantorovich–Rubinstein duality expresses the 1-Wasserstein distance $W_1$ (the standard Wasserstein metric) as the supremum of expectation differences over 1-Lipschitz functions, allowing us to control the change via the Lipschitz constant of the softmax kernel.

For the query part, we directly bound the kernel's Lipschitz dependence on $x$ and apply sparsity of the output projection. Combining both yields the stated Lipschitz bound.

**Lemma 9.** *Let* $\mathrm{Attn}_\theta \in \mathcal{A}(d_{\mathrm{attn}}, H, B_a, S_a)$ *be the attention operator as defined in Section 3. Assume that the query inputs satisfy* $\|x_1\|_\infty, \|x_2\|_\infty \leq B_x$, *and that for each* $i \in \{1,2\}$, *every* $y \in \mathrm{supp}(\mu_i)$ *satisfies* $\|y\|_\infty \leq B_y$. *Then* $\mathrm{Attn}_\theta$ *is Lipschitz in the joint variable* $(\mu, x)$ *in the sense that*

$$\|\mathrm{Attn}_\theta(\mu_1, x_1) - \mathrm{Attn}_\theta(\mu_2, x_2)\|_\infty$$
$$\lesssim H \exp\big(O(S_a^2 B_a^2 B_x B_y)\big) \cdot \big(W_1(\mu_1, \mu_2) + \|x_1 - x_2\|_2\big).$$

*Moreover, if* $d_{\mathrm{attn}} \lesssim D$, $H \lesssim D$, $B_a \lesssim \sqrt{\log\big(I\epsilon_2^{-1}\big)}$, *and* $S_a \lesssim d$, $B_x, B_y \lesssim 1$, *then the Lipschitz constant is bounded by* $D \exp\big(O(d^2 \log\big(I\epsilon_2^{-1}\big))\big)$.

*Proof.* **Bounding the difference in** $\mu$. We first bound the difference in the $\mu$-variable while keeping the query fixed. By (P-i) the matrix sparsity bound $\|Av\|_\infty \leq sb\|v\|_\infty$ when $A$ has at most $s$ nonzero entries per row and each entry bounded by $b$, we have

$$\|\mathrm{Attn}_\theta(\mu_1, x_1) - \mathrm{Attn}_\theta(\mu_2, x_1)\|_\infty$$
$$\leq \sum_h S_a^2 B_a^2 \left\| \int \frac{y \exp\left(\langle Q^h x, K^h y\rangle\right)}{\int \exp\left(\langle Q^h x_1, K^h z\rangle\right)\mathrm{d}\mu_1(z)}\mathrm{d}\mu_1 - \int \frac{y \exp\left(\langle Q^h x, K^h y\rangle\right)}{\int \exp\left(\langle Q^h x_1, K^h z\rangle\right)\mathrm{d}\mu_2(z)}\mathrm{d}\mu_2 \right\|_\infty$$

Next, Inserting intermediate terms to align denominators and numerators, we obtain,

$$\|\mathrm{Attn}_\theta(\mu_1, x_1) - \mathrm{Attn}_\theta(\mu_2, x_1)\|_\infty$$
$$\leq \sum_h S_a^2 B_a^2 \left\| \int \frac{y \exp\left(\langle Q^h x, K^h y\rangle\right)}{\int \exp\left(\langle Q^h x_1, K^h z\rangle\right)\mathrm{d}\mu_1(z)}\mathrm{d}\mu_1 - \int \frac{y \exp\left(\langle Q^h x, K^h y\rangle\right)}{\int \exp\left(\langle Q^h x_1, K^h z\rangle\right)\mathrm{d}\mu_2(z)}\mathrm{d}\mu_1 \right.$$
$$\left. + \int \frac{y \exp\left(\langle Q^h x_1, K^h y\rangle\right)}{\int \exp\left(\langle Q^h x_1, K^h z\rangle\right)\mathrm{d}\mu_2(z)}\mathrm{d}\mu_1 - \int \frac{y \exp\left(\langle Q^h x, K^h y\rangle\right)}{\int \exp\left(\langle Q^h x_1, K^h z\rangle\right)\mathrm{d}\mu_2(z)}\mathrm{d}\mu_2 \right\|_\infty$$
$$\leq \underbrace{\sum_h S_a^2 B_a^2 \left\| \int \frac{y \exp\left(\langle Q^h x_1, K^h y\rangle\right)}{\int \exp\left(\langle Q^h x_1, K^h z\rangle\right)\mathrm{d}\mu_1(z)}\mathrm{d}\mu_1 - \int \frac{y \exp\left(\langle Q^h x, K^h y\rangle\right)}{\int \exp\left(\langle Q^h x_1, K^h z\rangle\right)\mathrm{d}\mu_2(z)}\mathrm{d}\mu_1 \right\|_\infty}_{(i)}$$
$$+ \underbrace{\sum_h S_a^2 B_a^2 \left\| \int \frac{y \exp\left(\langle Q^h x_1, K^h y\rangle\right)}{\int \exp\left(\langle Q^h x_1, K^h z\rangle\right)\mathrm{d}\mu_2(z)}\mathrm{d}\mu_1 - \int \frac{y \exp\left(\langle Q^h x, K^h y\rangle\right)}{\int \exp\left(\langle Q^h x_1, K^h z\rangle\right)\mathrm{d}\mu_2(z)}\mathrm{d}\mu_2 \right\|_\infty}_{(ii)}.$$

**Bounding the term (i).** We have

$$(i) \leq \sum_h S_a^2 B_a^2 \left\| \int y \exp\left(\langle Q^h x_1, K^h y\rangle\right)\mathrm{d}\mu_1 \right\|_\infty$$
$$\times \left| \frac{1}{\int \exp\left(\langle Q^h x_1, K^h z\rangle\right)\mathrm{d}\mu_1(z)} - \frac{1}{\int \exp\left(\langle Q^h x_1, K^h z\rangle\right)\mathrm{d}\mu_2(z)} \right|$$
$$\leq \sum_h S_a^2 B_a^2 \left\| \int y \exp\left(\langle Q^h x_1, K^h y\rangle\right)\mathrm{d}\mu_1 \right\|_\infty$$
$$\times \left( \min\left\{ \int \exp\left(\langle Q^h x_1, K^h z\rangle\right)\mathrm{d}\mu_1(z), \int \exp\left(\langle Q^h x_1, K^h z\rangle\right)\mathrm{d}\mu_2(z) \right\} \right)^{-2}$$
$$\times \left| \int \exp\left(\langle Q^h x_1, K^h z\rangle\right)\mathrm{d}\mu_1(z) - \int \exp\left(\langle Q^h x_1, K^h z\rangle\right)\mathrm{d}\mu_2(z) \right| \quad \text{(by (P-iii))}$$
$$\lesssim \sum_h S_a^2 B_a^2 B_y \exp\big(3 S_a^2 B_a^2 B_x B_y\big) \left| \int \exp\left(\langle Q^h x_1, K^h z\rangle\right)\mathrm{d}(\mu_1 - \mu_2)(z) \right| \quad \text{(by (P-ii))},$$

where we used:

(P-i) the matrix sparsity bound $\|Av\|_\infty, \|Av\|_1 \le sb\|v\|_\infty$ when $A$ has at most $s$ nonzero entries per row and each entry bounded by $b$;

(P-ii) $|\langle Q^h x_1, K^h y \rangle| \le S_a B_a B_x \cdot S_a B_a B_y$. Indeed, since $\|Q^h\|_0, \|K^h\|_0 \le S_a$ (total number of nonzero entries) and $|Q_{jk}^h|, |K_{jk}^h| \le B_a$, while $\|x_1\|_\infty \le B_x$ and $\|y\|_\infty \le B_y$, we have

$$
\begin{aligned}
&|\langle Q^h x_1, K^h y \rangle| \\
\le& \|Q^h x_1\|_\infty \|K^h y\|_1 \\
\le& \Big( \max_j \sum_k |Q_{jk}^h| |x_{1,k}| \Big) \sum_{j,k} |K_{jk}^h| |y_k| \\
\le& (S_a B_a B_x)(S_a B_a B_y).
\end{aligned}
$$

Here the bound $\|Q^h x_1\|_\infty \le S_a B_a B_x$ follows because each coordinate of $Q^h x_1$ is a sum of at most $S_a$ terms, each of magnitude at most $B_a B_x$; similarly, $\|K^h y\|_1 \le \sum_{j,k} |K_{jk}^h| |y_k| \le S_a B_a B_y$ since there are at most $S_a$ nonzero matrix entries in total;

(P-iii) the bound

$$
\left| \frac{1}{\alpha_1} - \frac{1}{\alpha_2} \right| \le A^{-2} |\alpha_1 - \alpha_2|, \quad A < \min(\alpha_1, \alpha_2).
$$

The RHS is bounded as

$$
\begin{aligned}
((i) \lesssim) & \sum_h S_a^2 B_a^2 B_y \exp\big(3 S_a^2 B_a^2 B_x B_y\big) \left| \int \exp\big(\langle Q^h x_1, K^h z \rangle\big) \, \mathrm{d}(\mu_1 - \mu_2)(z) \right| \\
\lesssim & H S_a^4 B_a^4 B_x B_y \exp\big(4 S_a^2 B_a^2 B_x B_y\big) W_1(\mu_1, \mu_2)
\end{aligned}
$$

using the Kantorovich–Rubinstein duality

$$
W_1(\mu_1, \mu_2) = \sup_{\mathrm{Lip}(\phi) \le 1} \int \phi \, \mathrm{d}(\mu_1 - \mu_2),
$$

and the fact that $y \mapsto \exp\big(\langle Q^h x_1, K^h y \rangle\big)$ is $S_a^2 B_a^2 B_x \exp\big(S_a^2 B_a^2 B_x B_y\big)$-Lipschitz on $[-B_y, B_y]^{d_\mathrm{attn}}$ because

$$
\begin{aligned}
& \big| \exp\big(\langle Q^h x_1, K^h y_3 \rangle\big) - \exp\big(\langle Q^h x_1, K^h y_4 \rangle\big) \big| \\
\le & \exp\big(S_a^2 B_a^2 B_x B_y\big) \big| \langle Q^h x_1, K^h(y_3 - y_4) \rangle \big| \\
\le & S_a^2 B_a^2 B_x \exp\big(S_a^2 B_a^2 B_x B_y\big) \|y_3 - y_4\|_2.
\end{aligned}
$$

for $y_3, y_4 \in [-B_y, B_y]^{d_\mathrm{attn}}$.

**Bounding the term (ii).** We have

$$
\begin{aligned}
(ii) \lesssim & \sum_h S_a^2 B_a^2 \left| \frac{1}{\int \exp\big(\langle Q^h x_1, K^h z \rangle\big) \mathrm{d}\mu_2(z)} \right| \left\| \int y \exp\big(\langle Q^h x_1, K^h y \rangle\big) \, \mathrm{d}(\mu_1 - \mu_2)(y) \right\|_\infty \\
\lesssim & \sum_h S_a^2 B_a^2 \exp\big(S_a^2 B_a^2 B_x B_y\big) \max_{i=1,\ldots,d_\mathrm{attn}} \left| \int y_i \exp\big(\langle Q^h x_1, K^h y \rangle\big) \, \mathrm{d}(\mu_1 - \mu_2)(y) \right| \\
\lesssim & H(1 + S_a^2 B_a^2 B_x B_y) S_a^2 B_a^2 \exp\big(2 S_a^2 B_a^2 B_x B_y\big) W_1(\mu_1, \mu_2)
\end{aligned}
$$

using the Kantorovich–Rubinstein duality

$$
W_1(\mu_1, \mu_2) = \sup_{\mathrm{Lip}(\phi) \le 1} \int \phi \, \mathrm{d}(\mu_1 - \mu_2),
$$

and the fact that $y \mapsto y_i \exp\big(\langle Q^h x_1, K^h y \rangle\big)$ is $(1 + S_a^2 B_a^2 B_x B_y) \exp\big(S_a^2 B_a^2 B_x B_y\big)$-Lipschitz on $[-B_y, B_y]^{d_\mathrm{attn}}$.

Finally, we have

$$
\|\mathrm{Attn}_\theta(\mu_1, x_1) - \mathrm{Attn}_\theta(\mu_2, x_1)\|_\infty
$$

$$\lesssim H S_a^2 B_a^2 \left( S_a^2 B_a^2 B_x B_y \exp\left(4 S_a^2 B_a^2 B_x B_y\right) + (1 + S_a^2 B_a^2 B_x B_y) \exp\left(2 S_a^2 B_a^2 B_x B_y\right)\right)$$
$$\times W_1(\mu_1, \mu_2).$$

**Bounding the difference in $x$.** Next, we bound the difference in the query $x$. For fixed $\mu_1$, using similar interpolation and Lipschitz estimates in $x$,

$$\|\text{Attn}_\theta(\mu_1, x_1) - \text{Attn}_\theta(\mu_1, x_2)\|_\infty$$

$$\leq \sum_h S_a^2 B_a^2 \left\| \int \frac{y \exp\left(\langle Q^h x_1, K^h y\rangle\right)}{\int \exp\left(\langle Q^h x_1, K^h z\rangle\right) \mathrm{d}\mu_1(z)} - \frac{y \exp\left(\langle Q^h x_2, K^h y\rangle\right)}{\int \exp\left(\langle Q^h x_2, K^h z\rangle\right) \mathrm{d}\mu_1(z)} \mathrm{d}\mu_1(y) \right\|_\infty$$
$$+ S_a B_a \|x_1 - x_2\|_\infty$$

$$\leq \sum_h S_a^2 B_a^2 \left| \int \frac{y \exp\left(\langle Q^h x_1, K^h y\rangle\right)}{\int \exp\left(\langle Q^h x_1, K^h z\rangle\right) \mathrm{d}\mu_1(z)} \mathrm{d}\mu_1 - \int \frac{y \exp\left(\langle Q^h x_1, K^h y\rangle\right)}{\int \exp\left(\langle Q^h x_2, K^h z\rangle\right) \mathrm{d}\mu_1(z)} \mathrm{d}\mu_1(y) \right.$$

$$\left. + \int \frac{y \exp\left(\langle Q^h x_1, K^h y\rangle\right)}{\int \exp\left(\langle Q^h x_2, K^h z\rangle\right) \mathrm{d}\mu_1(z)} \mathrm{d}\mu_1 - \int \frac{y \exp\left(\langle Q^h x_2, K^h y\rangle\right)}{\int \exp\left(\langle Q^h x_2, K^h z\rangle\right) \mathrm{d}\mu_1(z)} \mathrm{d}\mu_1(y) \right|$$

$$+ S_a B_a \|x_1 - x_2\|_2$$

$$\lesssim (S_a B_a \vee H S_a^4 B_a^4 B_x B_y \exp\left(4 S_a^2 B_a^2 B_x B_y\right)) \|x_1 - x_2\|_2.$$

$\square$

By combining approximation and stability, we can control the complexity of the attention class via its covering number, as stated next.

We now bound the $\epsilon_2$-covering number of $\mathcal{A}(d_{\text{attn}}, H, B_a, S_a)$ in the parameter regime of interest.

**Lemma 10.** *The $\epsilon_2$-covering number of $\mathcal{A}(d_{\text{attn}}, H, B_a, S_a)$ is bounded by*

$$\mathcal{N}(\mathcal{A}(d_{\text{attn}}, H, B_a, S_a); \epsilon_2)_\infty$$
$$\lesssim \left(d_{\text{attn}}^2 \cdot \exp\left(O(\log(H) + S_a^2 B_a^2 B_x B_y)\right) \left(\epsilon_2^{-1} + 1\right)\right)^{O(S_a H)}.$$

*Furthermore, if $d_{\text{attn}} \lesssim d + D$, $H = O(D)$, $B_a \lesssim \sqrt{\log\left(I \epsilon_2^{-1}\right)}$, $S_a \lesssim d$, and $B_x, B_y = O(1)$, then the covering entropy is*

$$\log \mathcal{N}(\mathcal{A}(d_{\text{attn}}, H, B_a, S_a), \epsilon_2) \lesssim C_D \cdot D d^3 (\log I + \log \epsilon_2^{-1})^2 \cdot \log \epsilon_2^{-1}$$

*where $C_{d,D} \lesssim \text{poly}(\log D + \log d)$.*

*Proof.* We define a $\Omega(\bar{\epsilon})$-covering set of $\mathcal{A}(d_{\text{attn}}, H, B_a, S_a)$ as a set of mappings whose parameters can be constructed as follows:

- For each matrix $W^h, Q^h, K^h, V^h$ in each $h \in \{1, \ldots, H\}$,
    1. Choose $S_a$ matrix entries among $O(d_{\text{attn}}^2)$ entries.
    2. For each matrix entry,
        - Set its value from $\{(j \cdot \tilde{\epsilon} - 1) \cdot B_a \mid j = 0, \ldots, \lceil 2\tilde{\epsilon}^{-1}\rceil\}$ where $\tilde{\epsilon} \gtrsim \exp\left(-C_2(\log(H) + S_a^2 B_a^2 B_x B_y)\right) \bar{\epsilon}$ where $C_2$ is a sufficiently large constant.

- Set the value of chosen $S_a$ entries in $A$ from $\{(j \cdot \tilde{\epsilon} - 1) \cdot B_a \mid j = 0, \ldots, \lceil 2\tilde{\epsilon}'\rceil\}$ where $\tilde{\epsilon}' \simeq (S_a B_a)^{-1} \bar{\epsilon}$.

Let us prove that the above set of mappings is a $\Omega(\bar{\epsilon})$-covering. It is clear that for $W^h$ $h = 1, \ldots, H$, there exist matrices $\hat{W}^h$ in the $\bar{\epsilon}$-covering set such that

$$\|W^h - \hat{W}^h\|_\infty \lesssim H^{-1} \exp\left(-C_3(S_a^2 B_a^2 B_x B_y)\right) \bar{\epsilon}, \quad \|W^h - \hat{W}^h\|_0 \leq 2S_a$$

where $C_3$ is a sufficiently large constant. Similar inequalities hold true for $Q^h, K^h, V^h$, and $A$. Let $\hat{\theta} = (\hat{A}, (\hat{W}^h, \hat{Q}^h, \hat{K}^h, \hat{V}^h)_h)$. Then, for all $\mu \in \mathcal{P}([-B_y, B_y]^{d_{\text{attn}}})$ and $x \in [-B_x, B_x]^{d_{\text{attn}}}$,

$$\|\text{Attn}_\theta(\mu, x) - \text{Attn}_{\hat{\theta}}(\mu, x)\|_\infty$$

$$\leq \sum_h \left\| W^h \int \frac{V^h y \exp\left(\langle Q^h x, K^h y\rangle\right)}{\int \exp\left(\langle Q^h x, K^h z\rangle\right) d\mu(z)} d\mu - \hat{W}^h \int \frac{\hat{V}^h y \exp\left(\langle \hat{Q}^h x, \hat{K}^h y\rangle\right)}{\int \exp\left(\langle \hat{Q}^h x, \hat{K}^h z\rangle\right) d\mu(z)} d\mu \right\|_\infty$$

$$+ \|(A - \hat{A})x\|_\infty$$

$$\lesssim \sum_h \underbrace{\left\| W^h \int \frac{V^h y \exp\left(\langle Q^h x, K^h y\rangle\right)}{\int \exp\left(\langle Q^h x, K^h z\rangle\right) d\mu(z)} d\mu - \hat{W}^h \int \frac{V^h y \exp\left(\langle Q^h x, K^h y\rangle\right)}{\int \exp\left(\langle Q^h x, K^h z\rangle\right) d\mu(z)} d\mu \right\|_\infty}_{(i)}$$

$$+ \sum_h S_a B_a \underbrace{\left\| \int \frac{V^h y \exp\left(\langle Q^h x, K^h y\rangle\right)}{\int \exp\left(\langle Q^h x, K^h z\rangle\right) d\mu(z)} d\mu - \int \frac{V^h y \exp\left(\langle \hat{Q}^h x, K^h y\rangle\right)}{\int \exp\left(\langle Q^h x, K^h z\rangle\right) d\mu(z)} d\mu \right\|_\infty}_{(ii)}$$

$$+ \sum_h S_a B_a \underbrace{\left\| \int \frac{V^h y \exp\left(\langle \hat{Q}^h x, K^h y\rangle\right)}{\int \exp\left(\langle Q^h x, K^h z\rangle\right) d\mu(z)} d\mu - \int \frac{V^h y \exp\left(\langle \hat{Q}^h x, \hat{K}^h y\rangle\right)}{\int \exp\left(\langle Q^h x, K^h z\rangle\right) d\mu(z)} d\mu \right\|_\infty}_{(iii)}$$

$$+ \sum_h S_a B_a \underbrace{\left\| \int \frac{V^h y \exp\left(\langle \hat{Q}^h x, \hat{K}^h y\rangle\right)}{\int \exp\left(\langle Q^h x, K^h z\rangle\right) d\mu(z)} d\mu - \int \frac{V^h y \exp\left(\langle \hat{Q}^h x, \hat{K}^h y\rangle\right)}{\int \exp\left(\langle \hat{Q}^h x, K^h z\rangle\right) d\mu(z)} d\mu \right\|_\infty}_{(iv)}$$

$$+ \sum_h S_a B_a \underbrace{\left\| \int \frac{V^h y \exp\left(\langle \hat{Q}^h x, \hat{K}^h y\rangle\right)}{\int \exp\left(\langle \hat{Q}^h x, K^h z\rangle\right) d\mu(z)} d\mu - \int \frac{V^h y \exp\left(\langle \hat{Q}^h x, \hat{K}^h y\rangle\right)}{\int \exp\left(\langle \hat{Q}^h x, \hat{K}^h z\rangle\right) d\mu(z)} d\mu \right\|_\infty}_{(v)}$$

$$+ \sum_h S_a B_a \underbrace{\left\| \int \frac{V^h y \exp\left(\langle \hat{Q}^h x, \hat{K}^h y\rangle\right)}{\int \exp\left(\langle \hat{Q}^h x, \hat{K}^h z\rangle\right) d\mu(z)} d\mu - \int \frac{\hat{V}^h y \exp\left(\langle \hat{Q}^h x, \hat{K}^h y\rangle\right)}{\int \exp\left(\langle \hat{Q}^h x, \hat{K}^h z\rangle\right) d\mu(z)} d\mu \right\|_\infty}_{(vi)}$$

$$+ \underbrace{\|(A - \hat{A})x\|_\infty}_{(vii)}$$

$$\lesssim \bar{\epsilon}.$$

Each term is bounded as follows:

(i). Let $\boldsymbol{w} := \int \frac{V^h y \exp\left(\langle Q^h x, K^h y\rangle\right)}{\int \exp\left(\langle Q^h x, K^h z\rangle\right) d\mu(z)} d\mu$. Then, $(i) \leq \|(W^h - \hat{W}^h)\boldsymbol{w}\|_\infty \lesssim (2S_a) \cdot H^{-1} \exp\left(-C_3(S_a^2 B_a^2 B_x B_y)\right) \bar{\epsilon} \cdot \exp\left(O(S_a^2 B_a^2 B_x B_y)\right) \lesssim H^{-1}\bar{\epsilon}$ by (P-i,ii);

(ii). The second term is bounded by

$$(ii) \lesssim S_a B_a \left\| \sup_{\boldsymbol{y} \in \mathrm{supp}(\mu)} \left| \frac{V^h y_i}{\int \exp\left(\langle Q^h x, K^h z\rangle\right) d\mu(z)} \right. \right.$$
$$\left. \left. \times \left( \exp\left(\langle Q^h x, K^h y\rangle\right) - \exp\left(\langle \hat{Q}^h x, K^h y\rangle\right) \right) \right| \right\|_\infty$$
$$\lesssim S_a B_a \exp\left(O(S_a^2 B_a^2 B_x B_y)\right) \sup_{\boldsymbol{y} \in \mathrm{supp}(\mu)} \left| \langle Q^h x, K^h y\rangle - \langle \hat{Q}^h x, K^h y\rangle \right|$$
$$\lesssim S_a B_a \exp\left(O(S_a^2 B_a^2 B_x B_y)\right) \|(Q^h - \hat{Q}^h)x\|_\infty \sup_{\boldsymbol{y} \in \mathrm{supp}(\mu)} \left| \|K^h y\|_1 \right|$$
$$\lesssim \exp\left(O(S_a^2 B_a^2 B_x B_y)\right) \cdot H^{-1} \exp\left(-C_3(S_a^2 B_a^2 B_x B_y)\right) \bar{\epsilon}$$

$$\lesssim H^{-1}\bar{\epsilon}.$$

Note that the second inequality is derived by (P-ii,P-iv) and the fourth inequality is supported by (P-i).

(iii). The third term can be bounded in the same way as (ii).

(iv). The fourth term is bounded by

$$(iv) \lesssim S_a B_a \left\| \int V^h y \exp\left(\langle \hat{Q}^h x, \hat{K}^h y \rangle\right) \mathrm{d}\mu \right\|_\infty$$

$$\times \left| \frac{1}{\int \exp\left(\langle Q^h x, K^h z \rangle\right) \mathrm{d}\mu(z)} - \frac{1}{\int \exp\left(\langle \hat{Q}^h x, K^h z \rangle\right) \mathrm{d}\mu(z)} \right|$$

$$\lesssim S_a B_a \left\| \int V^h y \exp\left(\langle \hat{Q}^h x, \hat{K}^h y \rangle\right) \mathrm{d}\mu \right\|_\infty$$

$$\times \left( \min\left\{ \int \exp\left(\langle Q^h x, K^h z \rangle\right) \mathrm{d}\mu(z)^{-2}, \int \exp\left(\langle \hat{Q}^h x, K^h z \rangle\right) \mathrm{d}\mu(z) \right\} \right)^{-2}$$

$$\times \left| \int \exp\left(\langle \hat{Q}^h x, K^h z \rangle\right) - \exp\left(\langle \hat{Q}^h x, K^h z \rangle\right) \mathrm{d}\mu(z) \right| \quad \text{(by (P-iii))}$$

$$\lesssim \exp\left(O(S_a^2 B_a^2 B_x B_y)\right) \cdot \sup_{z \in \mathrm{supp}(\mu)} \left| \langle Q^h x, K^h z \rangle - \langle \hat{Q}^h x, K^h z \rangle \right|$$

$$\lesssim H^{-1}\bar{\epsilon};$$

(v). The fifth term is bounded in the similar way as (iv).

(vi). The sixth term is bounded in the same way as (i).

(vii). It is easily bounded by $H^{-1}\bar{\epsilon}$ using (P-i).

Please note that (P-i) $\|Ax\|_\infty, \|Ax\|_1 \le sb\|x\|_\infty$ where the number of non-zero entries in a matrix $A$ is bounded by $s$, and the absolute value of each entry is bounded by $b$, (P-ii) $|\langle Q^h x, K^h y \rangle| \le S_a B_a B_x \cdot S_a B_a B_y$ because each coordinate of $Q^h x$ is a sum of at most $S_a$ terms, each bounded by $B_a B_x$, and similarly each coordinate of $K^h y$ is bounded by $S_a B_a B_y$, (P-iii) $1/\alpha_1 - 1/\alpha_2 = (\alpha_1 - \alpha_2)/(\alpha_1 \alpha_2) \le A^{-2}|\alpha_1 - \alpha_2|$ when $A < \alpha_1, \alpha_2$, and (P-iv) $y \mapsto \exp(y)$ and $y \mapsto y \exp(y)$ are $\exp(O(B))$-Lipschitz over $[-B, B]$.

By the construction rule of the covering set, the covering number is bounded by

$$\mathcal{N}(\mathcal{A}(d_{\mathrm{attn}}, H, B_a, S_a); \bar{\epsilon})_\infty$$

$$\lesssim \left( \binom{d_{\mathrm{attn}}^2}{S_a} \bar{\epsilon}^{-S_a} \right)^{4H+1}$$

$$\lesssim \left( \binom{d_{\mathrm{attn}}^2}{S_a} \cdot \exp\left(O(S_a \log(H) + S_a^3 B_a^2 B_x B_y)\right) \left(\bar{\epsilon}^{-1} + 1\right)^{S_a} \right)^{4H+1}.$$

$\square$

Together, these results give a complete characterization of the first attention layer in the measure-theoretic setting: it can accurately realize $\phi_2$, does so in a stable manner, and has a covering number that scales favorably with $D$ and $\epsilon_2$.

## B.5 STEP 2-3: SECOND MLP LAYER

Having established in the previous subsections that the first MLP layer can approximate the Mercer basis functions $e_j$ and that the attention mechanism can extract the corresponding coefficients

$\int e_j \, \mathrm{d}\mu^{(i^*)}$ associated with the relevant component measure, we now turn to the next stage of the architecture.

In this step, the inputs to the model are effectively reduced to the finite collection of Mercer coefficients $(b_1, \ldots, b_D)$ together with the query vector $x$. The statistical problem is thus transformed into the approximation of a Lipschitz function defined over a $(D + O(1))$-dimensional domain.

Our goal in this section is twofold: first, to determine the appropriate truncation dimension $D$ that balances approximation error against complexity, and second, to establish approximation results for Lipschitz functions of $D$ variables using neural networks.

### B.5.1 DETERMINING THE DIMENSION $D$

As discussed above, after the first MLP and attention layers, the effective representation of the input measure $\mu_0$ is reduced to its Mercer coefficients with respect to the kernel eigenbasis $\{e_i\}_{i \geq 1}$. In practice, however, only a finite number of coefficients can be retained. Thus, a key question is: how many terms $D$ should be kept in the truncated expansion so that the approximation error remains negligible while the statistical complexity of the model is controlled? The following lemma quantifies the truncation error when approximating $\mu_0$ by its projection onto the first $D$ eigenfunctions.

**Lemma 11.** *Let $\mu_0 \in \mathcal{H}_0$ with Mercer expansion*

$$\frac{\mathrm{d}\mu_0}{\mathrm{d}\lambda} = \sum_{i=1}^{\infty} b_i e_i,$$

*where $\{e_i\}$ are the Mercer eigenfunctions associated with kernel eigenvalues $\{\lambda_i\}$ and $\lambda$ is the Lebesgue measure. Define the truncated approximation*

$$\tilde{\mu}_0 = \sum_{i=1}^{D} b_i e_i.$$

*If $\gamma_{\mathrm{f}} < 0$ and $\gamma_{\mathrm{b}} > 0$, then the truncation error in the $\gamma_{\mathrm{f}}$-norm is bounded as*

$$\|\mu_0 - \tilde{\mu}_0\|_{\gamma_{\mathrm{f}}} \leq \lambda_{D+1}^{\frac{-\gamma_{\mathrm{f}} + \gamma_{\mathrm{b}}}{2}}.$$

*Proof.* The LHS is bounded by

$$\|\mu_0 - \tilde{\mu}_0\|_{\gamma_{\mathrm{f}}}$$

$$= \sqrt{\sum_{i \geq D+1} \lambda_i^{-\gamma_{\mathrm{f}}} b_i^2}$$

$$= \sqrt{\lambda_{D+1}^{-\gamma_{\mathrm{f}} + \gamma_{\mathrm{b}}}} \sqrt{\sum_{i \geq D+1} \frac{\lambda_i^{-\gamma_{\mathrm{f}}}}{\lambda_{D+1}^{-\gamma_{\mathrm{f}} + \gamma_{\mathrm{b}}}} b_i^2}$$

$$\leq \sqrt{\lambda_{D+1}^{-\gamma_{\mathrm{f}} + \gamma_{\mathrm{b}}}} \sqrt{\sum_{i \geq D+1} \lambda_i^{-\gamma_{\mathrm{b}}} b_i^2} \qquad (\text{by} \quad \lambda_{D+1} \geq \lambda_i, \ i \geq D+1 \quad \text{and} \quad -\gamma_{\mathrm{f}} > 0)$$

$$\leq \sqrt{\lambda_{D+1}^{-\gamma_{\mathrm{f}} + \gamma_{\mathrm{b}}}} \qquad (\text{by} \quad \sqrt{\sum_i \lambda_i^{-\gamma_{\mathrm{b}}} b_i^2} \leq 1)$$

where we used $\frac{\lambda_i^{-\gamma_{\mathrm{f}}}}{\lambda_{D+1}^{-\gamma_{\mathrm{f}}}} \leq 1$ for $i \geq D+1$ in the first inequality, $\lambda_i^{-\gamma_{\mathrm{b}}} \geq \lambda_{D+1}^{-\gamma_{\mathrm{b}}}$ for $i \geq D+1$ and that $\mu_0$ is in the ball in the last inequality. $\qquad \square$

Having controlled the truncation error of the Mercer expansion, we next turn to the regularity of the target regression function $F^{\star}$. In particular, $F^{\star}$ is assumed to be Lipschitz continuous with respect to the product metric consisting of the $\gamma_{\mathrm{f}}$-weighted RKHS distance on measures and the standard Euclidean distance on the query variable. Formally, there exists $L > 0$ such that

$$\left| F^{\star}(\mu, x) - F^{\star}(\nu, y) \right| \leq L \left( \|\mu - \nu\|_{\mathcal{H}_0^{\gamma_{\mathrm{f}}}} + \|x - y\|_2 \right), \quad \forall \mu, \nu \in B(\mathcal{H}_0, \|\cdot\|_{\mathcal{H}_0^{\gamma_{\mathrm{b}}}}), \ x, y \in \mathbb{R}^{d_1}.$$

This Lipschitz property ensures that once the infinite-dimensional measure $\mu$ is replaced by its truncated $D$-dimensional approximation, the induced error on $F^\star$ can be directly bounded. The following corollary makes this reduction explicit.

**Corollary 2.** *Define the truncated regression function*

$$\bar{F}_D(\mu_0, v) \;:=\; F^\star\left(\sum_{i=1}^{D} b_i e_i,\, x\right), \quad \frac{\mathrm{d}\mu_0}{\mathrm{d}\lambda} = \sum_{i=1}^{\infty} b_i e_i, \quad x = \begin{bmatrix} v \\ 0 \end{bmatrix}.$$

*If $b_i = 0$ for $i \geq D + 1$, we simply write $\bar{F}_D(b_1, \ldots, b_D, v)$. Then,*

$$\sup_{\mu_0 \in B(\mathcal{H}_0, \|\cdot\|_{\mathcal{H}_0^{\gamma_\mathrm{b}}})} \left|\bar{F}_D(\mu_0, v) - F^\star(\mu_0, v)\right| \;\lesssim\; \lambda_{D+1}^{\frac{-\gamma_\mathrm{f} + \gamma_\mathrm{b}}{2}}.$$

*Moreover, $\bar{F}_D$ is Lipschitz with respect to the coefficients $(b_1, \ldots, b_D)$ and query $v$, satisfying*

$$\left|\bar{F}_D(b, v) - \bar{F}_D(c, v')\right| \;\leq\; L\sqrt{\max_{i \geq 1} \lambda_i^{-\gamma_\mathrm{f}}}\, \|b - c\|_2 + L\|v - v'\|_2$$

$$\lesssim O(1) \cdot \left\| \begin{bmatrix} b - c \\ v - v' \end{bmatrix} \right\|_2$$

*where $b = (b_1, \ldots, b_D)$ and $c = (c_1, \ldots, c_D)$. Note that $\gamma_\mathrm{f} < 0$, so the multiplicative factor in front of $\|b - c\|_2$ is $\Theta(1)$.*

### B.5.2 Approximating Finite-Dimensional Lipschitz Functions

Once the Mercer expansion has been truncated to $O(d_1 + D)$ coefficients, the infinite-dimensional regression problem reduces to approximating a Lipschitz function $f : [0,1]^{O(d_1+D)} \to \mathbb{R}$ with Lipschitz constant $K$. We now recall quantitative results on the approximation of such functions by deep ReLU networks.

**Lemma 12** (Schmidt-Hieber (2020)). *For any function $f \in \mathrm{Lip}_L([0,1]^{\bar{D}})$ and any integers $m \geq 1$ and $N \geq \exp\big(\Omega(\bar{D})\big)$. There exists a network*

$$\tilde{f} \in \mathcal{F}(\ell, (\bar{D}, 6(D+1)N, \ldots, 6(D+1)N, 1), s, \infty)$$

*with depth*

$$\ell \simeq (m+1)(1 + \log(\bar{D} + 1))$$

*and number of parameters*

$$s \lesssim (\bar{D} + 1)^{3 + \bar{D}} N(m + 6),$$

*such that*

$$\|\tilde{f} - f\|_{L^\infty} \lesssim (L+1)(1 + \bar{D}^2)6^{\bar{D}} N 2^{-m} + L N^{-1/\bar{D}}.$$

This lemma shows that deep ReLU networks can approximate any Lipschitz function on $[0,1]^{O(D)}$ with an explicit trade-off between network depth, width, and approximation error. The next remark connects this general result to our Mercer–RKHS setting.

**Lemma 13** (Schmidt-Hieber (2020)). *Let $V = \prod_{i=0}^{\ell}(p_i + 1)$. Then,*

$$\log \mathcal{N}(\mathcal{F}(\ell, p, s, F); \delta)_\infty \lesssim (s+1)\log\big(\delta^{-1} \ell V\big).$$

This bound shows that the covering entropy grows at most logarithmically with the resolution $\delta^{-1}$, once the architecture parameters $(\ell, p, s)$ are fixed. Applying our parameter selection yields the following implication.

Finally, for later use, we recall a useful estimate on the Lipschitz constant of a ReLU network in terms of its layer widths.

**Lemma 14** (From the proof of lemma 5 in Schmidt-Hieber (2020)). *The Lipschitz constant of NN (w.r.t. infinity norm) is $\prod_{i=0}^{\ell} p_i$.*

From the above lemmas, we have a specialized approximation results for our Mercer-RKHS setting:

**Corollary 3** (Specialization to Our Setting). *Under Setting 3 and assume $d_1 \simeq (\ln \epsilon^{-1})^{\beta^{-1}}$ so that*

$$\bar{D} \sim \left(\frac{-\gamma_f + \gamma_b}{2c}\right)^{-1/\alpha} (\ln \epsilon^{-1})^{1/\alpha} + d_1 \ \sim \ (\ln \epsilon^{-1})^{1/\min(\alpha,\beta)}.$$

*Letting*

$$m = O\left(\bar{D} \cdot \log \epsilon_3^{-1}\right), \qquad N = \epsilon_3^{-\bar{D}}$$

*in Lemma 12, there exists a ReLU network with depth*

$$\ell_2 \lesssim (\text{poly} \log(d + D) \cdot (d + D) \cdot \log \epsilon_3^{-1},$$

*width*

$$\|\boldsymbol{p}_2\|_\infty \lesssim (d + D)\epsilon_3^{-d+D}$$

*and the number of parameters*

$$s_2 \lesssim \tilde{O}\left(\epsilon_3^{-O\left((\ln \epsilon^{-1})^{1/\min(\alpha,\beta)}\right)} \cdot ((\ln \epsilon)^{\min(\alpha,\beta)})^{O((\ln \epsilon^{-1})^{1/\min(\alpha,\beta)})} \cdot (\text{poly} \log(d + D + \epsilon_3^{-1}))\right)$$

*that approximates $\bar{F}_D$, which was defined in Corollary 2, within sup-norm error $\lesssim \epsilon$.*

*Moreover, the covering entropy of the corresponding hypothesis class satisfies*

$$\log \mathcal{N}\left(\mathcal{F}(\ell, p, s, F); \epsilon_3\right)_\infty \lesssim \epsilon_3^{-O((\ln \epsilon^{-1})^{1/\min(\alpha,\beta)})} \cdot \text{poly} \log(\epsilon^{-1}\epsilon_3^{-1}),$$

*and the Lipschitz constant of the network (with respect to the $\ell_\infty$ norm) is bounded as*

$$\prod_{i=0}^{\ell} p_i \lesssim \epsilon_3^{-O\left(c_\epsilon(\ln \epsilon^{-1})^{\frac{2}{\min(\alpha,\beta)}} \cdot (\ln \epsilon_3^{-1})\right)}, \qquad c_\epsilon \lesssim \text{poly} \log \log \epsilon^{-1}.$$

*Instead of assuming $d_1 \lesssim (\log \epsilon^{-1})^{\beta^{-1}}$, if we assume that $F^\star$ is independent of $x_q$, then, by adding one layer $\mathbb{R}^{d+D} \ni x \mapsto x_{d+1:d+D} = -\text{ReLU}(-Ax) + \text{ReLU}(Ax) \in \mathbb{R}^D$ where $A = [O_{D \times d}; I_D]$, the ReLU network that approximates $\bar{F}_D$ with sup-error $\lesssim \epsilon_3$ is constructed with depth*

$$\ell_2 \lesssim (\text{poly} \log(D) \cdot (D) \cdot \log \epsilon_3^{-1},$$

*width*

$$\boldsymbol{p}_2 = (d + D, \underbrace{O(D\epsilon_3^{-D}), \ldots, O(D\epsilon_3^{-D})}_{\ell - 1 \ times})$$

*and the number of parameters*

$$s_2 \lesssim \tilde{O}\left(\epsilon_3^{-O\left((\ln \epsilon^{-1})^{1/\alpha}\right)} \cdot ((\ln \epsilon)^{1/\alpha})^{O((\ln \epsilon)^{1/\alpha})} \cdot (\text{poly} \log(D + \epsilon_3^{-1}) + D)\right).$$

*The covering entropy is bounded as*

$$\log \mathcal{N}\left(\mathcal{F}(\ell, p, s, F); \epsilon_3\right)_\infty \lesssim \epsilon_3^{-O((\ln \epsilon^{-1})^{1/\alpha})} \cdot \text{poly} \log(\epsilon^{-1}d\epsilon_3^{-1}),$$

*and the Lipschitz constant is*

$$\prod_{i=0}^{\ell} p_i \lesssim d_1 \cdot \epsilon_3^{-O\left(c_\epsilon(\ln \epsilon^{-1})^{\frac{2}{\alpha}} \cdot (\ln \epsilon_3^{-1})\right)}, \qquad c_\epsilon \lesssim \text{poly} \log \log \epsilon^{-1},$$

*where $D \simeq (\ln \epsilon^{-1})^{\alpha^{-1}}$.*

**Remark 4.** The original lemma of Schmidt-Hieber (2020) is stated for functions on $[0, 1]^{\bar{D}}$. In our setting, the domain is $[-O(1), O(1)]^{\bar{D}}$. A simple rescaling maps $[-O(1), O(1)]$ to $[0, 1]$, and this transformation only modifies the Lipschitz constant by a fixed multiplicative factor. Therefore, the approximation and covering results above remain valid up to universal constants.

This corollary consolidates the consequences of parameter selection in our setting: the effective input dimension $\bar{D}$ grows like $(\ln \epsilon^{-1})^{1/\alpha}$, the network size scales sub-exponentially in $1/\epsilon$, the covering entropy is controlled by $\epsilon^{-O((\ln \epsilon^{-1})^{1/\alpha})}$, and the Lipschitz constant grows at most quasi-polynomially in $\epsilon^{-1}$, when $\epsilon_3^{-1} \simeq \text{poly} \log \epsilon^{-1} \cdot \epsilon^{-1}$

## B.6 STEP 2-4: SECOND ATTENTION LAYER

Recall that the attention hypothesis class is parameterized as

$$\mathcal{A}(d_{\text{attn}}, H, B_a, B'_a, S_a, S'_a),$$

where $d_{\text{attn}}$ is the embedding dimension, $H$ is the number of heads, $B_a, B'_a$ are bounds on the operator norms of the weight matrices, and $S_a, S'_a$ are sparsity constraints.

In the present step, we only implement the skip connection of a scalar. We specialize to the case

$$d_{\text{attn}} = 1, \quad H = 1, \quad B_a = 0, \ B'_a = 1, \quad S_a = 0, \ S'_a = 1.$$

That is, the second attention layer belongs to the class

$$\mathcal{A}(1, 1, 0, 1, 0, 1).$$

This particular choice corresponds to a degenerate attention operator that is independent of the input measure and simply implements a skip connection acting as the identity on vectors, thereby ensuring consistency with the formal definition of the overall transformer class.

**Lemma 15.** *The $\epsilon_4$-covering number of $\mathcal{A}(d + D, 1, 0, 1, 0, d + D)$ satisfies*

$$\log \mathcal{N}(\mathcal{A}(1, 1, 0, 1, 0, 1); \delta)_{\infty} = O\big(\text{poly} \log \epsilon_4^{-1}\big).$$

*Proof.* omitted. □

**Lemma 16.** *Every attention operator* $\text{Attn} \in \mathcal{A}(1, 1, 0, 1, 0, 1)$ *is $O(1)$-Lipschitz with respect to the Euclidean norm.*

*Proof.* omitted. □

## B.7 DERIVING AN ESTIMATION ERROR UPPER BOUND

We now combine the approximation bounds established in the previous subsections to derive an estimation error guarantee for transformer-type architectures. Let TF denote the hypothesis class consisting of transformer models with the architecture and parameter constraints described in Section 3.

**Lemma 17** (Approximation by transformers). *In Setting 3, for all $\tilde{F}^{\star} \in \text{Lip}_L(B(\mathcal{H}_0, \|\cdot\|_{\mathcal{H}_0^{\gamma_b}}), \|\cdot\|_{\mathcal{H}_0^{\gamma_f}})$, there exists $\hat{F} \in \text{TF}$ such that, for any input of the form*

$$(\nu, x_{\text{q}}) \quad \text{where} \quad \nu = \frac{1}{I} \sum_{i=1}^{I} \mu_{v^{(i)}}^{(i)}, \quad x_{\text{q}} = \text{Emb}_{v^{(i^*)}}(0_{d_2})$$

*with $\mu_{v^{(i)}}^{(i)}$ generated according to Definition 1,*

$$|F^{\star}(\nu, x_{\text{q}}) - \hat{F}(\nu, x_{\text{q}})| \lesssim \epsilon,$$

*where the parameters $(d_j, H_j, B_{a,j}, B'_{a,j}, S_{a,j}, S'_{a,j}, \ell_j, \boldsymbol{p}_j, s_j)_{j=1}^2$ of the hypothesis set TF are defined as in Lemma 8 for $d_1, H_1, B_{a,1}, B'_{a,1}, S_{a,1}, S'_{a,1}$, Corollary 1 for $\ell_1, \boldsymbol{p}_1, s_1$, Lemma 15 for $d_2, H_2, B_{a,2}, B'_{a,2}, S_{a,2}, S'_{a,2}$, Corollary 3 for $\ell_2, \boldsymbol{p}_2, s_2$, respectively. The effective dimension $D$ in them are determined in Corollary 2. Determination of $\epsilon_i$, $i = 1, \ldots, 4$ are deferred to Lemma 18.*

The above lemma shows that the transformer hypothesis class is sufficiently rich to approximate any Lipschitz target function $\tilde{F}^{\star}$ on the admissible input domain, with uniform accuracy $\epsilon$. Please note that the output of each layer is uniformly bounded (we can add a clipping ReLU layer for each layer).

To analyze the statistical performance of ERM (empirical risk minimizer) within this class, we next require an upper bound on its covering entropy.

**Lemma 18** (Covering entropy of transformers). *The covering entropy of the transformer hypothesis class satisfies*

$$\log\mathcal{N}(\mathrm{TF};\epsilon)_\infty \lesssim \epsilon^{-O((\ln\epsilon^{-1})^{1/\min(\alpha,\beta)})} = \exp\Big(O(\ln\epsilon^{-1})^{(1+\min(\alpha,\beta))/\min(\alpha,\beta)}\Big)$$

*assuming that $I \leq d_1 \simeq (\ln\epsilon^{-1})^{\beta^{-1}}$ and $d_2 \simeq 1$.*

*Proof.* The claim follows from applying the composition lemma (Lemma 5) for covering numbers. In particular,

$$\log\mathcal{N}(\mathrm{TF};\epsilon)_\infty$$

$$\lesssim \log\mathcal{N}\left(\mathcal{A}(1,1,0,1,0,1);\epsilon\right)_\infty + \log\mathcal{N}\left(\mathcal{F}(\ell_2,p_2,s_2);\tilde\Omega(\epsilon)\right)_\infty$$

$$+ \log\mathcal{N}\left(\{\mathrm{Attn}_{\theta_1}\diamond\mathrm{MLP}_{\xi_1}\};\Omega(L_{\mathrm{MLP},2}^{-1}\epsilon)\right)$$

$$\lesssim \log\mathcal{N}\left(\mathcal{A}(1,1,0,1,0,1);\underbrace{\epsilon}_{=:\epsilon_4}\right)_\infty$$

$$+ \log\mathcal{N}\left(\mathcal{F}(\ell_2,p_2,s_2);\underbrace{\tilde\Omega(\epsilon)}_{=:\epsilon_3}\right)_\infty + \log\mathcal{N}\left(\mathcal{A}(d_{\mathrm{attn}},H,B_a,H_a);\underbrace{\tilde\Omega(L_{\mathrm{MLP},2}^{-1}\epsilon)}_{=:\epsilon_2}\right)_\infty$$

$$+ \log\mathcal{N}\left(\mathcal{F}(\ell_1,p_1,s_1);\underbrace{\tilde\Omega(L_{\mathrm{MLP},2}^{-1}(L_{\mathrm{Attn},1,W^1}+L_{\mathrm{Attn},1,\|\cdot\|_2})^{-1}\epsilon)}_{=:\epsilon_1}\right)_\infty$$

$$\lesssim \log\mathcal{N}\left(\mathcal{A}(d+D,1,0,1,0,d+D);\epsilon\right)_\infty$$

$$+ \log\mathcal{N}\left(\mathcal{F}(\ell_2,p_2,s_2);\tilde\Omega(\epsilon)\right)_\infty + \log\mathcal{N}\left(\mathcal{A}(d_{\mathrm{attn}},H,B_a,H_a);\exp\Big(-O(c_\epsilon\ln\epsilon^{-1})^{\frac{2+2\min(\alpha,\beta)}{\min(\alpha,\beta)}}\Big)\right)_\infty$$

$$+ \log\mathcal{N}\left(\mathcal{F}(\ell_1,p_1,s_1);I^{-1}\cdot\exp\Big(-O(c_\epsilon\ln\epsilon^{-1})^{\frac{2+2\min(\alpha,\beta)}{\min(\alpha,\beta)}}\Big)\right)_\infty,$$

where $c_\epsilon \lesssim \mathrm{poly}\log\log\epsilon^{-1}$. Here we used that

(i) Letting $\epsilon_4 = \epsilon$, the second attention layer is $\tilde O(1)$-Lipschitz (Lemma 16);

(ii) Letting $\epsilon_3 \gtrsim \tilde\Omega(\epsilon)$, the Lipschitz constant of the second MLP layer is bounded as $L_{\mathrm{MLP},2} \lesssim \exp\big(-O\big(c_\epsilon((\ln\epsilon^{-1})^{2/\min(\alpha,\beta)})\cdot(\ln\epsilon_3^{-1})^2\big)\big) = \exp\Big(c_\epsilon(\ln\epsilon^{-1})^{\frac{2+2\min(\alpha,\beta)}{\min(\alpha,\beta)}}\Big)$ ($c_\epsilon \lesssim \mathrm{poly}\log\log\epsilon^{-1}$) (Corollary 3);

(iii) Letting $\epsilon_2 \gtrsim \exp\Big(-O(c_\epsilon\ln\epsilon^{-1})^{\frac{2+2\min(\alpha,\beta)}{\min(\alpha,\beta)}}\Big)$, the Lipschitz constants of the first attention layer are bounded as $L_{\mathrm{Attn},1,W^1} + L_{\mathrm{Attn},1,\|\cdot\|_2} \lesssim D\exp\big(O(d^2\log(I\epsilon_2^{-1}))\big) \lesssim I^{O(1)}\cdot \exp\Big(O(c_\epsilon d^2(\ln\epsilon^{-1})^{\frac{2+2\min(\alpha,\beta)}{\min(\alpha,\beta)}})\Big)$ (Lemma 9);

(iv) We have $\epsilon_1 \gtrsim I^{-O(1)}\cdot\exp\Big(-O(c_\epsilon d^2(\ln\epsilon^{-1})^{\frac{2+2\min(\alpha,\beta)}{\min(\alpha,\beta)}})\Big)$ for the first MLP layer.

By Lemma 15 and Corollary 3,

$$\log\mathcal{N}\left(\mathcal{A}(1,1,0,1,0,1);\epsilon_4\right)_\infty \lesssim \mathrm{poly}\log\epsilon^{-1}$$

By Corollary 3,

$$\log\mathcal{N}\left(\mathcal{F}(\ell_2,p_2,s_2);\epsilon_3\right)_\infty \lesssim \epsilon^{-O((\ln\epsilon^{-1})^{1/\min(\alpha,\beta)})}.$$

By Lemma 10,

$$\log\mathcal{N}\left(\mathcal{A}(d_{\mathrm{attn}},H,B_a,H_a);\epsilon_2\right)_\infty \lesssim \mathrm{poly}(\log\epsilon^{-1}+\log I)\cdot d^3.$$

By Corollary 1

$$\log \mathcal{N}\left(\mathcal{F}(\ell_1, p_1, s_1); \epsilon_1\right)_\infty \lesssim \text{poly} \log \log I \cdot \text{poly} \log \epsilon^{-1} \cdot (((\log I) + \text{poly} \log \epsilon^{-1} \cdot d)^{4d_2} \cdot D + d).$$

Assuming that $I \leq d_1 \simeq (\log \epsilon)^{\beta^{-1}}$ and $d_2 = O(1)$, we have

$$\log \mathcal{N}\left(\mathcal{A}(d_{\text{attn}1}, H_1, B_{a,1}, H_{a,1}); \epsilon_2\right)_\infty + \log \mathcal{N}\left(\mathcal{F}(\ell_1, p_1, s_1); \epsilon_1\right)_\infty$$
$$+ \log \mathcal{N}\left(\mathcal{F}(\ell_2, p_2, s_2); \epsilon_3\right)_\infty + \log \mathcal{N}\left(\mathcal{A}(1, 1, 0, 1, 0, 1); \epsilon_4\right)_\infty$$

$$\lesssim \epsilon^{-O((\ln \epsilon^{-1})^{1/\min(\alpha, \beta)})}.$$

$$\square$$

With these ingredients, we can invoke a general statistical learning bound for ERM.

**Lemma 19** (Schmidt-Hieber (2020)). *Consider Gaussian regression, and let $\hat{F}$ be the empirical risk minimizer over a hypothesis class $\mathcal{F} \subset L^2(\mathbb{P}_{\nu, x_q})$. Suppose $\|f\|_{L^\infty} \leq A$ for all $f \in \mathcal{F}$. Then, for any $\delta > 0$, if $V(\delta)$ denotes the covering entropy of $\mathcal{F}$, it holds that*

$$R(\bar{F}^\star, \hat{F}) \lesssim \inf_{f \in \mathcal{F}} \|\hat{F} - \bar{F}^\star\|_{L^2(\mathbb{P}_{\nu, x_q})}^2 + \frac{(A^2 + \sigma^2)V(\delta)}{n} + (A + \sigma)\delta.$$

We are now ready to state the statistical rate achieved by transformer ERM.

### B.7.1 SUB-POLYNOMIAL CONVERGENCE RATE

**Theorem 5** (Sub-polynomial convergence). *Let $\hat{F}$ be the empirical risk minimizer whose hypothesis set is constructed in Lemma 17. In Setting 3, we have*

$$R(F^\star, \hat{F}) \lesssim \exp\left(-\Omega((\ln n)^{\frac{\min(\alpha, \beta)}{\min(\alpha, \beta)+1}})\right)$$

*assuming that the number of mixture components is bounded as $I \leq d_1 \lesssim (\ln n)^{\frac{\beta^{-1}\min(\alpha, \beta)}{\min(\alpha, \beta)+1}}$ and $d_2 \simeq 1$.*

*Proof.* Let $\epsilon \simeq \exp\left(-c'(\ln n)^{\frac{\min(\alpha, \beta)}{\min(\alpha, \beta)+1}}\right)$ for sufficiently small constant $c' > 0$. Combining Lemmas 17 and 18 with Lemma 19, we obtain

$$R(F^\star, \hat{F}) \lesssim \epsilon + \frac{\exp\left(O((\ln \epsilon^{-1})^{1/\min(\alpha, \beta)+1})\right)}{n}$$

$$\lesssim \epsilon + \frac{\exp\left(O(c'^{(\min(\alpha, \beta)^{-1}+1)}(\ln n))\right)}{n}$$

$$\leq \epsilon + \frac{n^{c''}}{n} \quad \text{for some } 0 < c'' < 1,$$

$$\lesssim \exp\left(-\Omega((\ln n)^{\frac{\min(\alpha, \beta)}{1+\min(\alpha, \beta)}})\right),$$

assuming that the number of mixture components is bounded as $I \leq d_1 \lesssim (\ln \epsilon^{-1})^{\beta^{-1}}$ $\square$

**Remark 5** (Interpretation of Theorem 5). A common statistical learning bound for nonparametric regression takes the form

$$R(\hat{F}, \bar{F}^\star) \lesssim n^{-\Theta(1/d)},$$

where $d$ is the (effective) dimension of the problem. In our setting, however, the eigenvalue decay assumption $\lambda_j \simeq \exp(-cj^\alpha)$ implies that the effective dimension grows only as

$$d \sim (\ln n)^{1/(\alpha+1)}.$$

Consequently, the bound in Theorem 5 can be interpreted as a direct analogue of the classical $n^{-\Theta(1/d)}$ rate, but with $d$ replaced by $(\ln n)^{1/(\alpha+1)}$. Importantly, this shows that the estimator bypasses the usual combinatorial difficulty of associative recall tasks. In our framework, each element to be recalled is not a finite symbol but rather a *probability measure*, i.e. an infinite-dimensional object. Despite this intrinsic complexity, the analysis reveals that the statistical behavior is governed purely by the eigenvalue decay of the underlying kernel, leading to the rate characteristic of infinite-dimensional regression.

### B.7.2 BEYOND LOGARITHMIC CAPACITY

In Theorem 5, we discussed the case that the number of components (the "capacity" in terms of the associative memory) is bounded as

$$I \leq d_1 \lesssim (\ln n)^{\frac{\beta^{-1} \min(\alpha,\beta)}{\min(\alpha,\beta)+1}},$$

which is logarithmic with respect to not only the sample size $n$, but also the number of "parameters", which we consider as the covering number, of our Transformer models. This is because the lipschitz functions over $\mathbb{S}^{d_1-1}$, not $B(\mathcal{H}_0, \|\cdot\|_{\mathcal{H}_0^{\gamma_b}})$, becomes too complex when $d_1$ is large. On the other hand, in Appendix B.4, we observed that the number of the actual parameters attention matrix is linear in $d_1 (\geq I)$.

Here we consider the following additional assumption:

**Assumption 9.** The target function $\tilde{F}^\star(\mu_0^{(i^*)}, x_q)$ is independent of $x_q$ and only dependent of $\mu_0^{(i^*)}$. (i.e. we can write $\tilde{F}^\star(\mu_0^{(i^*)}, x_q) = \tilde{F}^\star(\mu_0^{(i^*)})$.)

Then, we have the polynomial "capacity" even in the associative recall task with the infinite-dimensional measure-valued components:

**Lemma 20.** *Assume that*

$$I \leq d_1 \simeq \exp\Big(O((\log \epsilon^{-1})^{\frac{\alpha+1}{\alpha}})\Big), \quad d_2 = O(1).$$

*In Setting 3 and under the additional Assumption 9,*

$$\log \mathcal{N}(\mathrm{TF}; \epsilon)_\infty \lesssim \exp\Big(O(\ln \epsilon^{-1})^{(1+\alpha)/\alpha}\Big)$$

*Proof.* The main strategy of this lemma follows Lemma 18. We only mention the differences from the preceding lemma.

Let $\bar{D} = d + D \simeq d_1$ (consider the case $D \ll d_1$).

(i) Letting $\epsilon_4 = \epsilon$, the second attention layer is $O(1)$-Lipschitz (Lemma 16);

(ii) Letting $\epsilon_3 \gtrsim \tilde{\Omega}(\epsilon)$, the Lipschitz constant of the second MLP layer is bounded as $L_{\mathrm{MLP},2} \lesssim d_1 \exp\big(-O\big(c_\epsilon D^2 \cdot (\ln \epsilon^{-1})^2\big)\big)$, $(c_\epsilon \lesssim \mathrm{poly} \log \log \epsilon^{-1})$ (modifying Corollary 3 to ignore the first $d$ indices );

(iii) Letting $\epsilon_2 \gtrsim d_1 \exp\big(-O\big(c_\epsilon D^2 \cdot (\ln \epsilon^{-1})^2\big)\big)$, the Lipschitz constants of the first attention layer are bounded as $L_{\mathrm{Attn},1,W^1} + L_{\mathrm{Attn},1,\|\|_2} \lesssim D \exp\big(O(d^2 \log\big(I\epsilon_2^{-1}\big))\big) \lesssim \exp\Big(O(c_\epsilon d_1^2 (\ln d_1)^{O(1)} (\ln \epsilon^{-1})^{2+2\alpha^{-1}})\Big)$ (Lemma 9 and $I \leq d_1$);

(iv) We have $\epsilon_1 \gtrsim \exp\Big(-O(c_\epsilon d_1^2 (\ln d_1)^{O(1)} (\ln \epsilon^{-1})^{2+2\alpha^{-1}})\Big)$ for the first MLP layer.

By Lemma 15 and Corollary 3,

$$\log \mathcal{N}\left(\mathcal{A}(1,1,0,1,0,1); \epsilon_4\right)_\infty \lesssim \mathrm{poly} \log \epsilon^{-1}$$

By modifying Corollary 3 to ignore the first $d$ indices (corresponding to the query),

$$\log \mathcal{N}\left(\mathcal{F}(\ell_2, p_2, s_2); \epsilon_3\right)_\infty \lesssim \epsilon^{-O((\ln \epsilon^{-1})^{1/\alpha})}.$$

By Lemma 10,

$$\log \mathcal{N}\left(\mathcal{A}(d_{\mathrm{attn}1}, H_1, B_{a,1}, H_{a,1}); \epsilon_2\right)_\infty \lesssim \mathrm{poly}(\log\big(\epsilon^{-1} d_1\big)) \cdot d^3.$$

By Corollary 1

$$\log \mathcal{N}\left(\mathcal{F}(\ell_1, p_1, s_1); \epsilon_1\right)_\infty \lesssim \mathrm{poly} \log \log I \cdot \mathrm{poly} \log \epsilon^{-1} \cdot (((\log I) + \mathrm{poly} \log \epsilon^{-1} \cdot d)^{4d_2} \cdot D + d).$$

Assuming that $I \leq d_1 \simeq \exp\Big(O((\log \epsilon)^{\frac{\alpha+1}{\alpha}})\Big)$ and $d_2 = O(1)$, we have

$$\log \mathcal{N}\left(\mathcal{A}(d_{\mathrm{attn}1}, H_1, B_{a,1}, H_{a,1}); \epsilon_2\right)_\infty + \log \mathcal{N}\left(\mathcal{F}(\ell_1, p_1, s_1); \epsilon_1\right)_\infty$$

$$+ \log \mathcal{N} \left( \mathcal{F}(\ell_2, p_2, s_2); \epsilon_3 \right)_\infty + \log \mathcal{N} \left( \mathcal{A}(1, 1, 0, 1, 0, 1); \epsilon_4 \right)_\infty$$
$$\lesssim \epsilon^{-O((\ln \epsilon^{-1})^{1/\alpha})}.$$

$\square$

In the same vein, we obtain the similar result as in Theorem 5:

**Theorem 6.** *Let $\hat{F}$ be the empirical risk minimizer whose hypothesis set is constructed in Lemma 17. Assume that*
$$I \leq d_1 \simeq \exp(o(\log n)) = n^{o(1)}$$
*and the target function $F^\star(\mu_0^{(i^*)}, x_q)$ is independent of $x_q$ (Assumption 9). In Setting 3, we have*
$$R(F^\star, \hat{F}) \lesssim \exp\left(-\Omega((\ln n)^{\frac{\alpha}{\alpha+1}})\right).$$

*Proof.* The proof is the same as in Theorem 5. $\square$

## C    MINIMAX LOWER BOUND

### C.1    PROOF SKETCH

The goal of this section is to establish an information-theoretic minimax lower bound for the associative recall problem in Setting 4. Our proof strategy consists of two main steps: a reduction to a pure infinite-dimensional regression task, and the derivation of covering/packing entropy bounds for the corresponding Lipschitz functionals.

**Step 1: Reduction from Infinite-Dimensional Regression (Appendix C.2).**    We first reduce a regression problem on measures to the associative recall problem. Let
$$\mathcal{F}^\star = \mathrm{Lip}_L(B(\mathcal{H}_0, \|\cdot\|_{\mathcal{H}_0^{\gamma_b}}) \times \mathcal{X}_q, d_{\mathrm{prod}}), \qquad \bar{\mathcal{F}}^\star = \mathrm{Lip}_L(B(\mathcal{H}_0, \|\cdot\|_{\mathcal{H}_0^{\gamma_b}}), \|\cdot\|_{\mathcal{H}_0^{\gamma_f}}),$$

where $\mathcal{F}^\star$ is the full class of Lipschitz functions depending on both $(\mu, x)$, and $\bar{\mathcal{F}}^\star \subset \mathcal{F}^\star$ is the subclass depending only on $\mu$.

Here, the key observation is that each $\nu$ can be written as an average of pushforward measures
$$\nu \;=\; \frac{1}{I} \sum_{i=1}^{I} \mu_{v^{(i)}}^{(i)} \;=\; \frac{1}{I} \sum_{i=1}^{I} (\mathrm{Emb}_{v^{(i)}})_\sharp \mu_0^{(i)}.$$

Therefore, estimating $F(\mu_0^{(i^*)}, x)$ with the "noisy" input $\nu$ is at least as hard as estimating with the "pure" input $\mu_0^{(i^*)}$.

Formally, the two observation models differ as follows:
$$\mathcal{S}_n = \{(\nu_t, (x_q)_t, y_t)\}_{t=1}^n, \quad y_t = \tilde{F}^\star(\mu_0^{(i^*)}, (x_q)_t) + \xi_t, \quad \tilde{F}^\star \in \mathcal{F}^\star,$$
$$\mathcal{U}_n = \{(\mu_0^{(i^*)}, y_t)\}_{t=1}^n, \quad y_t = \tilde{F}^\star(\mu_0^{(i^*)}) + \xi_t, \quad \tilde{F}^\star \in \bar{\mathcal{F}}^\star \subset \mathcal{F}^\star.$$

That is, under $\mathcal{S}_n$ we observe outputs of a general Lipschitz function $\tilde{F}^\star \in \mathcal{F}^\star$, whereas under $\mathcal{U}_n$ the outputs are restricted to the subclass $\bar{\mathcal{F}}^\star$.

Consequently, remembering that $F^\star(\nu, x) := \tilde{F}^\star(\mu_0^{(i^*)}, x)$, the minimax risk satisfies
$$\inf_{\hat{F}} \sup_{\tilde{F}^\star \in \mathcal{F}^\star} \mathbb{E}_{\mathcal{S}_n}\left[\|\hat{F}(\nu, x) - F^\star(\nu, x)\|^2\right] \;\geq\; \inf_{\hat{F}} \sup_{\tilde{F}^\star \in \bar{\mathcal{F}}^\star} \mathbb{E}_{\mathcal{U}_n}\left[\|\hat{F}(\mu_0) - \tilde{F}^\star(\mu_0)\|^2\right].$$

In words: the associative recall problem with dataset $\mathcal{S}_n$ and hypothesis class $\mathcal{F}^\star$ is at least as hard as the reduced regression problem with dataset $\mathcal{U}_n$ and restricted class $\bar{\mathcal{F}}^\star$. This reduction allows us to focus on an *infinite-dimensional regression* setting.

**Step 2: Entropy Bounds for Lipschitz Functionals.** The minimax lower bound is based on the general Gaussian regression minimax bound in Yang & Barron (1999): in short,

$$\log \mathcal{M}(\mathcal{F}^\star; \epsilon)_{L^2(\mathbb{P}_{\mu_0})} \simeq n\epsilon^2 \quad \Rightarrow \quad \inf_{\hat{F}} \sup_{F^\star} R(F^\star, \hat{F}) \gtrsim \epsilon^2,$$

where $\mathcal{M}(\mathcal{F}^\star; \epsilon)_{L^2(\mathbb{P}_{\mu_0})}$ denotes the $\epsilon$-packing number of the function class $\mathcal{F}^\star$ with respect to the $L^2(\mathbb{P}_{\mu_0})$ metric. Thus, to obtain a lower bound it suffices to evaluate the packing entropy (i.e., the metric entropy $\log \mathcal{M}$ ($\simeq \log \mathcal{N}$)) of the set of Lipschitz functionals $G^\star$ under $L^2(\mathbb{P}_{\mu_0})$. To apply the classical information-theoretic results, we derive both upper and lower bounds for the covering/packing entropy of the relevant Lipschitz functional class.

STEP 2-1: UPPER BOUND (APPENDIX C.3). It is known (see Boissard (2011)) that for a Lipschitz class,

$$\log \mathcal{N}(\text{Lip}(A, d); \epsilon)_{L^\infty} \lesssim \mathcal{N}(A; \epsilon)_d \cdot (\text{poly} \log \epsilon^{-1}),$$

where $A$ is the input set and $d$ is the underlying metric. Applying this principle, we show that

$$\log \mathcal{N}\big(B(\mathcal{H}_0, \|\cdot\|_{\mathcal{H}_0^{\gamma_b}}); \epsilon\big)_{\|\cdot\|_{\mathcal{H}_0^{\gamma_f}}} \lesssim \exp\left(c_u \log \epsilon^{-1}\right)^{\frac{\alpha+1}{\alpha}}, \quad c_u < \infty.$$

The proof relies on the following isometric transformation: for $f = \sum_{i=1}^\infty b_i e_i$ and parameters $a, b, c$,

$$f \mapsto \phi_{a,b,c}(f) := \sum_{i=1}^\infty \lambda_i^{\frac{c-b}{2}} b_i e_i,$$

which is an isometric bijection from $(B(\mathcal{H}_0, \|\cdot\|_{\mathcal{H}_0^a}), \|\cdot\|_{\mathcal{H}_0^b})$ to $(B(\mathcal{H}_0, \|\cdot\|_{\mathcal{H}_0^{a-b+c}}), \|\cdot\|_{\mathcal{H}_0^c})$. We apply this with $a = \gamma_b$, $b = \gamma_f$, and $c = 0$. Then, we employ a standard argument of covering an infinite-dimensional ellipsoid endowed with $\ell^2$ metric.

STEP 2-2: LOWER BOUND (APPENDIX C.4). The key difficulty is that the Lipschitz constant is *anisotropic*: differences along low-index directions (small $j$) are heavily penalized, while directions with larger $j$ are effectively much smoother. Formally, for $F$ in our class one has

$$\left|F\left(\sum_j b_j e_j\right) - F\left(\sum_j c_j e_j\right)\right| \leq L \sqrt{\sum_j \lambda_j^{-\gamma_f} (b_j - c_j)^2}, \quad \gamma_f < 0,$$

which clearly shows that directions with larger eigenvalues $\lambda_j$ (small $j$) contribute far more to the Lipschitz bound than those with smaller eigenvalues (large $j$). In other words, the geometry of the function class is highly distorted across coordinates.

To make this structure explicit, we construct a rescaling map that embeds the standard cube $[0, 1]^d$ into our measure-input space. After rescaling each coordinate according to the eigenvalue decay $\{\lambda_j\}$, a Lipschitz function on $[0, 1]^d$ becomes a function on $B(\mathcal{H}_0, \|\cdot\|_{\mathcal{H}_0^{\gamma_b}})$ with Lipschitz constant proportional to

$$\lambda_d^{(-\gamma_f + \gamma_d)/2}.$$

This shows that our class contains an embedded copy of the $d$-dimensional Lipschitz ball, up to a rescaling factor.

Consequently, the packing entropy of our class is at least as large as that of $\text{Lip}_1([0, 1]^d, \|\cdot\|_{\ell^\infty})$ at resolution $R\epsilon/\lambda_d^{(-\gamma_f + \gamma_d)/2}$. By combining this rescaling argument with known packing lower bounds for Lipschitz functions on $[0, 1]^d$ and the Yang–Barron information-theoretic inequality, we obtain the desired minimax lower bound for the associative recall problem.

**Remark 6.** In Setting 4, we assume that the density $\frac{d\mu_0}{d\lambda}$ is *nonnegative*, which can always be ensured by adding a sufficiently large constant shift. We then relax the additional constraint that $\mu_0$ must be a probability measure, and instead only require that $\frac{d\mu_0}{d\lambda}$ belongs to a bounded ball in the ambient function space. This modification does not affect the minimax difficulty of the problem, since the essential hardness arises from the *infinite-dimensionality* of the domain, whereas the normalization constraint corresponds merely to a *finite-dimensional* restriction.

## C.2 STEP 1: REDUCTION FROM INFINITE-DIMENSIONAL REGRESSION

By Lemma 21, we will show that the associative recall problem is at least as hard as a Gaussian regression problem where the input variables are measures. Remember that a standard Gaussian regression problem is defined as follows: we observe i.i.d. random variables $(X_t, Y_t)_{t=1}^n$ such that

$$Y_t = F^\star(X_t) + \xi_t, \quad t = 1, \dots, n,$$

where $\xi_t$ are i.i.d. Gaussian noise, independent of $X_t$. On the other hand, in our recall-and-predict problem, the observed input $\nu$ is noisy: $(\mu_{v^{(i)}}^{(i)})_{i \neq i^*}$ are mixed in $\nu$, but they are irrelevant to the output $y$. We will show that we can obtain a better estimator when we "eliminate" the noises in the inputs. Formally, we obtain the following corollary:

**Corollary 4.** *Let*

$$\mathcal{F}^\star := \mathrm{Lip}_L(B(\mathcal{H}_0, \|\cdot\|_{\mathcal{H}_0^{\gamma b}}) \times \mathcal{X}_q, d_{\mathrm{prod}}), \qquad \bar{\mathcal{F}}^\star := \mathrm{Lip}_L(B(\mathcal{H}_0, \|\cdot\|_{\mathcal{H}_0^{\gamma b}}), \|\cdot\|_{\mathcal{H}_0^{\gamma f}}).$$

*Here, $\mathcal{F}^\star$ denotes the class of L-Lipschitz functions in both $(\mu, x)$, while $\bar{\mathcal{F}}^\star$ denotes the subclass of L-Lipschitz functions depending only on $\mu$. Note that $\tilde{\mathcal{F}}^\star \subset \mathcal{F}^\star$.*

*Consider datasets*

$$\mathcal{S}_n = \{(\nu_t, (x_q)_t, y_t)\}_{t=1}^n, \quad y_t = \tilde{F}^\star(\mu_0^{(i^*)}, (x_q)_t) + \xi_t, \quad \tilde{F}^\star \in \mathcal{F}^\star,$$

$$\mathcal{U}_n = \{(\mu_0^{(i^*)}, y_t)\}_{t=1}^n, \quad y_t = \tilde{F}^\star(\mu_0^{(i^*)}) + \xi_t, \quad \tilde{F}^\star \in \bar{\mathcal{F}}^\star \subset \mathcal{F}^\star.$$

*sampled as in Setting 4. Then, remembering that $F^\star(\nu, x_q) = \tilde{F}^\star(\mu_0^{(i^*)}, x_q)$, we have*

$$\inf_{\mathcal{S}_n \mapsto \hat{F} \in L^2(\mathbb{P}_{\nu, x_q})} \sup_{\tilde{F}^\star \in \mathcal{F}^\star} \mathbb{E}_{\mathcal{S}_n}\big[\|\hat{F}(\nu, x) - F^\star(\nu, x)\|_{L^2(\mathbb{P}_{\nu, x_q})}^2\big]$$

$$\geq \inf_{\mathcal{U}_n \mapsto \hat{F} \in L^2(\mathbb{P}_{\mu_0})} \sup_{\tilde{F}^\star \in \bar{\mathcal{F}}^\star} \mathbb{E}_{\mathcal{U}_n}\big[\|\hat{F}(\mu_0) - \tilde{F}^\star(\mu_0)\|_{L^2(\mathbb{P}_{\mu_0})}^2\big].$$

In words: the estimation problem with query-dependent target functions is at least as hard as the restricted problem where the target depends only on $\mu$. Hence, establishing a lower bound for the latter suffices.

To prove Corollary 4, we state the following lemma.

**Lemma 21.** *Let $\mathcal{S}_n = \{(\nu_t, x_t, y_t)\}_{t=1}^n$ and $\bar{\mathcal{S}}_n = \{(\mu_0^{(i^*)}, x_t, y_t)\}_{t=1}^n$ be datasets sampled as in Definition 1. Then, for any estimator $\hat{F} : \mathcal{S}_n \mapsto \hat{F}$, there exists an estimator $\tilde{F}_1 : \bar{\mathcal{S}}_n \mapsto \tilde{F}_1$ such that*

$$\mathbb{E}_{\mathcal{S}_n}\big[\|\hat{F}(\nu, x) - F^\star(\nu, x)\|_{L^2(\mathbb{P}_{\nu, x_q})}^2\big] \geq \mathbb{E}_{\bar{\mathcal{S}}_n}\big[\|\tilde{F}_1(\mu_0, x) - \tilde{F}^\star(\mu_0, x)\|_{L^2(\mathbb{P}_{\mu_0, x_q})}^2\big].$$

*Moreover, if $\tilde{F}^\star$ is independent of the query $x$, i.e. $\tilde{F}^\star(\mu, x) = \tilde{F}_2^\star(\mu)$, then there exists an estimator $\tilde{F}_2$ depending only on $\{(\mu_0^{(i^*)}, y_t)\}_{t=1}^n$ such that*

$$\mathbb{E}_{\mathcal{S}_n}\big[\|\hat{F}(\nu, x) - \tilde{F}_2^\star(\mu_0)\|_{L^2}^2\big] \geq \mathbb{E}_{\{(\mu_0^{(i^*)}, y_t)\}_t}\big[\|\tilde{F}_2(\mu_0) - \tilde{F}_2^\star(\mu_0)\|_{L^2}^2\big].$$

*Proof.* Remember that

$$\nu = \frac{1}{I} \sum_i \mu_{v^{(i)}}^{(i)} = \frac{1}{I} \sum_i (\mathrm{Emb}_{v^{(i)}})_\sharp \mu_0^{(i)}, \quad \mu_0^{(i)} \overset{\text{i.i.d.}}{\sim} \mathbb{P}_{\mu_0}, \quad (v^{(i)})_{i=1}^I \sim \mathbb{P}_v.$$

We want to eliminate the dependency of $\mu_0^{(i)}$ and $v^{(i)}$ for $i \in [1:I] \setminus \{i^*\}$ from the original estimator and make a better estimate. We will construct a Bayes-estimated mapping

$$\bar{\mathcal{S}}_n \longmapsto \{(\mu_0^{(i^*)}, x_q) \mapsto \mathbb{E}_{\nu, \mathcal{T}_n}[\hat{F}_{\mathcal{S}_n}(\nu, x_q) \mid \mu_0^{(i^*)}, x_q, \bar{\mathcal{S}}_n] =: \tilde{F}_1(\mu_0^{(i^*)}, x_q)\} \in L^2(\mathbb{P}_{\mu_0, x_q}),$$

where $\mathcal{T}_n := \{(\mu_0^{(i)})_t, (v^{(i)})_t \mid \text{ for } \forall i \in [1:I] \setminus \{i^*\} \text{ and } \forall t \in [1, n]\}$ and $\bar{\mathcal{S}}_n := ((\mu_0^{(i^*)})_t, (x_q)_t, y_t)_{t=1}^n$. This estimator $\bar{\mathcal{S}}_n \mapsto \tilde{F}_1$ is well-defined as the mapping from $\bar{\mathcal{S}}_n$ to a

function in $L^2(\mathbb{P}_{\mu_0, x_{\mathrm{q}}})$ because (i) we can deterministically construct $\mathcal{S}_n$ from $\bar{\mathcal{S}}_n$ and $\mathcal{T}_n$ because $\nu = \frac{1}{I}\sum_i \mu_{v^{(i)}}^{(i)} = \frac{1}{I}\sum_i (\mathrm{Emb}_{v^{(i)}})_\sharp \mu_0^{(i)}$ and (ii) $\nu$ only has the randomness of the noises $\{(\mu_0^{(i)}, v^{(i)})\}_{i \neq i^*}$, given $(\mu_0^{(i^*)}, x_{\mathrm{q}})$. Note that the above estimator does not use the oracle of sampling an input/output pair. In short, $\tilde{F}_1$ is not cheating in the context of the standard Gaussian regression: To take the expectation with respect to $\nu | \mu_0^{(i^*)}$ and $\mathcal{T}_n$, we are additionally observing only the *noises* of the input $\nu$, which do not exist in the standard Gaussian regression. From now, we will explicitly write $\hat{F} = \hat{F}_{\mathcal{S}_n}$. The loss is lower bounded as

$$\mathbb{E}_{\mathcal{S}_n}[\|\hat{F}_{\mathcal{S}_n}(\nu, x_{\mathrm{q}}) - F^\star(\nu, x_{\mathrm{q}})\|^2_{L^2(\mathbb{P}_{\nu, x_{\mathrm{q}}})}]$$

$$=\mathbb{E}_{\mathcal{S}_n}[\|(\hat{F}_{\mathcal{S}_n}(\nu, x_{\mathrm{q}}) - \mathbb{E}_{\nu, \mathcal{T}_n}[\hat{F}_{\mathcal{S}_n} \mid \mu_0^{(i^*)}, x_{\mathrm{q}}, \bar{\mathcal{S}}_n])$$
$$+ (\mathbb{E}_{\nu, \mathcal{T}_n}[\hat{F}_{\mathcal{S}_n} \mid \mu_0^{(i^*)}, x_{\mathrm{q}}, \bar{\mathcal{S}}_n] - \tilde{F}^\star(\mu_0^{(i^*)}, x_{\mathrm{q}}))\|^2_{L^2(\mathbb{P}_{\nu, x_{\mathrm{q}}})}]$$

$$=\mathbb{E}_{\mathcal{S}_n}[\|\hat{F}_{\mathcal{S}_n}(\nu, x_{\mathrm{q}}) - \mathbb{E}_{\nu, \mathcal{T}_n}[\hat{F}_{\mathcal{S}_n} \mid \mu_0^{(i^*)}, x_{\mathrm{q}}, \bar{\mathcal{S}}_n]\|^2_{L^2(\mathbb{P}_{\nu, x_{\mathrm{q}}})}] \quad \ldots (i)$$

$$+ 2\mathbb{E}_{\mathcal{S}_n}[\langle \hat{F}_{\mathcal{S}_n}(\nu, x_{\mathrm{q}}) - \mathbb{E}_{\nu, \mathcal{T}_n}[\hat{F}_{\mathcal{S}_n} \mid \mu_0^{(i^*)}, x_{\mathrm{q}}, \bar{\mathcal{S}}_n],$$
$$\mathbb{E}_{\nu, \mathcal{T}_n}[\hat{F}_{\mathcal{S}_n} \mid \mu_0^{(i^*)}, x_{\mathrm{q}}, \bar{\mathcal{S}}_n] - \tilde{F}^\star(\mu_0^{(i^*)}, x_{\mathrm{q}})\rangle_{L^2(\mathbb{P}_{\nu, x_{\mathrm{q}}})}] \quad \ldots (ii)$$

$$+ \mathbb{E}_{\mathcal{S}_n}[\|(\mathbb{E}_{\nu, \mathcal{T}_n}[\hat{F}_{\mathcal{S}_n} \mid \mu_0^{(i^*)}, x_{\mathrm{q}}, \bar{\mathcal{S}}_n] - \tilde{F}^\star(\mu_0^{(i^*)}, x_{\mathrm{q}}))\|^2_{L^2(\mathbb{P}_{\nu, x_{\mathrm{q}}})}]$$

$$\geq \mathbb{E}_{\mathcal{S}_n}[\|(\tilde{F}_{1, \bar{\mathcal{S}}_n}(\mu_0, x_{\mathrm{q}}) - \tilde{F}^\star(\mu_0, x_{\mathrm{q}}))\|^2_{L^2(\mathbb{P}_{\mu_0, x_{\mathrm{q}}})}].$$

For the term (i), this is greater than zero. As for (ii), we have

$$(ii) = \mathbb{E}_{\mathcal{S}_n}[\langle \hat{F}_{\mathcal{S}_n}(\nu, x_{\mathrm{q}}) - \mathbb{E}_{\nu, \mathcal{T}_n}[\hat{F}_{\mathcal{S}_n} \mid \mu_0^{(i^*)}, x_{\mathrm{q}}, \bar{\mathcal{S}}_n],$$
$$\mathbb{E}_{\nu, \mathcal{T}_n}[\hat{F}_{\mathcal{S}_n} \mid \mu_0^{(i^*)}, x_{\mathrm{q}}, \bar{\mathcal{S}}_n] - \tilde{F}^\star(\mu_0^{(i^*)}, x_{\mathrm{q}})\rangle_{L^2(\mathbb{P}_{\nu, x_{\mathrm{q}}})}]$$

$$=\mathbb{E}_{\bar{\mathcal{S}}_n}[\mathbb{E}_{\mathcal{T}_n}[\langle \hat{F}_{\mathcal{S}_n}(\nu, x_{\mathrm{q}}) - \mathbb{E}_{\nu, \mathcal{T}_n}[\hat{F}_{\mathcal{S}_n} \mid \mu_0^{(i^*)}, x_{\mathrm{q}}, \bar{\mathcal{S}}_n],$$
$$\mathbb{E}_{\nu, \mathcal{T}_n}[\hat{F}_{\mathcal{S}_n} \mid \mu_0^{(i^*)}, x_{\mathrm{q}}, \bar{\mathcal{S}}_n] - \tilde{F}^\star(\mu_0^{(i^*)}, x_{\mathrm{q}})\rangle_{L^2(\mathbb{P}_{\nu, x_{\mathrm{q}}})} \mid \bar{\mathcal{S}}_n]]$$

$$=\mathbb{E}_{\bar{\mathcal{S}}_n}[\langle \mathbb{E}_{\mathcal{T}_n}[\hat{F}_{\mathcal{S}_n}(\nu, x_{\mathrm{q}}) \mid \nu, x_{\mathrm{q}}, \bar{\mathcal{S}}_n] - \mathbb{E}_{\nu, \mathcal{T}_n}[\hat{F}_{\mathcal{S}_n} \mid \mu_0^{(i^*)}, x_{\mathrm{q}}, \bar{\mathcal{S}}_n],$$
$$\mathbb{E}_{\nu, \mathcal{T}_n}[\hat{F}_{\mathcal{S}_n} \mid \mu_0^{(i^*)}, x_{\mathrm{q}}, \bar{\mathcal{S}}_n] - \tilde{F}^\star(\mu_0^{(i^*)}, x_{\mathrm{q}})\rangle_{L^2(\mathbb{P}_{\nu, x_{\mathrm{q}}})}]$$

$$(\mathbb{E}_{\nu, \mathcal{T}_n}[\hat{F}_{\mathcal{S}_n} \mid \mu_0^{(i^*)}, x_{\mathrm{q}}, \bar{\mathcal{S}}_n] \text{ and } \tilde{F}^\star(\mu_0^{(i^*)}, x_{\mathrm{q}}) \text{ are independent of } \mathcal{T}_n \mid \bar{\mathcal{S}}_n \text{ if } \bar{\mathcal{S}}_n \text{ is given})$$

$$=\mathbb{E}_{\bar{\mathcal{S}}_n}\left[\int \left\{ \left(\int (\mathbb{E}_{\mathcal{T}_n}[\hat{F}_{\mathcal{S}_n}(\nu, x_{\mathrm{q}}) \mid \nu, x_{\mathrm{q}}, \bar{\mathcal{S}}_n] - \mathbb{E}_{\nu, \mathcal{T}_n}[\hat{F}_{\mathcal{S}_n} \mid \mu_0^{(i^*)}, x_{\mathrm{q}}, \bar{\mathcal{S}}_n]) \mathrm{d}\mathbb{P}_\nu(\nu) \right) \right.\right.$$

$$\left.\left. \times (\mathbb{E}_{\nu, \mathcal{T}_n}[\hat{F}_{\mathcal{S}_n} \mid \mu_0^{(i^*)}, x_{\mathrm{q}}, \bar{\mathcal{S}}_n] - \tilde{F}^\star(\mu_0^{(i^*)}, x_{\mathrm{q}})) \right\} \mathrm{d}\mathbb{P}_{x_{\mathrm{q}}}(x_{\mathrm{q}}) \right]$$

$$=\mathbb{E}_{\bar{\mathcal{S}}_n}\left[\int \left\{ \left(\int (\mathbb{E}_{\mathcal{T}_n}[\hat{F}_{\mathcal{S}_n}(\nu, x_{\mathrm{q}}) \mid \nu, x_{\mathrm{q}}, \bar{\mathcal{S}}_n] - \mathbb{E}_{\nu, \mathcal{T}_n}[\hat{F}_{\mathcal{S}_n} \mid \mu_0^{(i^*)}, x_{\mathrm{q}}, \bar{\mathcal{S}}_n]) \mathrm{d}\mathbb{P}_{\nu | \mu_0^{(i^*)}}(\nu) \mathrm{d}\mathbb{P}_{\mu_0^{(i^*)}}(\mu_0^{(i^*)}) \right) \right.\right.$$

$$\left.\left. \times (\mathbb{E}_{\nu, \mathcal{T}_n}[\hat{F}_{\mathcal{S}_n} \mid \mu_0^{(i^*)}, x_{\mathrm{q}}, \bar{\mathcal{S}}_n] - \tilde{F}^\star(\mu_0^{(i^*)}, x_{\mathrm{q}})) \right\} \mathrm{d}\mathbb{P}_{x_{\mathrm{q}}}(x_{\mathrm{q}}) \right]$$

$$=0.$$

In the same vein, we can omit the dependence of $x_{\mathrm{q}}$ if $F^\star$ is independent of $x_{\mathrm{q}}$. We give an estimated mapping as

$$\mu_0 \mapsto \mathbb{E}_{x_{\mathrm{q}}, ((x_{\mathrm{q}})_t)_{t=1}^n}[\tilde{F}_{1, \mathcal{S}_n} \mid \mu_0, ((\mu_0^{(i^*)})_t, y_t)_t]$$

that can be constructed only with the observation $((\mu_0^{(i^*)})_t, y_t)_t$. We omit the details for the second statement. $\square$

## C.3 STEP 2-1: UPPER-BOUND OF THE ENTROPY

Thanks to Corollary 4, the lower-bound analysis reduces to a Gaussian regression problem in which the inputs $\mu_0^{(i^*)} \sim \mathbb{P}_{\mu_0}$ are generated according to Setting 4. The key step is to control the cov-

ering/packing numbers of the underlying function class so that we can apply the general minimax bound of Yang & Barron (1999).

**Lemma 22** (Yang & Barron (1999)). *Let $\tilde{\mathcal{F}}^\star := \mathrm{Lip}_L(B(\mathcal{H}_0, \|\cdot\|_{\mathcal{H}_0^{\gamma_b}}), \|\cdot\|_{\mathcal{H}_0^{\gamma_f}})$. Consider the dataset $\mathcal{U}_n = \{(\mu_0^{(i^*)}, y_t)\}_{t=1}^n$ generated as in Setting 4. Suppose there exist $\delta, \epsilon > 0$ such that*

$$\log_2 \mathcal{N}(\mathcal{F}^\star; \epsilon)_{L^2} \leq \frac{n\epsilon^2}{2\sigma^2}, \qquad \log_2 \mathcal{M}(\mathcal{F}^\star; \delta)_{L^2} \geq \frac{2n\epsilon^2}{\sigma^2} + 2\log 2.$$

*Then*

$$\inf_{\mathcal{U}_n \mapsto \hat{F}} \sup_{F^\star \in \tilde{\mathcal{F}}^\star} \mathbb{E}_{\mathcal{U}_n}\left[\|\hat{F} - F^\star\|_{L^2(\mathbb{P}_{\mu_0})}^2\right] \gtrsim \delta^2.$$

**Remark 7.** For lower-bound analysis, we adopt the $L^2$ norm when defining covering and packing numbers.

**Goal of this subsection.** To apply Lemma 22, we need tight control of the covering entropy of the Lipschitz function class. Specifically, we aim to establish an upper bound on

$$\log \mathcal{N}\left(\mathrm{Lip}_1(B(\mathcal{H}_0, \|\cdot\|_{\mathcal{H}_0^{\gamma_b}}), \|\cdot\|_{\mathcal{H}_0^{\gamma_f}}); \epsilon\right)_{L^2(\mathbb{P}_{\mu_0})}.$$

A lower bound is deferred to Appendix C.4.

By proposition B.2 in Boissard (2011), the covering entropy of a (1-)Lipschitz class can be controlled by the covering entropy of its input set:

$$\log \mathcal{N}(\mathrm{Lip}(A, d); \epsilon)_{L^\infty} \lesssim \mathcal{N}(A; \epsilon)_d \cdot \mathrm{polylog}(\epsilon^{-1}),$$

where $A$ is the input domain (under some weak assumptions) and d is the underlying metric. Thus, our task reduces to bounding the entropy of the input set $A = B(\mathcal{H}_0, \|\cdot\|_{\mathcal{H}_0^{\gamma_b}})$ equipped with the metric $\|\cdot\|_{\mathcal{H}_0^{\gamma_f}}$.

To this end, we establish an isometric correspondence between balls in weighted Hilbert spaces, which allows us to switch to the standard $L^2$ metric.

**Lemma 23.** *Let $a, b, c \in \mathbb{R}$. A mapping $\phi_{a,b,c}$:*

$$f = \sum_{i=1}^\infty b_i e_i \mapsto \phi(f) = \sum_{i=1}^\infty \lambda_i^{\frac{c-b}{2}} b_i e_i$$

*is an isometric bijection from $(B(\mathcal{H}_0, \|\|_{\mathcal{H}_0^a}), \|\|_{\mathcal{H}_0^b})$ to $(B(\mathcal{H}_0, \|\|_{\mathcal{H}_0^{a-b+c}}), \|\|_{\mathcal{H}_0^c})$.*

*Proof.* First, for all $f \in B(\mathcal{H}_0, \|\|_{\mathcal{H}_0})$,

$$\|\phi(f)\|_{\mathcal{H}_0^c} = \sum_i \lambda_i^{-c}\left(\lambda_i^{\frac{c-b}{2}} b_i\right)^2 = \sum_i \lambda_i^{-c+c-b} b_i^2 = \|f\|_{\mathcal{H}_0^b}.$$

This implies that $\phi$ is isometry and injective. Next, $\phi$ is also a surjection because

$$f = \sum_{i=1}^\infty b_i e_i \in B(\mathcal{H}_0, \|\|_{\mathcal{H}_0^a})$$

$$\Leftrightarrow \sum_i \lambda_i^{-a} b_i^2 \leq 1$$

$$\Leftrightarrow \sum_i \lambda_i^{-a+b-c}(\lambda_i^{\frac{c-b}{2}} b_i)^2 \leq 1$$

$$\Leftrightarrow \phi(f) = \sum_{i=1}^\infty \lambda_i^{\frac{c-b}{2}} b_i e_i \in B(\mathcal{H}_0, \|\|_{\mathcal{H}_0^{a-b+c}})$$

$\square$

Using the above isometry and eigenvalue decay properties, we obtain the following upper bound on the entropy of the input set:

**Lemma 24.** *The covering entropy of $B(\mathcal{H}_0, \|\|\|_{\mathcal{H}_0^{\gamma_b}})$ endowed with the distance $\|\|\|_{\gamma_f}$ is upper bounded as*

$$\log \mathcal{N}(B(\mathcal{H}_0, \| \cdot \|_{\mathcal{H}_0^{\gamma_b}}); \epsilon)_{\|\|\|_{\mathcal{H}_0^{\gamma_f}}} \lesssim \ln\left(\epsilon^{-1}\right)^{\frac{\alpha+1}{\alpha}}.$$

*Proof.* By Lemma 23, letting $a = \gamma_b$, $b = \gamma_f$, $c = 0$, it is sufficient to show that

$$\log \mathcal{N}(B_\phi; \epsilon)_{L^2} \lesssim \ln\left(\epsilon^{-1}\right)^{\frac{\alpha+1}{\alpha}}$$

where $B_\phi = B(\mathcal{H}_0, \|\|\|_{\mathcal{H}_0^{\gamma_b - \gamma_f}})$. We construct a $J_\epsilon$-dimensional set $\mathcal{E}_{J_\epsilon} = \{(b_1, \ldots, b_{J_\epsilon}) \mid f = \sum_i b_i e_i \in B_\phi\}$ such that, for all $f = \sum_i b_i e_i \in B_\phi$,

$$\sum_{j \geq J_\epsilon + 1} b_i^2 < \frac{1}{4}\epsilon^2.$$

This can be satisfied if $J_\epsilon \simeq (c^{-1}(\gamma_b - \gamma_f) \ln \epsilon^{-1})^{1/\alpha}$ because $\sum_{j \geq J_\epsilon + 1} b_i^2 \leq \sum_{j \geq J_\epsilon + 1} \lambda_j \lesssim \lambda_{J_\epsilon}$ with exponential decay and use $\lambda_j \simeq \exp(-cj^\alpha)$. To construct a $\frac{1}{2}\epsilon$-covering on $\mathcal{E}_{J_\epsilon}$, we need at most $O(1 \vee (\lambda_j^{\frac{\gamma_b - \gamma_f}{2}}/\epsilon))$ patterns for each dimension, so the covering entropy is bounded as

$$\begin{aligned}
\log \mathcal{N}(\epsilon) &\lesssim \sum_{j=1}^{J_\epsilon} \log \frac{\lambda_j^{\frac{\gamma_b - \gamma_f}{2}}}{\epsilon} \\
&\lesssim \sum_{j=1}^{J_\epsilon} \left(-\left(\frac{\gamma_b - \gamma_f}{c}\right) j^\alpha + \left(\frac{\gamma_b - \gamma_f}{c}\right)^{-1} J_\epsilon^\alpha\right) \\
&\lesssim \left(\frac{\gamma_b - \gamma_f}{c}\right)^{-1} J_\epsilon^{\alpha+1} \\
&\simeq \left(\frac{\gamma_b - \gamma_f}{c}\right)^{-\alpha^{-1}} (\ln \epsilon^{-1})^{(\alpha+1)/\alpha}
\end{aligned}$$

$\square$

Finally, results in Boissard (2011) and Lemma 24 yield the desired entropy bound for the Lipschitz class.

**Lemma 25** (Based on Boissard (2011)). *The metric entropy of $\mathrm{Lip}_1(B(\mathcal{H}_0, \| \cdot \|_{\mathcal{H}_0^{\gamma_b}})); \| \cdot \|_{\mathcal{H}_0^{\gamma_f}})$ with respect to the Bochner $L^2(\mathbb{P}_{\mu_0})$-norm satisfies*

$$\log \mathcal{N}(\mathrm{Lip}_1(B(\mathcal{H}_0, \| \cdot \|_{\mathcal{H}_0^{\gamma_b}}), \| \cdot \|_{\mathcal{H}_0^{\gamma_f}}); \epsilon)_{\|\|\|_{L^2(\mathbb{P}_{\mu_0})}} \lesssim \exp\left(c_u (\ln \epsilon^{-1})^{\frac{\alpha+1}{\alpha}}\right)$$

*for some $c_u > 0$.*

*Proof.* By Lemma 24, $\log \mathcal{N}(B(\mathcal{H}_0, \| \cdot \|_{\mathcal{H}_0^{\gamma_b}}); \epsilon)_{\| \cdot \|_{\mathcal{H}_0^{\gamma_f}}} \lesssim \ln\left(\epsilon^{-1}\right)^{\frac{\alpha+1}{\alpha}}$. Then, by Proposition B.2 in Boissard (2011) and $\epsilon^{-1} = \exp\left(\ln \epsilon^{-1}\right) = o(\exp\left((\ln \epsilon^{-1})^{\frac{\alpha+1}{\alpha}}\right))$ for any $\alpha > 0$, we have

$$\log \mathcal{N}(\mathrm{Lip}_1(B(\mathcal{H}_0, \| \cdot \|_{\mathcal{H}_0^{\gamma_b}}), \| \cdot \|_{\mathcal{H}_0^{\gamma_f}}); \epsilon)_{\|\|\|_\infty} \lesssim \exp\left(c_u (\ln \epsilon^{-1})^{\frac{\alpha+1}{\alpha}}\right).$$

By the inequality $\sqrt{\mathbb{P}_{\mu_0}(B(\mathcal{H}_0, \| \cdot \|_{\mathcal{H}_0^{\gamma_b}}))}\| \cdot \|_\infty \geq \| \cdot \|_{L^2(\mathbb{P}_{\mu_0})}$ and $\sqrt{\mathbb{P}_{\mu_0}(B(\mathcal{H}_0, \| \cdot \|_{\mathcal{H}_0^{\gamma_b}}))} = 1$, we obtain the desired bound. $\square$

## C.4 Step 2-2: Lower Bound of the Entropy

To apply the general minimax bound in Lemma 22, We will provide the lower bound of the infinite-dimensional lipschitz class.

The main difficulty lies in the anisotropic nature of the Lipschitz constant: for $F$ in our class, one has

$$|F(\sum_j b_j e_j) - F(\sum_j c_j e_j)| \leq L \sqrt{\sum_j \lambda_j^{-\gamma_f}(b_j - c_j)^2}, \qquad -\gamma_f > 0,$$

which implies that the functional is significantly smoother in directions corresponding to high-index coefficients $e_j$. To capture this effect, we construct the following embedding:

$$\iota_d \,:\, L^p([0,1]^d) \hookrightarrow L^p(\mu_{\mathcal{H}_0}), \qquad (\iota_d g)(f_\mu) := g\left(\Phi_1\left(\frac{f_{\mu,1}}{\lambda_1^{\gamma_d/2}}\right), \ldots, \Phi_d\left(\frac{f_{\mu,d}}{\lambda_d^{\gamma_d/2}}\right)\right),$$

where

$$f_\mu = \sum_{j=1}^\infty \lambda_j^{\gamma_d/2} Z_j e_j, \quad d\mu(x) = f_\mu(x)\,dx,$$

with independent coefficients $Z_j \sim \rho_j$ and cumulative distribution functions $\Phi_j(z) = \int_{-\infty}^z \rho_j(u)du$.

We prove that

$$\iota_d\big(\mathrm{Lip}_1([0,1]^d)\big) \subset \mathrm{Lip}_{R/\lambda_d^{(-\gamma_f+\gamma_d)/2}}\big(B(\mathcal{H}_0, \|\cdot\|_{\mathcal{H}_0^{\gamma_b}}), \|\cdot\|_{\mathcal{H}_0^{\gamma_f}}\big),$$

and hence obtain a packing lower bound

$$\log \mathcal{M}\big(\mathrm{Lip}_1(B(\mathcal{H}_0, \|\cdot\|_{\mathcal{H}_0^{\gamma_b}}), \|\cdot\|_{\mathcal{H}_0^{\gamma_f}}); \epsilon\big)_{L^p} \geq \log \mathcal{M}\left(\mathrm{Lip}_1([0,1]^d, \|\cdot\|_{\ell^\infty}), \frac{R\epsilon}{\lambda_d^{(-\gamma_f+\gamma_d)/2}}\right)_{L^p}.$$

Combining this rescaling argument with standard Lipschitz-packing lower bounds and the information-theoretic results of Yang–Barron yields the desired minimax lower bound for the associative recall problem.

First, we construct an embedding $\iota_d$. This construction suggests that, after a suitable rescaling of coordinates, functions on $[0,1]^d$ can be embedded isometrically into our measure-input space. The next lemma formalizes this embedding and quantifies how the Lipschitz constant is rescaled.

**Lemma 26** (An extension of Lanthaler (2024)). *Let $\mathbb{P}_{\mu_0}$ be a probability measure on $B(\mathcal{H}_0, \|\cdot\|_{\mathcal{H}_0^{\gamma_b}})$ in Setting 4. For any $p \in [1,\infty)$ and $d \in \mathbb{N}$, there exists an isometric embedding*

$$\iota_d \,:\, L^p([0,1]^d) \hookrightarrow L^p(\mathbb{P}_{\mu_0}),$$

*such that $\iota_d(\mathrm{Lip}_1([0,1]^d; \|\|_\infty)) \subset \mathrm{Lip}_{R/(\lambda_d^{(-\gamma_f+\gamma_d)/2})}(B(\mathcal{H}_0, \|\cdot\|_{\mathcal{H}_0^{\gamma_b}}), \|\cdot\|_{\mathcal{H}_0^{\gamma_f}}))$, where the Lipschitz norm on $[0,1]^d$ is defined with respect to the $\infty$-norm on $[0,1]^d$.*

*Proof.* By Assumption 8, $\mu \sim \mathbb{P}_{\mu_0}$ is sampled as

$$f_\mu = \sum_j \lambda_j^{\frac{\gamma_d}{2}} Z_j e_j, \quad \frac{d\mu}{d\lambda} = f_\mu,$$

where $Z_j, j \geq 1$ are independent and $Z_j \sim \rho_j(z)dz$. We define the cumulative distribution $\Phi_j(z) = \int_{-\infty}^z \rho_j dz$. We know that $F_j$ is Lipschitz, whose Lipschitz constant is bounded by $\sup_j \|\rho_j\|_\infty \leq R$. We define $f_{\mu,i} = \langle f_\mu, e_i \rangle$. We will show that the mapping

$$\iota_d \,:\, L^p([0,1]^d) \hookrightarrow L^p(\mathbb{P}_{\mu_0}), \quad (\iota_d g)(f_\mu) = g(\Phi_1(f_{\mu,1}/\lambda_1^{\frac{\gamma_d}{2}}), \ldots, \Phi_d(f_{\mu,d}/\lambda_d^{\frac{\gamma_d}{2}}))$$

is the isometric embedding that we want. For $g \in L^p([0,1]^d)$, the $L^p(\mathbb{P}_{\mu_0})$-norm of $\iota_d g$ is equal to

$$\mathbb{E}_{f_\mu}|(\iota_d g)(f_\mu)|^p = \mathbb{E}_{f_\mu}|g(\Phi_1(f_{\mu,1}/\lambda_1^{\frac{\gamma_d}{2}}), \ldots, \Phi_d(f_{\mu,d}/\lambda_d^{\frac{\gamma_d}{2}}))|^p$$

$$= \mathbb{E}_{f_\mu} |g(\Phi_1(Z_1), \ldots, \Phi_d(Z_d))|^p$$

$$= \int_{[0,1]^d} |g(x_1, \ldots, x_d)|^p \mathrm{d}x \quad (\Phi_j(Z_j) \sim \mathrm{Unif}[0,1])$$

$$= \|g\|_{L^p([0,1]^d)}^p,$$

which shows that $\iota_d$ is isometric embedding. Next, we evaluate the image of $\iota_d$. A mapping

$$h_d : (B(\mathcal{H}_0, \|\cdot\|_{\mathcal{H}_0^{\gamma_b}}), \|\cdot\|_{\mathcal{H}_0^{\gamma_f}}) \to ([0,1]^d, \|\|_\infty), \quad f_\mu \mapsto (\Phi_1(f_{\mu,1}/\lambda_1^{\frac{\gamma_d}{2}}), \ldots, \Phi_d(f_{\mu,d}/\lambda_d^{\frac{\gamma_d}{2}}))$$

is lipschitz because

$$\|h_d(f_{\mu_1}) - h_d(f_{\mu_2})\|_\infty$$

$$\leq \max_j \left\{ \left| \Phi_j(f_{\mu_1,j}/\lambda_j^{\frac{\gamma_d}{2}}) - \Phi_j(f_{\mu_2,j}/\lambda_j^{\frac{\gamma_d}{2}}) \right| \right\}$$

$$\leq \max_j \left\{ \mathrm{Lip}(\Phi_j) \lambda_j^{\frac{-\gamma_d}{2}} |f_{\mu_1,j} - f_{\mu_2,j}| \right\}$$

$$\leq \max_j \left\{ \mathrm{Lip}(\Phi_j) \lambda_j^{\frac{\gamma_f - \gamma_d}{2}} \cdot \lambda_j^{\frac{-\gamma_f}{2}} |f_{\mu_1,j} - f_{\mu_2,j}| \right\}$$

$$\leq \max_j \left\{ \mathrm{Lip}(\Phi_j) \lambda_j^{\frac{\gamma_f - \gamma_d}{2}} \right\} \sum_j (\lambda_j^{-\frac{\gamma_f}{2}} |f_{\mu_1,j} - f_{\mu_2,j}|)$$

and thus

$$\mathrm{Lip}(h_d) \leq \frac{R}{\lambda_d^{\frac{-\gamma_f + \gamma_d}{2}}}.$$

Therefore, $\forall g \in \mathrm{Lip}_1([0,1]^d, \infty)$, Lipschitz constant of $\iota_d g$ is bounded as

$$\mathrm{Lip}(\iota_d g) = \mathrm{Lip}(g \circ h_d) \leq \mathrm{Lip}(g) \cdot \mathrm{Lip}(h_d) \leq \frac{R}{\lambda_d^{\frac{-\gamma_f + \gamma_d}{2}}}.$$

Furthermore, we also have $\|\iota_d g\|_{C(B(\mathcal{H}_0, \|\|_{\mathcal{H}_0^{\gamma_b}})} \leq 1$. $\qquad\square$

Lemma 26 ensures that the Lipschitz class on $[0,1]^d$ can be viewed as a subclass of our infinite-dimensional Lipschitz class, up to a scaling factor depending on $\lambda_d$. This immediately yields a lower bound on the packing numbers of our class in terms of the well-studied packing numbers of $\mathrm{Lip}_1([0,1]^d)$.

**Corollary 5.** *Under the assumptions of Lemma 26, we have*

$$\log \mathcal{M}(\mathrm{Lip}_1(B(\mathcal{H}_0, \|\cdot\|_{\mathcal{H}_0^{\gamma_b}}), \|\cdot\|_{\mathcal{H}_0^{\gamma_f}}); \epsilon)_{L^p(\mathbb{P}_{\mu_0})} \gtrsim \log \mathcal{M}\left( \mathrm{Lip}_1([0,1]^d, \|\|_\infty), \frac{R\epsilon}{\lambda_d^{\frac{-\gamma_f + \gamma_d}{2}}} \right)_{L^p([0,1]^d)}$$

*Proof.* By rescaling the function, we have

$$\mathcal{M}(\mathrm{Lip}_1(B(\mathcal{H}_0, \|\cdot\|_{\mathcal{H}_0^{\gamma_b}}), \|\cdot\|_{\mathcal{H}_0^{\gamma_f}}); \epsilon)_{L^p}$$

$$= \mathcal{M}\left( \mathrm{Lip}_{R/(\lambda_d^{(-\gamma_f + \gamma_d)/2})}(B(\mathcal{H}_0, \|\cdot\|_{\mathcal{H}_0^{\gamma_b}}), \|\cdot\|_{\mathcal{H}_0^{\gamma_f}}); \frac{R\epsilon}{(\lambda_d^{(-\gamma_f + \gamma_d)/2})} \right)_{L^p}$$

$$\gtrsim \mathcal{M}\left( \iota_d(\mathrm{Lip}_1([0,1]^d; \ell^\infty)); \frac{R\epsilon}{(\lambda_d^{(-\gamma_f + \gamma_d)/2})} \right)_{L^p}$$

$$\gtrsim \mathcal{M}\left( \mathrm{Lip}_1([0,1]^d; \ell^\infty); \frac{R\epsilon}{(\lambda_d^{(-\gamma_f + \gamma_d)/2})} \right)_{L^p([0,1]^d)}.$$

$\qquad\square$

Thus, the problem of estimating the packing entropy of the infinite-dimensional class reduces to that of estimating the entropy of finite-dimensional Lipschitz functions on the cube. Fortunately, sharp lower bounds for the latter are available in the literature. Note that packing and covering are almost equivalent when $\epsilon \to 0$.

**Lemma 27** (Lanthaler (2024))**.** *For $p \in [1, \infty)$ and $d \in \mathbb{N}$, there exists a constant $c > 0$ independent of $d$ such that*

$$\log \mathcal{M}(\mathrm{Lip}_1([0,1]^d, \|\|_\infty); \epsilon)_{\|\|_{L^p}} \gtrsim \left(\frac{c}{d\epsilon}\right)^d, \quad \forall \epsilon \in (0, c/d].$$

Combining the embedding argument with the finite-dimensional lower bounds above, we arrive at the following result, which provides the desired exponential lower bound on the entropy growth.

**Lemma 28** (Parallel to Lanthaler (2024))**.** *Let $\mathbb{P}_{\mu_0}$ be a probability measure on $B(\mathcal{H}_0, \| \cdot \|_{\mathcal{H}_0^{\gamma_b}})$ in Setting 4. The packing entropy of $\mathrm{Lip}_1(B(\mathcal{H}_0, \| \cdot \|_{\mathcal{H}_0^{\gamma_b}}), \| \cdot \|_{\mathcal{H}_0^{\gamma_f}})$ with respect to the Bochner $L^p(\mathbb{P}_{\mu_0})$-norm, satisfies*

$$\log \mathcal{M}(\mathrm{Lip}_1(B(\mathcal{H}_0, \| \cdot \|_{\mathcal{H}_0^{\gamma_b}}), \| \cdot \|_{\mathcal{H}_0^{\gamma_f}}), \epsilon)_{L^2(\mathbb{P}_{\mu_0})} \gtrsim \exp\left(c_l (\ln \epsilon^{-1})^{\frac{\alpha+1}{\alpha}}\right)$$

*for some constant $c_l > 0$.*

*Proof.* Combining Corollary 5 and Lemma 27,

$$\log \mathcal{M}(\mathrm{Lip}_1(B(\mathcal{H}_0, \| \cdot \|_{\mathcal{H}_0^{\gamma_b}}), \| \cdot \|_{\mathcal{H}_0^{\gamma_f}}), \epsilon)_{L^p} \gtrsim \left(\frac{c_1 \lambda_d^{\frac{-\gamma_f + \gamma_d}{2}}}{8 d_\epsilon \epsilon}\right)^d$$

where $c_1 > 0$ is a constant, provided $\epsilon \le \frac{c_1 \lambda_d^{\frac{-\gamma_f + \gamma_d}{2}}}{d}$. Then, by $\lambda_d \gtrsim \exp(-cd^\alpha)$, the RHS is lower bounded by

$$(\log \mathcal{M} \gtrsim) \left(\frac{c_1 \exp(-c'd^\alpha))}{d\epsilon}\right)^d \quad \text{if } \epsilon \le c_2 d^{-1} \exp(-c'd^\alpha),$$

where $c' = \frac{c(-\gamma_f + \gamma_d)}{2}$ and $c_2 > 0$ is another constant. Assuming that $\epsilon$ is sufficiently small, let us take $d$ as

$$d = \left(\frac{\log(c_2 \epsilon^{-1})}{c' + 1}\right)^{1/\alpha} \quad (\simeq (\ln \epsilon^{-1})^{1/\alpha}).$$

By rearranging the above inequality, we also obtain

$$\epsilon = c_2 \exp(-(c'+1)d^\alpha),$$

which can satisfy $\epsilon \le c_2 d^{-1} \exp(-c'd^\alpha)$ asymptotically, because $\exp(-d^\alpha) \ll d^{-1}$ where $d \simeq (\ln \epsilon^{-1})^{1/\alpha} \to \infty$ as $\epsilon \to 0$. Then, we have

$$\log \mathcal{M}(\mathrm{Lip}_1(B(\mathcal{H}_0, \| \cdot \|_{\mathcal{H}_0^{\gamma_b}}), \| \cdot \|_{\mathcal{H}_0^{\gamma_f}}), \epsilon)_{L^p}$$
$$\gtrsim \left(\frac{c_1 \exp(-c'd^\alpha))}{d\epsilon}\right)^d$$
$$\gtrsim (\exp(-c'd^\alpha + (c'+1)d^\alpha - \ln d + O(1)))^d$$
$$\gtrsim \exp(\Omega(d^{\alpha+1}))$$
$$\gtrsim \exp\left(\Omega((\ln \epsilon^{-1})^{\frac{\alpha+1}{\alpha}})\right),$$

where we used $d^\alpha \simeq \ln \epsilon^{-1}$ in the fourth inequality. $\qquad\square$

## C.5 PROOF OF MINIMAX LOWER BOUND

**Theorem 7** (Minimax Lower Bound)**.** *Under Setting 4, we have*

$$\inf_{\mathcal{S}_n \mapsto \hat{F}} \sup_{\tilde{F}^\star \in \mathcal{F}^\star} \mathbb{E}_{\mathcal{S}_n}\left[\|\hat{F} - F^\star\|^2_{L^2(\mathbb{P}_{\nu, x_q})}\right] \gtrsim \exp\left(-O\left((\ln n)^{\frac{\alpha}{\alpha+1}}\right)\right).$$

*Proof.* By Lemma 21, it is sufficient to evaluate

$$\inf_{\mathcal{U}_n \mapsto \hat{F} \in L^2(\mathbb{P}_{\mu_0})} \sup_{F^\star \in \tilde{\mathcal{F}}^\star} \mathbb{E}_{\mathcal{U}_n}\big[\|\hat{F}(\mu_0) - F^\star(\mu_0)\|^2_{L^2(\mathbb{P}_{\mu_0})}\big]$$

where $\mathcal{U}_n = \{(\mu_0^{(i^*)}, y_t)\}_{t=1}^n$ sampled as in Setting 4, instead of the original problem. Let $V(\epsilon_n) := \log \mathcal{N}(\mathrm{Lip}_1(B(\mathcal{H}_0, \| \cdot \|_{\mathcal{H}_0^{\gamma_b}}), \|\|_{\mathcal{H}_0^{\gamma_f}}); \epsilon)_{L^2}$ and $M(\delta_n) := \log \mathcal{M}(\mathrm{Lip}_1(B(\mathcal{H}_0, \| \cdot \|_{\mathcal{H}_0^{\gamma_b}}), \|\|_{\mathcal{H}_0^{\gamma_f}}); \epsilon)_{L^2}$.

First, let $\epsilon_n = C_{1,\sigma} \exp\big(-c_{\mathrm{E}}(\ln n)^{\frac{\alpha}{\alpha+1}}\big)$ where $c_{\mathrm{E}} = \left(\frac{1}{2c_u^2}\right)^{\frac{\alpha}{\alpha+1}}$. Then, by Lemma 25,

$$\begin{aligned}
\frac{V(\epsilon_n)}{\epsilon_n^2} &\lesssim \exp\left(c_u(\log \epsilon_n^{-1})^{\frac{\alpha+1}{\alpha}} + 2c_{\mathrm{E}}(\ln n)^{\frac{\alpha}{\alpha+1}}\right) \\
&\lesssim \exp\left(c_u(c_{\mathrm{E}}(\ln n)^{\frac{\alpha}{\alpha+1}})^{\frac{\alpha+1}{\alpha}} + 2c_{\mathrm{E}}(\ln n)^{\frac{\alpha}{\alpha+1}}\right) \\
&\lesssim \exp\left(\frac{1}{2}\ln n + O((\ln n)^{\frac{\alpha}{\alpha+1}})\right) \\
&\lesssim \frac{n}{\sigma^2}.
\end{aligned}$$

Note that $\sigma = \Theta(1)$. Next, we lower bound the packing entropy. Taking $\delta_n = C_\sigma \exp\big(-c_{\mathrm{d}}(\ln n)^{\frac{\alpha}{\alpha+1}}\big)$ where $c_{\mathrm{d}} = c_l^{-\frac{\alpha}{\alpha+1}}$ and $C_\sigma \gtrsim \frac{2}{\sigma^2} + \frac{2\log 2}{1}$ is a sufficiently large constant only dependent of $\sigma = \Theta(1)$, using Lemma 28,

$$\frac{M(\delta_n)}{\epsilon_n^2} \gtrsim \exp\left(c_l(c_{\mathrm{d}}^{\frac{\alpha+1}{\alpha}}\ln n) + O((\ln n)^{\frac{\alpha}{\alpha+1}})\right) \cdot \left(\frac{2}{\sigma^2} + \frac{2\log 2}{1}\right) \gtrsim n\left(\frac{2}{\sigma^2} + \frac{2\log 2}{n}\right).$$

Finally, applying Lemma 22, we have

$$\inf_{\mathcal{S}_n \mapsto \hat{F}} \sup_{F \in \mathcal{F}^\circ} \mathbb{E}\left[\|\hat{F} - F\|^2_{L^2}\right] \gtrsim \delta_n^2 \gtrsim \exp\left(-O((\ln n)^{\frac{\alpha}{\alpha+1}})\right).$$

$\square$

# D  SYNTHETIC EXPERIMENT ON MEASURE-VALUED ATTENTION

To provide a minimal empirical sanity check of our risk bounds, we design a simple synthetic experiment where the input is a *measure-valued context* on $[0, 1] \times \{-1, +1\}$ and the model is a single MLP→Attention→MLP block. The goal is to recover a scalar functional of an underlying "associative" measure from a mixture of associative and non-associative components.

**Data-generating process.** Fix a truncation level $M = 16 \in \mathbb{N}$ and an orthonormal trigonometric basis on $[0, 1]$,

$$\phi_0(x) = 1, \quad \phi_j(x) = \sqrt{2}\sin(\pi j x), \quad j = 1, \dots, M - 1.$$

For a smoothness parameter $\alpha > 0$ we define eigenvalues

$$\lambda_j = \exp(-j^\alpha), \qquad j = 1, \dots, M - 1.$$

For each training example we sample two sets of coefficients $Z_1, Z_2 \sim \mathcal{N}(0, I_{M-1})$ independently, $Z_{1,0}, Z_{2,0} = 0$, and form the (unnormalized) densities on $[0, 1]$

$$\tilde{\mu}_k(x) = \sum_{j=0}^{M-1} \lambda_j Z_{k,j} \phi_j(x), \qquad k \in \{1, 2\}.$$

We discretize $[0, 1]$ on a uniform grid $\{x_t\}_{t=1}^T$, $T = 32$, clamp the density to be nonnegative, and normalize to obtain a probability mass function $(p_k(t))_{t=1}^T$:

$$\mu_k^{\mathrm{raw}}(x_t) = \tilde{\mu}_k(x_t), \qquad \mu_k^+(x_t) = \max\{\mu_k^{\mathrm{raw}}(x_t), \varepsilon\}, \qquad p_k(t) = \frac{\mu_k^+(x_t)}{\sum_{s=1}^T \mu_k^+(x_s)},$$

with a small cutoff $\varepsilon > 0$ [3]

Independently, we sample a "query label" $v^{(1)} \in \{-1, +1\}$ uniformly and set $v^{(2)} := -v^{(1)}$. We then define a product-measure mixture on $[0, 1] \times \{-1, +1\}$ by

$$\nu = \tfrac{1}{2}\big(\mu_1 \otimes \delta_{v^{(1)}}\big) + \tfrac{1}{2}\big(\mu_2 \otimes \delta_{v^{(2)}}\big),$$

where $\mu_k$ is the discrete measure assigning mass $p_k(t)$ to $x_t$. To construct the input token sequence, we draw $n_{\text{tokens}} = 5000$ i.i.d. Monte Carlo samples $(X_i, Q_i) \sim \nu$:

$$(X_i, V_i) = \begin{cases} (x_{T_1}, v^{(1)}), & \text{with prob. } \tfrac{1}{2}, \ T_1 \sim p_1, \\ (x_{T_2}, v^{(2)}), & \text{with prob. } \tfrac{1}{2}, \ T_2 \sim p_2, \end{cases} \qquad i = 1, \ldots, n_{\text{tokens}}.$$

Finally, we append a single "query token" $(0, v^{(1)})$ at the end of the sequence, so that each input example is a sequence

$$\big\{(X_i, V_i)\big\}_{i=1}^{n_{\text{tokens}}} \cup \{(0, v^{(1)})\} \ \in \ \big([0, 1] \times \{-1, +1\}\big)^{n_{\text{tokens}}+1}.$$

The target output $Y$ depends only on the *associative* measure $\mu_1$ (and is independent of $\mu_2$ and $v^{(2)}$):

$$Y \ \simeq \ \tilde{F}^{\star}(\mu_1, \underbrace{X_{n_{\text{tokens}}+1}}_{\text{query token}}) \ := \ v^{(1)} \cdot \sum_{j=0}^{M-1} \lambda_j \, Z_{1,j}^2.$$

Intuitively, the model must use the final query token $(0, v^{(1)})$ to attend to tokens consistent with $q_1$ and recover information about the hidden coefficients $Z_1$ from Monte Carlo samples of $\mu_1$. Note that we add a small Gaussian noise with std $= 0.01$ in training.

**Model and training.** We use a minimal architecture that mirrors the theoretical measure-attention operator:

$$\text{context/query MLP} \ \rightarrow \ \text{measure attention} \ \rightarrow \ \text{MLP HEAD}.$$

For each example we construct a sequence of $T_{\text{ctx}}$ context tokens $(x_t, v_t) \in \mathbb{R}^2$ together with a final query token $(0, v_{\text{query}})$. The context tokens and the query token are embedded by separate two-layer MLPs into $\mathbb{R}^{d_{\text{model}}}$ with $d_{\text{model}} = 8$ and hidden width $d_{\text{hidden}} = 8$. The resulting query embedding provides the $Q$ vector, while the context embeddings provide the $K$ and $V$ vectors for a single 4-head softmax attention layer. This "measure-attention" layer outputs a single $d_{\text{model}}$-dimensional representation, which is fed into a final two-layer MLP head to produce the scalar prediction $\hat{Y}$. We train with the squared loss $\ell(\hat{Y}, Y) = (\hat{Y} - Y)^2$ using Adam with an exponentially decaying learning rate for 20 epochs.

For each $\alpha \in \{\alpha_1, \ldots, \alpha_L\}$ we generate independent training sets of sizes $n \in \{n_{\text{min}}, \ldots, n_{\text{max}}\}$ ( $n = 2^k$ for $k = 2, \ldots, 6$) and measure the empirical risk $L(n)$ on a held-out validation set ($n_{\text{val}} = 2000$).

**Risk scaling.** Theory predicts that in this setting the minimax risk decays as

$$L^{\star}(n) \ \approx \ \exp\big(-c\,(\log n)^{\alpha/(\alpha+1)}\big),$$

up to multiplicative constants. To compare with this prediction, for each $\alpha$ we fit the parametric form

$$\log L(n) \ \approx \ A_\alpha \ - \ C_\alpha \,(\log n)^{\alpha/(\alpha+1)}$$

by least squares over $(A_\alpha, C_\alpha)$ using the measured pairs $\{(\log n_i, \log L(n_i))\}_i$. Figure 4 shows $\log L(n)$ against $(\log n)^{\alpha/(\alpha+1)}$ together with the fitted curves.

As a minimal sanity check, this synthetic experiment (Fig. 4) in which varying the spectral decay parameter $\alpha$ systematically affects the convergence speed: heavier-tailed spectra (smaller $\alpha$) lead to visibly slower decay of the empirical risk. This is qualitatively consistent with the theoretical prediction, although we do not attempt to match the precise asymptotic rate.

---

[3] In our theory, we did not explicitly investigated such an cutoff or the normalization for simplicity.

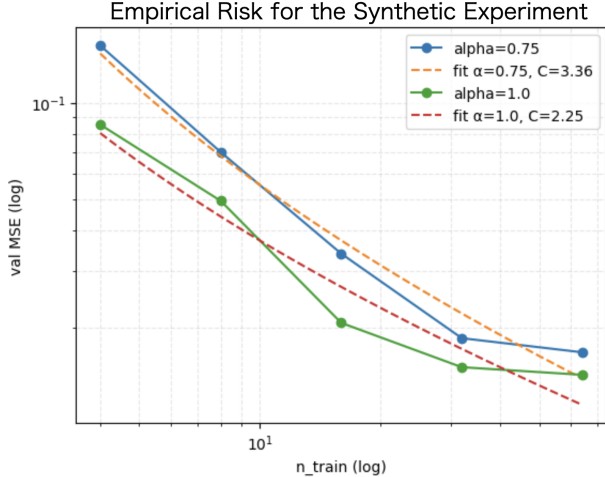

Figure 4: Empirical risk $L(n)$ for the synthetic measure-valued experiment, plotted on a transformed axis $(\log n)^{\alpha/(\alpha+1)}$ together with the fitted curves. Each risk was calculated with 2000 unknown samples.

**Attention-weight analysis.** To check whether the attention layer actually uses the query tag, we inspect the softmax attention weights of the trained model on the validation set. For a given example, let

$$\{(X_t, V_t)\}_{t=1}^{T} \in \big([0,1] \times \{-1,+1\}\big)^{T}$$

denote the $T$ context tokens, and let

$$(X_q, V_q) = (0, v^{(1)})$$

be the query token appended at the end of the sequence. For each attention head $h = 1, \ldots, H$ we write

$$a^{(h)} \in [0,1]^{T}$$

for the softmax attention weights from the query to the $T$ context positions, so that

$$\sum_{t=1}^{T} a_t^{(h)} = 1.$$

We are interested in how the query token redistributes its attention mass over tokens whose tag matches the query versus those with the opposite tag. Accordingly, we define the index sets

$$S_{\text{same}} := \{1 \leq t \leq T : V_t = V_q\}, \qquad S_{\text{diff}} := \{1 \leq t \leq T : V_t \neq V_q\},$$

that is, we only consider the context tokens and exclude the query token itself from both sets. For each head $h$ we then compute the average per-token attention weight assigned by the query to same-tag and different-tag tokens,

$$\bar{w}_{\text{same}}^{(h)} := \frac{1}{|S_{\text{same}}|} \sum_{t \in S_{\text{same}}} a_t^{(h)}, \qquad \bar{w}_{\text{diff}}^{(h)} := \frac{1}{|S_{\text{diff}}|} \sum_{t \in S_{\text{diff}}} a_t^{(h)},$$

as well as the total attention mass

$$m_{\text{same}}^{(h)} := \sum_{t \in S_{\text{same}}} a_t^{(h)}, \qquad m_{\text{diff}}^{(h)} := \sum_{t \in S_{\text{diff}}} a_t^{(h)}.$$

In practice, we implement this by adding a flag to the attention module that, when enabled, stores the last softmax attention tensor $A \in \mathbb{R}^{B \times H \times 1 \times T}$ on the CPU after a forward pass (where $B$ is the batch size), and we extract $a^{(h)}$ as the length-$T$ vector $A_{b,h,1,:}$ corresponding to the query-to-context weights for each example $b$ and head $h$.

Table 1: Average attention weight from the query token to same-tag vs. different-tag context tokens (mean and standard deviation over 1000 validation examples, with 64 training data and $\alpha = 1.0$). With $T_{\text{ctx}} = 5000$ and roughly half of the tokens sharing the query tag, the uniform baseline is $\bar{w} \approx 2 \times 10^{-4}$, while a head that focuses almost exclusively on same-tag tokens reaches $\bar{w}_{\text{same}} \approx 4 \times 10^{-4}$. We also report the validation MSE with the original queries and after shuffling queries within 1000 data for validation.

| Head | $\bar{w}_{\text{same}}$ | $\bar{w}_{\text{diff}}$ | $\text{std}(w_{\text{same}})$ | $\text{std}(w_{\text{diff}})$ |
|---|---|---|---|---|
| 0 | $1.96 \times 10^{-4}$ | $2.04 \times 10^{-4}$ | $2.80 \times 10^{-4}$ | $2.85 \times 10^{-4}$ |
| 1 | $1.44 \times 10^{-19}$ | $4.00 \times 10^{-4}$ | $2.10 \times 10^{-19}$ | $4.00 \times 10^{-4}$ |
| 2 | $\mathbf{4.00 \times 10^{-4}}$ | $\mathbf{1.17 \times 10^{-14}}$ | $4.00 \times 10^{-4}$ | $1.70 \times 10^{-14}$ |
| 3 | $\mathbf{4.00 \times 10^{-4}}$ | $\mathbf{2.86 \times 10^{-27}}$ | $4.00 \times 10^{-4}$ | 0 (too small) |
| mean | $2.49 \times 10^{-4}$ | $1.51 \times 10^{-4}$ | $2.70 \times 10^{-4}$ | $1.71 \times 10^{-4}$ |

|  | original queries | shuffled queries |
|---|---|---|
| val MSE | $1.44 \times 10^{-2}$ | $7.75 \times 10^{-1}$ |

We report in Table 1 the mean and standard deviation of $m_{\text{same}}^{(h)}$ and $m_{\text{diff}}^{(h)}$ over 1000 validation examples for each attention head $h$. On this synthetic task, two of the four heads concentrate essentially all of their attention mass on tokens whose tag matches the query tag, while another head exhibits the opposite preference and one head remains nearly symmetric. Averaged across heads, the query token assigns a larger total mass to tokens with the same tag than to those with the opposite tag, indicating a net bias toward tag-conditioned retrieval.

The absolute scale of the averaged weights $\bar{w}_{\text{same}}$ and $\bar{w}_{\text{diff}}$ is small (on the order of $10^{-4}$) simply because the attention distribution is normalized over a long context of $T_{\text{ctx}} \approx n_{\text{tokens}} = 5000$ positions. Under an approximately uniform baseline, we would have

$$\bar{w}_{\text{unif}} \approx \frac{1}{T_{\text{ctx}}} \approx \frac{1}{5000} \approx 2 \times 10^{-4},$$

so the reported values should be interpreted relative to this $1/T_{\text{ctx}}$ scale rather than as absolute probabilities. In our construction, each context token independently comes from $\mu_1 \otimes \delta_{v^{(1)}}$ or $\mu_2 \otimes \delta_{v^{(2)}}$ with probability $1/2$, so typically $|S_{\text{same}}| \approx |S_{\text{diff}}| \approx T_{\text{ctx}}/2$. Consequently, values around $\bar{w} \approx 2 \times 10^{-4}$ correspond to almost-uniform attention over all context tokens, whereas values around $\bar{w}_{\text{same}} \approx 4 \times 10^{-4}$ and $\bar{w}_{\text{diff}} \approx 0$ indicate that a head places essentially all of its attention mass on the same-tag subset (and analogously for the opposite tag).

As a sanity check that the model genuinely uses the query input, we perform a "query shuffle" experiment at evaluation time: within each mini-batch we randomly permute the last (query) token across examples, while keeping the context tokens and targets fixed, and recompute the validation loss (the bottom of Table 1). On this synthetic task, shuffling the query tokens increases the validation MSE, confirming that the model relies nontrivially on the query input.

This minimal experiment is not intended as a thorough empirical study, but it provides a sanity check that the qualitative order of the risk predicted by our theory is reproducible in a simple measure-valued attention setting.

