# OpenReview forum: "Transformers as Measure-Theoretic Associative Memory: A Statistical Perspective and Minimax Optimality"
_ICLR.cc/2026/Conference — ICLR 2026 Poster_

### Official Review · Reviewer_coHc · 2025-10-28

**Soundness:** 4
**Presentation:** 3
**Contribution:** 3
**Rating:** 8
**Confidence:** 3

**Summary:**

In the paper, the authors cast associative memory in transformers as a problem in retrieving the correct component from a mixture distribution via querying. The authors show that a (measure-theoretic) transformer trained on empirical risk minimization is able to implement via an approximately one-hot softmax the optimal target function with sub-polynomial rate.

**Strengths:**

The authors carry out a sophisticated theoretical analysis of the problem. Particularly nice is the tight characterization of the excess-risk rate. Despite the complex nature of the work, the paper is relatively well written and organized. I especially like that the authors reserved some space to give a quick proof sketch for theorems in the main body. The proofs seem reasonable, even though I could not check the appendix in detail.

**Weaknesses:**

The limit of the work is obviously the lack of clear practical implications, being a purely theoretical work. The lack of experimental results is also a consequence of the same fact. Nonetheless, the results are interesting and a worthy theoretical contribution.

On the minor side:
1. Lines 176-177 are repeated at lines 178-179;
2. The definition at line 211 is confusing: there is no F in the argument;
3. I would take a little space to explain what RKHS in Sec. 3.1.1;
4. There is an extra "that" at line 406.

**Questions:**

1.Do you think that this work could be relevant and practically useful for the community in any way? More specifically, why do you think that an infinite-dimensional context in the form of a mixture distribution can be meaningful in practice?
2. The assumptions are not discussed extensively and they seem quite strict. In the conclusion you mention the possibility to extend your results to broader regimes such as polynomial instead of exponential regimes. What is currently the main difficulty in carrying out such a generalization?

---

> ### Author Response · Authors · 2025-11-21
> **Rebuttal by Authors**
>
> Dear Reviewer coHc,
>
> We thank Reviewer **coHc** for the careful reading and helpful comments. For issues that are common across reviewers (measure-theoretic formulation, spectral decay, experiments), we provide a detailed response in the **Official Comment**; here we only summarize points that are more specific to your report, especially the meaning of the “mixture” in our setting.
>
> ---
>
> ### Meaning of measures and mixtures
>
> Regarding the use of **measures**, we refer to the Official Comment for details. Very briefly, contexts are modeled as probability measures to obtain a length-independent, permutation-invariant notion of "context embedding," on which we can use Wasserstein/spectral tools.
>
> More specifically for the **mixture** structure: Intuitively, each component measure $\mu_{v^{(i)}}$ represents the token
> distribution of a single document $i$, with a fixed document-level feature
> $v^{(i)}$ and content distribution $\mu^{(i)}_0$ . The mixture
> $$
> \nu = \sum\_{i=1}^{I} w_i \,\mu^{(i)}\_{v^{(i)}},
>   \qquad w\_i \ge 0,\ \sum\_{i} w_i = 1,
> $$
> should then be read as the law of the following two-stage sampling procedure: first pick a document index $i$ with probability $w_i$, and then sample a token $x = (v^{(i)}, Z)$ with $Z \sim \mu^{(i)}_0$. In other words, $\nu$ encodes a large corpus where tokens coming from document $i$ appear with relative frequency $w_i$.
>
> Given such a mixture $\nu$ and a query $x_{\mathrm{q}}$ whose first $d_1$ coordinates align with some $v^{(i^\star)}$ , we consider the following associative memory behavior. Intuitively, $x_{\mathrm{q}}$ plays the role of a question that is *tagged* with the document-level feature $v^{(i^\star)}$ , i.e., a query that explicitly asks about document $i^\star$ rather than the rest of the corpus. From the aggregated context $\nu$, the model should then effectively "lock on" to the single component $\mu^{(i^\star)}\_0$ selected by $x_{\mathrm{q}}$ . It should finally predict a scalar quantity that depends only on this underlying content distribution.
>
>
> We have added a short explanation in the introduction and the problem-setting section making this semantics explicit, so that the role of the mixture and its relationship to associative recall is clearer.
>
> ---
>
> ### Spectral assumptions
>
> Your comments on the spectral decay assumption are addressed in detail in the **Official Comment**. In brief, we:
> - justify the **exponential** decay assumption via standard kernels such as the heat kernel / Gaussian on bounded domains; and
> - briefly note that **polynomial** decay is a natural next step beyond this initial exponential setting, which we defer to future work to keep the current exposition focused and simple.
>
> ---
>
> ### Experiments
>
> As explained in the **Official Comment**, we have added a **minimal synthetic experiment (Appendix D)** that instantiates our assumptions in a simple setting and tracks excess risk as the sample size grows, highlighting the dependence on the spectral decay parameter $\alpha$ and illustrating the attention behavior in a simple setting.
>
> ---
>
> ### Typos and notation
>
> Finally, we have corrected the typos and minor notational inconsistencies you pointed out and slightly streamlined the notation to improve readability.
>
> We are grateful for your feedback, which helped us clarify the interpretation of both the measure and mixture structures and sharpen the overall presentation.

---

> > ### Comment · Reviewer_coHc · 2025-11-26
> >
> > I would like to thank the authors for their thorough response. I also appreciate the revised paper, which greatly improves the presentation. Therefore I confirm my score and my positive assessment of the paper.

---

### Official Review · Reviewer_wjf3 · 2025-10-30

**Soundness:** 2
**Presentation:** 1
**Contribution:** 2
**Rating:** 4
**Confidence:** 2

**Summary:**

The paper develops a measure-theoretic view of Transformers for
associative recall.
Instead of treating a context as a finite token sequence, the input is a
mixture of probability measures over tokens; a query identifies the
relevant component measure (recall), and the predictor maps that
recalled measure (plus the query) to an output (predict).
Technically, attention is lifted to an integral operator on measures, so
sums over tokens become integrals.
Under a spectral assumption, i.e. Mercer eigenvalues of the RKHS
kernel decay exponentially, the authors construct a shallow (depth-2)
Transformer+MLP and prove a sub-polynomial excess-risk bound of the
form
$\exp\{-\Theta((\log n)^{\alpha/(\alpha+1)})\}$ for ERM, and provide a
matching minimax lower bound (same exponent, up to constants).
They argue learned softmax attention can implement near one-hot,
query-adaptive recall that isolates the correct component measure and
aggregates a small set of its Mercer coefficients for prediction.

**Strengths:**

To the extent of my understanding, which is limited, the modeling of contexts as probability distributions seems an interesting line of research, with several other recent papers treating it.

**Weaknesses:**

Clarity:
I found the paper extremely hard to parse (which may be my fault, and not a weakness, I let the area chair judge). For example:
- line 73: it is not clear what is meant by "context"
- line 74: it is not clear on which space do the measuere nu, mu^(i) live in
- line 75: it is not clear which space the query x_q lives in
- line 76: what is the "target map". Is it some sort of ground-truth function? This should be defined clearly
- etc...
While stated as "informal", I find this introductory section quite confusing.
The paragraph in line 149-167 is not illuminating either on this, and one needs to reach line 169 to finally know what are the objects we are speaking about.
While the paragraph starting at line 169 provides definition, they are hard to understand: I would suggest providing examples here!

More generally, important intuitions (effective dimension $(\log n)^{1/(1+\alpha)}$, role of the second attention block, sensitivity to query–key geometry) are
scattered across sections and appendices, hampering readability for a
broader audience.

Model:
The query explicitly contains the relevant tag $v^{(i^\*)}$ and the tags
are well-separated (inner products $\le 0$), making recall potentially substantially
easier than in natural setting.

Lack of any empirical validation:
The work is purely theoretical. Even small
synthetic experiments would help test the sharpness/spikiness of learned
softmax and test the assumptions (e.g., performance deterioration
when spectra are polynomial).

**Questions:**

Polynomial spectra. Can you extend (even heuristically) the analysis to
kernels with polynomial eigen-decay? Which proof steps break and what
rates would you expect?

Query robustness. What happens if $x_q$ does explicitly contain
$v^{(i^\*)}$ or if the tags are weakly separated/noisy? Can the softmax
gate still become sufficiently spiky under ERM?

Empirics. Could you add synthetic experiments varying $\alpha$ and
$I$, and compare learned softmax vs. linear attention / frozen kernels, to
illustrate the theory?

typos:
- line 179 repeats line 176
- line 180: \Chi_0 not defined
- line 211: inside the argmin should be F? Same in line 214

---

> ### Author Response · Authors · 2025-11-21
> **Rebuttal by Authors**
>
> Dear Reviewer wjf3,
>
> We thank Reviewer wjf3 for the careful reading and constructive feedback. Below we summarize our main clarifications and changes. For points that are common across reviewers (measure-theoretic formulation, spectral decay, experiments), we refer to the **Official Comment**.
>
> ---
>
> ### Presentation, typos, and notation
>
> - We have streamlined the **problem setting** with a clearer narrative and a simple running example so that the mathematical structure is easier to follow on a first read.
> - We now explicitly introduce and distinguish **context**, **query**, **ground-truth functions**, and the relevant **measure spaces** before stating the main results, and we collect the key intuitions needed to understand the phenomena behind the theorems in one place.
> - Typographical and notational issues you pointed out have been corrected, and we slightly simplified some notation to improve readability.
>
> ---
>
> ### Model assumptions and tag inner products
>
> You raised concerns about the strength and realism of the assumptions on tags, in particular the explicit tagging and the negative/orthogonal inner-product structure.
>
> - In the current work, we intentionally adopt a **clean and somewhat idealized tag structure** to keep the analysis tractable and to focus on the measure-theoretic aspects. Assumptions of a similar flavor on inner products also appear in recent work on factual knowledge extraction and finetuning (e.g., Ghosal et al., 2024), so our setting is aligned with existing theoretical models of factual knowledge extraction.
>
> - Questions about *query robustness*—e.g., when the query does not perfectly expose the relevant tag or tags are only weakly separated—are beyond the scope of the present work.
>
> - We agree that the inner-product assumption can be relaxed. A natural direction, inspired by Bietti et al. (2023), is to treat tags as **high-dimensional random vectors**, e.g., uniformly drawn on the sphere, so that they are only **approximately orthogonal**. In such a setting, inner products scale as $O(1/\sqrt{d_1})$ in the tag dimension $d_1$. We expect that our arguments can be adapted to this regime, with the error terms controlled via these $O(1/\sqrt{d_1})$ correlations.
>
> - In the present paper, however, we chose to **focus on an $O(1)$-dimensional tag space** so as not to dilute our main message about associative recall over measures and its statistical complexity. Extending the theory to high-dimensional random tags is a promising direction for future work in order to keep the present exposition focused.
>
> We have added a short explanation in the main text explaining this modeling choice, its relation to prior work.
>
> ---
>
> ### Experiments
>
> Following your suggestion, we added a **minimal synthetic experiment (Appendix D)** that instantiates our assumptions in a simple setting and tracks the excess risk as the sample size grows. As explained in the Official Comment, this experiment is deliberately minimal and is used to illustrate how the convergence behavior depends on the spectral decay parameter $\alpha$ and to demonstrate the associated softmax attention patterns.
>
> ---
>
> ### Spectral decay
>
> Your comments on the spectral decay assumption are addressed in detail in the **Official Comment**. Briefly, we now:
>
> - justify the **exponential** decay assumption via standard kernels such as the heat kernel / Gaussian on bounded domains, and
> - briefly note that **polynomial** decay would be a natural next step beyond this first-step exponential setting, which we leave to future work to keep the present exposition focused and simple.
>
> We are grateful for your comments, which helped us clarify the presentation, better explain the modeling choices around tags, and make the scope and limitations of our assumptions more transparent.
>
> ---
>
> **References (cited in this response)**
>
> - Ghosal, G., Hashimoto, T., & Raghunathan, A. (2024). *Understanding finetuning for factual knowledge extraction*. arXiv:2406.14785.
> - Bietti, A., Cabannes, V., Bouchacourt, D., Jegou, H., & Bottou, L. (2023). *Birth of a transformer: A memory viewpoint*. NeurIPS 36, 1560–1588.

---

> > ### Comment · Reviewer_wjf3 · 2025-11-25
> >
> > I thank the authors for engaging with my comments. The authors made a noticeable effort to improve the presentation, which is now clearer and was the main roadblock to understanding the material for me. I will raise my score to 6.

---

> > > ### Author Response · Authors · 2025-11-26
> > >
> > > Dear reviewer wjf3,
> > >
> > > Thank you very much for taking the time to re-read our paper and for your helpful feedback — we are glad to hear that the revised presentation is clearer.

---

### Official Review · Reviewer_rkiJ · 2025-10-30

**Soundness:** 2
**Presentation:** 1
**Contribution:** 1
**Rating:** 2
**Confidence:** 3

**Summary:**

The paper attempts to study the associative memory problem theoretically. It formulates the "context" to a transformer as a mixture of measures.  The goal of the transformer is, on reading a query corresponding to one of these constituent measures, to identify the associated measure.

**Strengths:**

The paper attempts a theoretical analysis of a practically relevant setup in transformers, which is always welcome and appreciated.  It presents matching upper and lower bounds to the risk of a statistical estimation problem that they pose as a proxy to the associative memory problem.

**Weaknesses:**

The biggest issue of this paper is that the mathematical notation is (a) far too dense for a conference such as ICLR, and (b) just poor, objectively (there are too many to list, but see Questions for a few).  At a high level, the paper reads as follows to me: the paper spend 7 pages (!) setting up notation and making several assumptions (some of which, admittedly, do seem reasonable, but, e.g., the eigenvalue decay rate, just show up out of nowhere) and then obtains some results that likely do not hold if any of the assumptions are violated even slightly (e.g., if the eigenvalue decay rate is different).

Is there really a need for the entire paper to be measure-theoretic?  What would be different if we just assumed these were just a linear combinations "discrete distributions" of some finite set and then went with it?  The insistence on measure theory and functional-analytic analysis makes the paper very specialized, so if it could be simplified at all, it would be greatly appreciated (I understand that often we do actually require this level of sophistication to analyze some setups, but this setup feels like a more interesting analysis could be made with simpler mathematics).

What exactly is the statistical estimation problem introduced in Section 3?  As far as I understand, it is an abstraction of the memory recall problem (I might be wrong, but this is what I understood from the paper).  In this case, what exactly is the value of such an abstraction and analysis?  I do not think there is value in knowing at exactly what rate it decays, given that the problem itself is a "fake" problem invented only a a proxy.  I think there would be value if the paper were able to provide insights/explain some curious phenomena on the general problem of associate memory and recall, but I do not see any such insights.

**Questions:**

1. Why this choice of eigenvalue decay (lambda_j = exp(-c j^alpha))?  This seems rather arbitrary and I do not see any justification/explanation for it.  More importantly, this alpha is crucial and shows up in the final results, making them very fragile (not robust to this assumption).

2. Example 1 lists some examples satisfying the assumptions, but it is not clear if these examples satisfy all or some subset of the assumptions, and whether these examples are interesting for the setup being considered (e.g., one could list any arbitrary assumption and say that there are functions satisfying that assumption, but whether it makes sense to consider that example for that setup is important).  Comments?

Minor comments:
1. Equation at line 211: should be F, not F^\star
2.  Section 3.2, one of the expressions for either SAttn or MSAttn must be wrong: both have a V^h x at the end, and MSAttn does not have a W^h  (guessing that SAttn is right and MSAttn has W instead of V)?
3. Definition 5, the equation: what is nu_1, did you mean mu_1?

---

> ### Author Response · Authors · 2025-11-21
> **Rebuttal by Authors**
>
> Dear Reviewer rkiJ,
>
> We thank Reviewer **rkiJ** for the careful reading and many insightful comments. Below we summarize our main clarifications and changes; for points common to multiple reviewers, we refer to the **Official Comment**.
>
> ---
>
> ### Presentation
>
> - We have streamlined the introduction and the problem setting with a clearer narrative and a simple running example.
> - Purely technical assumptions that do not directly affect the rates are now only explained informally in the main text, with full statements deferred to the **appendix**.
>
> ---
>
> ### Measure-theoretic settings
>
> Since these points are discussed in detail in the **Official Comment**, we only note here that representing token sequences as probability measures gives a length-independent, permutation-invariant notion of context and enables the use of standard tools such as Wasserstein distances and spectral analysis.
>
> ---
>
> ### Insights from our estimation problem
>
> You asked whether our estimation problem is just a proxy or genuinely captures the difficulty of associative measure recall.
>
> - We agree that it is an **abstraction**: the object to be recalled is a **measure**, but in standard ML pipelines the final output is usually a **real number or vector**, trained with losses such as squared loss. It is therefore natural to study real-valued functionals of
>   - a mixture of measures (context), and
>   - a query specifying which associative measure should be recalled,
>   rather than reconstructing the full associative measure itself, in line with associative-memory–based sequence models where recalled vectors serve as intermediate representations (Ramsauer et al., 2021).
>
> - Our analysis, however, goes beyond a formal proxy. Around **Lemma 8**, we show that by combining
>   (i) "classical" associative memory on samples from the measure, and
>   (ii) **softmax attention** forming a sparsely weighted average,
>   the model can **indirectly reconstruct the associative measure** in a parameter-efficient way.
>
> - A key structural point is that information extraction occurs through **integrals over the entire measure**, which makes the “peaked” (near one-hot) behavior of softmax weights essential. This provides a conceptual explanation of the routing/gating role of attention.
>
> - The learning rate in **Theorem 1** is obtained by analyzing this associative-measure–recall mechanism, so the estimation problem is not merely an ad hoc proxy but captures key aspects of the underlying recall behavior.
>
> We have revised the corresponding discussion to make this connection and these insights more explicit.
>
> ---
>
> ### Dependence of the statistical rate on spectral decay (Q1)
>
> You raised an important question about the role and robustness of the spectral decay parameter $\alpha$.
>
> - It is natural that $\alpha$ appears in the rate: in our setting, $\alpha$ controls the **effective complexity** of the measures. Roughly, the information can be represented in an **effective dimension** of order $(\log n)^{1/(\alpha+1)}$. Plugging this into classical nonparametric rates of the form $n^{-\beta/(2\beta + d)}$ (for $\beta$-H\"{o}lder smooth functions, in Schmidt-Hieber, 2020) recovers the sub-polynomial behavior we obtain, so our bounds are consistent with the classical picture once this effective dimension is made explicit.
>
> - As for robustness, it is standard in statistical learning that different **problem classes** (here indexed by $\alpha$) have different **minimax rates** (e.g., Schmidt-Hieber, 2020). One first fixes such a class and then asks (i) what the minimax rate is, and (ii) whether a concrete method achieves it. Our contribution is to show that, under the spectral assumption encoded by $\alpha$, a **transformer-based model** attains the corresponding minimax rate. Thus, the dependence on $\alpha$ reflects the intrinsic difficulty of the problem class rather than a fragile artifact of our analysis.
>
> We have added a short discussion on effective dimension and the role of $\alpha$ to make this clearer in the main text.
>
> ---
>
> ### Choice of spectral decay and examples (Q2)
>
> The choice of exponential spectral decay, concrete kernel examples, and remarks on polynomial decay are discussed in more detail in the **Official Comment**, to which we refer for all technical details.
>
> ---
>
> ### Experiments
>
> Following your suggestion, we conducted a minimal synthetic experiment; for brevity, we refer you to the **Official Comment** for a concise description of the setup and findings.
>
> We are very grateful for your thoughtful and constructive feedback, which has helped us substantially clarify both the conceptual and technical aspects of the paper.
>
> ---
>
> **References**
>
> - Ramsauer, H., et al. (2021). Hopfield Networks is All You Need. In ICLR, 2021.
> - Schmidt-Hieber, J. (2020). Nonparametric regression using deep neural networks with ReLU activation function. The Annals of Statistics, 48(4).

---

> > ### Comment · Reviewer_rkiJ · 2025-11-24
> > **concerns addressed, thanks!**
> >
> > I thank the authors very much for taking the effort to update the paper.  They have succeeding in addressing my concerns, namely:
> >
> > 1. the presentation of the setup and results (the paper is now rather pleasant to read, and in fact, being able to cleanly formalize the setup of associative recall is now a strength and contribution of this paper),
> > 2. the reason for the measure-theoretic framework (there was nothing inherently bad about using a measure-theoretic framework, but it being so difficult to read and parse made me wonder if it was necessary to be so dense --- the improved presentation makes this easier to digest),
> > 4. the relevance of the assumptions (moving all the details to the appendix and keeping only the essentials before the results greatly helps, the heat kernel example is also well-explained),
> > 3. the role of the statistical regression problem (not merely as a proxy, but as the concrete goal of the recall problem) and value of the results (of course, the minimax rate was certainly always going to depend on alpha, but now that this alpha seems less arbitrary and the regression problem feels more concrete and relevant to the problem, the result does not feel "empty" or "artificial").
> >
> > Thus, I will raise my score to 8 (review will be updated at the end of the discussion phase):  The paper is well-written, studies the relevant problem of associative recall, formalizes it using a solid mathematical framework, and shows the meaningful result that ERM + depth-2 transformer with MLP is in fact minimax optimal for this task (and for completeness, a minimal synthetic experiment also verifies the theoretical claims).
> >
> > A minor comment: Figure 1 seems to have been updated --- the third box refers to mu_0^{(2)}, this should surely be mu_0^{i^*}.

---

> > > ### Author Response · Authors · 2025-11-25
> > >
> > > Dear Reviewer rkiJ,
> > >
> > > Thank you very much for carefully reading the revised version and for your positive follow-up. We are glad that the updated exposition, the motivation for the measure-theoretic framework, and the discussion of the regression formulation helped address your concerns. We have also corrected Figure 1 so that the third box now reads $\mu_0^{(i^*)}$ as you suggested.

---

### Author Response · Authors · 2025-11-21
**Official Comment**

Dear Reviewers,

We sincerely thank all reviewers for their thoughtful and detailed feedback. We have revised the paper accordingly, and below we summarize the main clarifications and changes that are common across the reviews.

### Measure-theoretic settings

Our use of probability measures is **not** to make the setting artificially abstract, but to formalize in a clean way what is already implicit in practice.

- Conceptually, representing a token sequence as a measure is analogous to representing a whole document by its empirical word distribution, essentially a document-level embedding. If we want a mathematically precise notion of "context embedding" that does **not** depend on the context length, using probability measures is a natural and robust choice.
- Technically, prior theoretical work already treats self-attention as an **integral operator** on function spaces to study expressivity and generalization independently of sequence length (e.g., Vuckovic et al., 2020). Our framework follows this line rather than introducing measure theory for its own sake.
- In our paper, the measure appears as the **limit of empirical distributions** of a finite token multiset: as the number of tokens $N$ grows, the law of large numbers allows us to represent a very large token collection by a single measure, so the problem can be formulated independently of $N$.

The advantages of working directly with measures are:

1. We can define similarity between contexts via distances between measures (e.g., Wasserstein, MMD), instead of ad hoc sequence-level metrics.
2. We can exploit smooth densities and their spectral (Mercer) expansions.
3. In the unmasked setting, permutation invariance of the context is built in: a measure is inherently insensitive to token order.

By staying entirely in the "world of measures," we can safely introduce distances between sets of measures and notions such as covering numbers and effective dimension, and use them to **quantify**, both:
- the smoothness of functions that take token sets as input, and
- the intrinsic difficulty (minimax rates) of the associated estimation problem.

### Spectral decay

Regarding the spectral decay assumption, we now clarify its role and give concrete examples.

- As a first-step setting, we assume **exponential eigenvalue decay**, inspired by widely used kernels such as the **Gaussian kernel**. In our bounded-domain context, this corresponds to truncated Gaussian-type kernels.
- As a rigorous example compatible with our assumptions, we emphasize the **heat kernel** on $[0,1]$. The Laplace operator $d^2/dx^2$ has trigonometric eigenfunctions with eigenvalues proportional to $j^2$ , and the associated heat kernel has spectrum $\exp(-c j^2)$, i.e., $\lambda_j \simeq \exp(-c j^\alpha)$ with $\alpha = 2$ . This is a standard textbook example (Grigor’yan, 2006) and serves as a natural toy model for our theory.
- In this work we deliberately focus on the **exponential** eigen-decay case as a clean first step. For **polynomial** decay, $\lambda_j \simeq j^{-2\alpha}$, the covering entropy of the model class in Section 3 is already known to satisfy a lower bound of order $\exp(\Omega(\varepsilon^{-1/(\alpha+1)}))$ (Lanthaler, 2024) and an upper bound of order $\exp(O(\varepsilon^{-1/\alpha}))$ (via Boissard, 2011). We expect the upper-bound argument in **Theorem 1** to extend to this setting with moderate changes, but closing this gap to obtain a matching **minimax lower bound** (Theorem 2) is technically nontrivial, so we leave a full treatment of the polynomial regime to future work.

### Experiments

Our main contribution is theoretical, but we add a **small synthetic experiment (Appendix D)** as a sanity check. In a controlled setting, we vary the spectral decay parameter $\alpha$ and observe faster excess-risk decay for larger $\alpha$, and we inspect the learned softmax attention, which tends to concentrate on tokens associated with the queried component.

### Typos and notation

We thank the reviewers for carefully pointing out several typographical and notational issues. We have corrected all such issues and streamlined the notation to improve readability.

Once again, we are very grateful to all reviewers for their insightful comments and suggestions, which have significantly improved the clarity and presentation of our work.

---

- Vuckovic, J., Baratin, A., & Combes, R. T. D. (2020). A mathematical theory of attention. arXiv preprint arXiv:2007.02876.
- Grigor’yan, A. (2006). *Heat kernels on weighted manifolds and applications*. Contemp. Math. 398.
- Lanthaler, S. (2024). Operator learning of lipschitz operators: An information-theoretic perspective. arXiv preprint arXiv:2406.18794.
- Boissard, E. (2011). Simple bounds for the convergence of empirical and occupation measures in 1-Wasserstein distance. *Electronic Journal of Probability, 16*.

---

### Author Response · Authors · 2025-11-30
**Summary for the Area Chair**

As the official reviews currently reflect their pre-discussion versions, we would like to briefly summarize the main points that were raised and how we addressed them, so that the paper can be assessed in light of the full rebuttal and discussion. During the discussion phase, after several concerns were clarified, the reviewers’ overall assessments became significantly more positive (score changes: rkiJ 2→8, wjf3 4→6, coHc 8→8).

----

Across the reviews, the main concerns were about
- (i) the motivation for the measure-theoretic formulation and its implications for statistical learning, and
- (ii) the clarity of the presentation, including how the theory is illustrated empirically.

In our response and proposed revisions, we have

- clarified why a measure-theoretic viewpoint is natural for studying associative recall at the level of distributions (rather than individual tokens), and how this connects to classical notions such as learning rates and minimax optimality;
- reorganized the exposition and added intuitive explanations before the technical measure-theoretic development, moving some details to the appendix to keep the main text readable; and
- newly added a small experimental section in the appendix with simple synthetic experiments that qualitatively illustrate the phenomena predicted by our theory.

Reviewers also raised more technical questions about the level of idealization in our setup, in particular the use of well-separated (essentially orthogonal) tags/queries in the associative recall task and the assumption of
exponentially decaying eigenvalues. We clarified that these are simplifying assumptions made to isolate the core statistical and measure-theoretic aspects of the problem and to keep the analysis tractable, and we now
explicitly acknowledge them as idealizations and limitations. Extending the theory to more realistic, noisy, or only weakly separated tags/queries and to more general spectral decay conditions is an interesting direction for future work, but is beyond the scope of the present paper.

----

To the best of our understanding, these steps address the main concerns raised in the reviews and explain why the reviewers’ impressions improved during the discussion phase. We kindly ask that our response and the prior discussion be taken into account when assessing the paper.

---

### Meta-Review · Area_Chair_TnZd · 2026-01-06

**Summary:**

This paper presents a conceptually elegant and technically strong framework that recasts associative memory in Transformers at the level of probability measures. Modeling context as a distribution and attention as an integral operator is both natural and insightful, and the resulting recall and predict decomposition clarifies the role of attention in long context reasoning. The analysis of learned softmax attention, rather than a fixed kernel, trained via empirical risk minimization is particularly compelling and aligns well with modern practice.

The theoretical results are rigorous and sharp. The recovery guarantee under a spectral assumption is clean, and the matching minimax lower bound convincingly establishes optimality of the obtained rates. Overall, the work provides a principled and general foundation for understanding content addressable retrieval in Transformers, with clear implications for both theory and architecture design. This represents a meaningful advance and merits acceptance.

**Reviewer Scores:**

can't predict

---

### Decision · Program_Chairs · 2026-01-26

Accept (Poster)